# A semantics, energy-based approach to automate biomodel composition

**Niloofar Shahidi**[ID][1]*, **Michael Pan**[ID][2,3,4], **Kenneth Tran**[ID][1], **Edmund J. Crampin**[2,3,4,5†], **David P. Nickerson**[ID][1]

**1** Auckland Bioengineering Institute, The University of Auckland, Auckland, New Zealand, **2** Systems Biology Laboratory, School of Mathematics and Statistics, and Department of Biomedical Engineering, University of Melbourne, Melbourne, Victoria, Australia, **3** ARC Centre of Excellence in Convergent Bio-Nano Science and Technology, Faculty of Engineering and Information Technology, University of Melbourne, Melbourne, Victoria, Australia, **4** School of Mathematics and Statistics, Faculty of Science, University of Melbourne, Victoria, Australia, **5** School of Medicine, University of Melbourne, Melbourne, Victoria, Australia

† Deceased.
* nsha457@aucklanduni.ac.nz

## Abstract

Hierarchical modelling is essential to achieving complex, large-scale models. However, not all modelling schemes support hierarchical composition, and correctly mapping points of connection between models requires comprehensive knowledge of each model's components and assumptions. To address these challenges in integrating biosimulation models, we propose an approach to automatically and confidently compose biosimulation models. The approach uses bond graphs to combine aspects of physical and thermodynamics-based modelling with biological semantics. We improved on existing approaches by using semantic annotations to automate the recognition of common components. The approach is illustrated by coupling a model of the Ras-MAPK cascade to a model of the upstream activation of EGFR. Through this methodology, we aim to assist researchers and modellers in readily having access to more comprehensive biological systems models.

**Data Availability Statement:** The reference MAPK cascade model is available from: https://github.com/mic-pan/Modularity-SysBio The reference model of the EGFR pathway is available from: https://models.physiomeproject.org/e/47f/

## 1 Introduction

Modelling complex biological systems such as cells, organs, and organisms allows researchers to integrate and study different aspects of a biological entity, reveal the limits and shortcomings of our knowledge, and obtain new insights into disease treatment [1]. Motivated by such aims, hierarchical modelling is an approach that assists researchers in constructing system-level models, which are continuously expanding in detail, scope, and size [2]. An approach to automatically and hierarchically construct sophisticated models of biology that ultimately leads to the generation of biologically and physically correct models is currently missing.

Hierarchical models are composed of pre-existing smaller models, referred to as modules. Each module can operate and be examined independently, thus reducing model composition errors and facilitating large-scale model generation due to pre-existing models. To accelerate hierarchical model composition, one can capitalise on a myriad of the existing modules created by others. A requirement for this is that the modules be both accessible and reusable [2]. Over

kholodenko_demin_moehren_hoek_1999.cellml/
docgen All the model files for this manuscript are
available on GitHub: https://github.com/Niloofar-
Sh/EGFR_MAPK.

**Funding:** 1. NS was supported by an Aotearoa
Fellowship to DPN from the Aotearoa Foundation.
2. MP was supported by a Postdoctoral Research
Fellowship from the School of Mathematics and
Statistics, University of Melbourne. 3. KT was
supported by a Marsden Fast-Start grant
(UOA1703) from the Royal Society of New Zealand
(https://www.royalsociety.org.nz) and a Sir Charles
Hercus Health Research Fellowship (21/116) from
the Health Research Council of New Zealand
(https://gateway.hrc.govt.nz/funding/career-
development-awards/2021-sir-charles-hercus-
health-research-fellowship). 4. EJC was supported
by the Australian Research Council Centre of
Excellence in Convergent Bio-Nano Science and
Technology (project number CE140100036)
(http://purl.org/au-research/grants/arc/
CE140100036). 5. DPN was supported by an
Aotearoa Fellowship from the Aotearoa Foundation
and the Center for Reproducible Biomedical
Modeling P41 EB023912/EB/NIBIB NIH HHS/
United States (https://projectreporter.nih.gov/
project_description.cfm?projectnumber=
5P41EB023912-03). The funders had no role in
study design, data collection and analysis, decision
to publish, or preparation of the manuscript.

**Competing interests:** The authors have declared
that no competing interests exist.

the past decade biosimulation models have become increasingly accessible on public repositories such as the Physiome Model Repository (PMR) [3] and BioModels [4], which store models in XML-based format such as CellML [5] and the Systems Biology Markup Language (SBML) [6, 7].

The main challenges in hierarchical model composition include: (a) incompatible code languages (using dramatically different modelling languages such as Object-oriented, graphical, and continuous/discrete time), (b) different modelling formalisms (such as using rule-based modelling, differential equations, neural networks, and Boolean networks), (c) post-composition adjustments (manual edits in mathematical equations, rules, or parameters to make the composed models biologically sensible [8]), and (d) physically implausible resultant models. Each integrating system or platform addresses some of these challenges. While existing model integration platforms, such as the SBML Hierarchical package [9] and PySB [10] can resolve issues with code and modelling formalism compatibility, they are limited to the biochemical domain. The resultant model often needs further adjustments to be executable and yet it might not follow the laws of thermodynamics and physics (such as energy, mass, and charge conservation) [11, 12]. This is referred to as physical feasibility. Several formulations and frameworks have been developed to ensure biochemical models follow the laws of thermodynamics [13–15] but most of them are purely mathematical and are often difficult to implement for model composition due to non-standardised rate laws, and lacking an easy append/delete graphical structure. Furthermore, most of the model composition tools are not applicable to multi-physics systems and cannot be generalised to more complex biological systems. One solution to these issues is combining a hierarchical modelling approach (to help with the post-composition adjustments) with an energetic and multi-physics framework that explicitly models energy (expressing kinetic rate laws of biochemical reactions in terms of chemical energy level differences [15]) and ensures adherence to the laws of physics and is executable in multi-physics modelling (such as cardiomyocytes electromechanical coupling). The bond graph approach addresses these issues.

Bond graphs provide a domain-independent hierarchical framework that generates models based on the laws of physics and thermodynamics. Initially introduced by Paynter [16], bond graphs were primarily intended for engineering applications. The application of bond graphs was extended to the chemical domain by Oster et al. [17, 18] and subsequently by Cellier [19]. Gawthrop and Crampin have recently developed the bond graphs framework to model and analyse biochemical and electrochemical systems [12, 20]. Two physical co-variables form the energy-based foundation of bond graphs: *effort* (*e*) and *flow* (*f*). Power is the product of effort and flow ($p = ef$) and energy is the power over time: $E = \int p \, dt$. Effort and flow are general terms that represent voltage and current in the electrical domain, force and velocity in the mechanical domain, chemical potential and molar flux in the chemical domain, respectively. Bond graphs represent systems as graphical representations which consist of a set of elements, *i.e.*, components and junctions. Components represent physical entities (such as ions, complexes, genes, atoms in microscopic level and resistors, capacitors, dampers, and mass in macroscopic level) and are defined as general configurations of electrical, mechanical, or chemical elements. For instance, *C* components in bond graphs are charge storage components *i.e.* capacitors in electrical circuits, springs in mechanical systems, or chemical species in chemical reactions. Energy is conserved and travels between components bidirectionally through bonds (shown by harpoons). A common effort between the components is shown by a '0' junction where the sum of flows equals the outward flow, and a '1' junction shows a common flow where the sum of efforts equals the inward effort. Junctions are analogous to the Kirchhoff's laws in electrical circuits, equilibrium of forces in mechanical systems, and stoichiometry

balance in chemical reactions [21]. These conservation laws in different domains follow the same mathematical principles, hence they can be represented by generalised equations in bond graphs. Readers can find a more detailed description of the bond graph theory in [22–24]. The extension of bond graphs to biochemical domain and the constitutive equation for each component will be explained in more details in Section 2.1.

In the context of biochemistry, modellers widely use traditional kinetic models. However, in general, kinetic models are not thermodynamically consistent (*i.e.* energy conserving) unless the parameters satisfy certain detailed balance constraints. Specifically, detailed balance constraints are required to ensure that biochemical loops have zero flux (*i.e.* dissipate no energy) at equilibrium. These detailed balance constraints become increasingly difficult to derive as biochemical networks become larger. Because bond graph models assign a chemical potential to each species, they automatically adhere to detailed balance constraints. Hence, parameters can be modified without violating thermodynamic consistency [25]. This ensures that model composition respects the constraints on thermodynamics for biochemical systems.

Semantic annotation is labelling the mathematical content of models or data with standard machine-readable descriptions [26]. These are crucial for the reusability and interoperability of models. However, biological and biochemical complexities can give rise to inconsistencies in semantic annotations. Many species and chemical compounds are not simply defined by a single semantic term. Subtle variations of names for species have long been an obstacle for semantics-based merging tools to integrate models based on identifying similarly annotated species. In this paper, we have used identical semantics for same species and leave it to the scientific community to develop a harmonising system for annotating biomodels.

An automated model composition approach significantly assists researchers in creating large-scale models from existing modules [27]. Shahidi et al. [28] introduced a general hierarchical model composition method by encoding bond graph modules in CellML and constructing a complex model using the SemGen merger tool [29]. The SemGen merger tool uses the biological semantics of the components in models to identify and interpret them unambiguously. Although this method facilitated the integration of annotated bond graph models, bottlenecks may arise when a modification in the CellML bond graph modules is needed (modellers must know the bond graph conservation laws). Moreover, it required adding auxiliary variables as ports to each module and connecting them manually using the semi-automated SemGen merger tool. While annotations are readily incorporated into bond graphs, using annotations in model composition has not been conducted in this context.

Here, as an extension to our previous work, we have incorporated annotations to bond graphs in a new platform. This platform allows us to automatically construct a composed model from annotated CellML files treated as modules. Because the CellML files do not contain the bond graph structure, a separate bond graph library in Python (BondGraphTools [25]) automatically deals with any required changes in the conservation laws. The annotated parameters from the CellML models are then extracted and their values assigned to their equivalent bond graph parameters. Since the equations of a bond graph can be automatically generated from their network structure, we only need to parameterise them using the parameter values from the CellML files. Thereafter, any common biochemical, biological, or physical entities among the modules are identified and merged to render a composed model. We demonstrate this by an example where a bond graph model is constructed from its constitutive modules, *i.e.*, the Epidermal Growth Factor Receptor (EGFR) signalling pathway, Ras activation, and the Mitogen-Activated Protein Kinase (MAPK) cascade are merged to construct a model of the entire EGFR-Ras-MAPK signalling pathway. This type of model integration provides a reliable and consistent framework that is consistent with energy conservation; secondly

ensures that reactions can only operate in the direction of decreasing chemical potential; and third, automates the model composition and merging using the modules' rich semantics.

In this paper, we review the use of bond graphs in modelling biochemical reactions (Section 2.1) and describe how the bond graph modules of the EGFR-Ras-MAPK signalling pathway are constructed based on the existing work by Kholodenko et al. [30] on the EGFR model, Brightman & Fell [31] on the Ras model, and Pan et al. [32] on the MAPK model (Section 2.2). We introduce our automated model composition approach in Section 2.3. Its prerequisites along with the generic method description are reviewed in Sections 2.3.1 & 2.3.2. As an example of our method application, we utilise it to create the EGFR-Ras-MAPK signalling pathway in Section 2.4. Next, we describe how we verified the simulation results of our composed model in Section 2.5. In Section 3 we demonstrate the simulation results both for the constitutive modules and our composed bond graph model, and in Section 4 we discuss and analyse the behaviour of bond graph modules and then verify the behaviour of our composed model. Possible improvements and shortcomings are also discussed in this section. The main features of our method and the future developments are summarised in Section 5.

## 2 Materials and methods

In this section, we give a brief introduction to bond graph modelling of biochemical networks consisting of multiple reactions. Later, we demonstrate how a mathematical model of a biochemical network can be converted into bond graphs. We utilised this approach to create a bond graph model of the EGFR pathway and the Ras activation pathway. Next, we discuss our energy-conserving semantics-based model composition approach: the prerequisites and the approach. This is a generic model composition approach since the idea is domain-independent and could in principle be applied to models in different physical domains (e.g. electrical or mechanical). We will show that by having the bond graph model of any physical or chemical system, our method can merge the identically annotated entities within the models by automatically rewiring the connections between components and modules. The whole composed model in bond graphs will then be ready for simulation or connection to other modules. Bond graphs allow more than two levels of hierarchy which supports model integration. We demonstrate this by applying our method to automatically compose and generate a EGFR-Ras-MAPK signalling model.

### 2.1 Bond graph modelling of biochemical reactions

This section delineates how biochemical reactions are represented and composed in bond graphs.

To facilitate reusability, biochemical models must obey the laws of physics and thermodynamics [33]. In the context of biochemistry, this means that the storage of chemical energy within chemical species must be characterised by physical laws (here, nonlinear capacitive constitutive relation), and reactions are bond to advance in the direction of decreasing chemical potential. In conventional modelling approaches, modellers often ignore the energy transfer; thus, the reactions may proceed against chemical potential gradients and lead to physically implausible models [12]. Since bond graphs are based on energy conservation and thermodynamic laws, fluxes are always in the direction of decreasing potential. Biochemical bond graph models contain components for the species ($C_e$), stoichiometry ($TF$: $N$), and reaction ($Re$). To highlight the notion of the bond graph junctions for sharing a *common molar flux* or a *common chemical potential*, we indicate them by '1: $v$' and '0: $u$', respectively. Here,

- The chemical potential is $u$ (J mol$^{-1}$), stored within the biochemical species, and the molar flux is $v$ (mol s$^{-1}$), driven by the reactions;

- The biochemical species are defined using the component $C_e$, given by the constitutive relation $u = RT \ln(K_q q)$ (**Boltzmann's** formula), where $R$ (J mol$^{-1}$ K$^{-1}$) is the ideal gas constant, $T$ (K) is the absolute temperature, $q$ (mol m$^{-3}$) is the molar concentration of the species, and $K_q$ (mol$^{-1}$) is the species thermodynamic constant [34]. $K_q$ is related to the kinetic free energy of species to participate in reactions and is defined as $K_q = \frac{1}{V_c} \frac{1}{q_{\text{ref}}} e^{\frac{u_q^{\text{ref}}}{RT}}$ where $V_c$ is the volume of the compartment, $q_{\text{ref}}$ is the reference concentration (normally 1 mol), and $u_q^{\text{ref}}$ is the standard free energy formation of the species [25];

- Following the definition in [35], species with fixed concentrations are called chemostats ($C_S$) in bond graph terminology. Such species have a constant chemical potential [20];

- In bond graphs, a reaction represents a dissipative process where chemical energy is lost in the form of heat [36]. In the case of reversible mass action kinetics, a reaction is defined in bond graphs by an *Re* component with the constitutive relation $v = \kappa(e^{u_r}/(RT) - e^{u_p}/(RT))$ (**Marcelin–de Donder** equation), where $\kappa$ is the reaction rate constant and $u_r$ and $u_p$ are total chemical potentials of the reactants and products, respectively;

- Stoichiometries are represented by transformer *TF*: $N$, in which the transformer ratio ($N$) corresponds to stoichiometry.

For further discussion of bond graph modelling of biomolecular and chemical systems, the reader is referred to the works by Gawthrop & Crampin [32, 37].

As an example, a reaction with two reactants and two products is demonstrated in Fig 1 along with its equivalent bond graph representation.

As shown in Fig 1B, the species complexes (*A* & *B* as reactants and *C* & *D* as products) at either side of the reaction are connected to the *Re* component through '1: $v$' junctions because the pairs share common flows. The corresponding '0: $u$' junction for a species can be directly connected to an *Re* component if it is the only reactant or product of that reaction. Here, the reaction flow rate for the *Re* component in Fig 1B is given by:

$$v = \kappa\left(e^{(\alpha u_A + u_B)/RT} - e^{(\beta u_C + u_D)/RT}\right) \tag{1}$$

or if we substitute the chemical potentials with the Boltzmann's formula:

$$v = \kappa\left(K_A^\alpha \; q_A^\alpha \; .K_B q_B - K_C^\beta \; q_C^\beta \; .K_D q_D\right) \tag{2}$$

which can be generally described by mass action kinetics:

$$v = \kappa \left[ \prod_i (K_{r_i} \; q_{r_i})^{\alpha_i} - \prod_j (K_{p_j} \; q_{p_j})^{\beta_j} \right] \tag{3}$$

where $K_{r_i}$ and $K_{p_j}$ are the thermodynamic constants, $q_{r_i}$ and $q_{p_j}$ are the concentrations, and $\alpha_i$ and $\beta_j$ are the stoichiometries of reactants and products, respectively. Reversible Michaelis-Menten kinetics can also be represented using bond graphs [20]. However, because the default *Re* components in BondGraphTools follow the mass actions kinetics, we have chosen to approximate Michaelis-Menten kinetics using elementary mass action reactions (see [12]).

Fig 2 illustrates an example of composing together two reactions in bond graphs. Our framework recognises that the $C_e$: $C$ component is the same in both reactions and merges them. When two components from two modules are merged, the conservation equations at

A

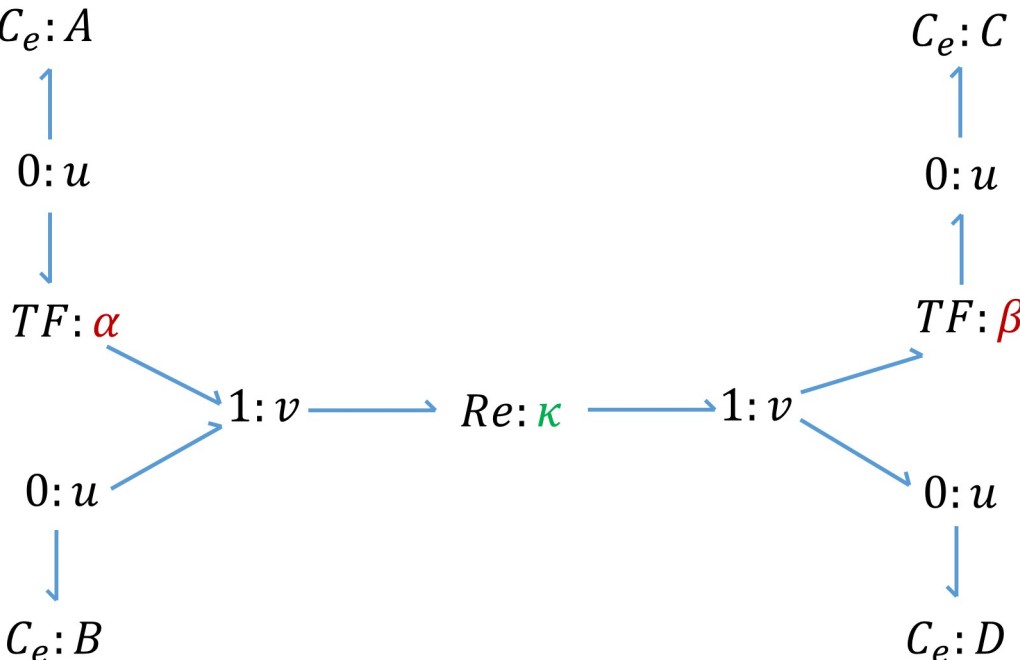

B

**Fig 1. A chemical reaction and its bond graph equivalent.** A chemical reaction with two reactants and two products. (A) Schematic of a chemical reaction where $\kappa$ is the reaction rate constant, $A$ & $B$ are the reactants, $C$ & $D$ are the products, and $\alpha$ & $\beta$ are stoichiometries; (B) Bond graph equivalent of the reaction where $C_e$ components correspond to the species, $Re$ corresponds to the reaction, and $TF$ components represent the stoichiometries. Since the consumption/production rate of all the contributing species in a reaction is equal to the reaction flow rate, they share a common flow with the $Re$ component through a '1: $v$' junction.

their corresponding '0: $u$' junction changes. C Section in S1 Text details the conservation laws and constitutive equations in each reaction separately as well as in the case where both reactions are combined to create the composition.

In the next section we illustrate how the bond graph approach toward biochemical reactions is utilised to create models of three exemplar biochemical pathways.

## 2.2 Modules for EGFR-Ras-MAPK signalling: Bond graph models of the pathways

In this section, we describe the required bond graph modules to compose the EGFR-Ras-MAPK model. Here, we summarise the applied methods to convert the existing mathematical models of the modules into bond graphs.

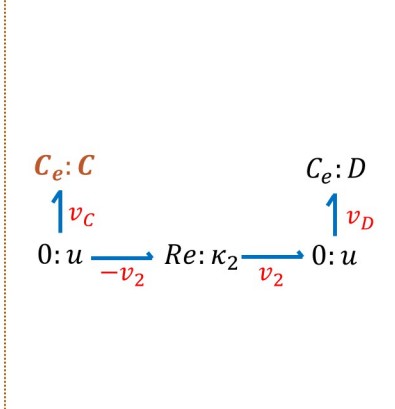

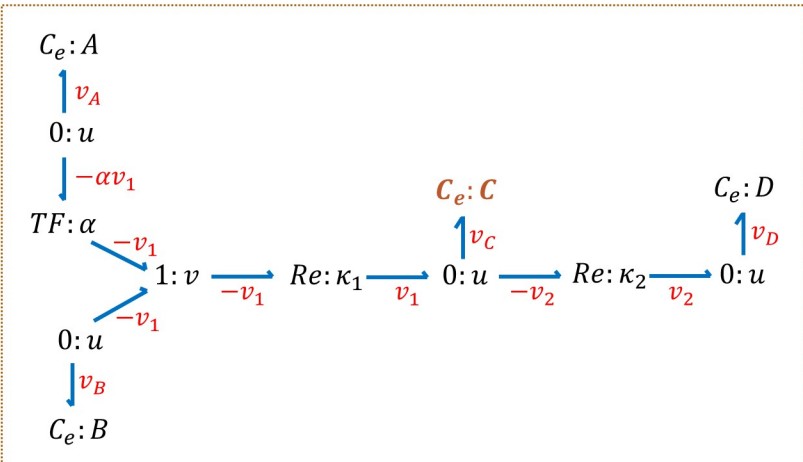

**Fig 2. An example of composing two reactions in bond graphs.** Reactions 1 and 2 represent two separate reactions in which the species C is common. To compose the reactions, the common species (C) is merged and the conservation equation at its corresponding '0: $u$' junction alters to account for the imposed changes in structure. The conservation equation at the '0: $u$' junction connected to the species C is $v_c = v_1$ in Reaction 1 and in Reaction 2 is $v_C = -v_2$ and in the composed reaction it changes to $v_C = v_1 - v_2$.

The EGFR-Ras-MAPK is a signalling pathway that transduces signals from the extracellular environment to the cell nucleus [38]. It participates in multiple biological functions in mammalian cells, including growth and differentiation, cell migration, and wound healing [30, 31, 38] and consists of three major parts: the EGFR pathway, the Ras activation pathway, and MAPK cascade. The EGFR pathway forms a complex (RShGS) which mediates the Ras activation through an intermediate pathway. Ras protein activation signals stimulate the downstream Ras-MAPK cascade [30]. As such, we consider the RShGS complex as a mutual species in the EGFR pathway and the Ras activation pathway. Also, we take Ras protein to be the mutual species in the Ras activation pathway and the MAPK cascade. The bond graph models of the EGFR pathway and the Ras activation pathway are created based on the models by Kholodenko et al. [30] (CellML model available from: *Kholodenko 1999*) and Brightman & Fell [31] (CellML model available from: *Brightman 2000*). The bond graph model of the MAPK

cascade is adopted from the work by Pan et al. [32]. Here, we detail how the bond graph modules of these systems were constructed.

**2.2.1 The EGFR pathway module.** This section describes the creation of the EGFR bond graph module in terms of equations and structure. We inferred the bond graph specific parameters from the kinetic parameters of each model and a set of simulation data. We derived the bond graph structure of the models from their original developed network. We also added some missing biological entities (where applicable).

Fig 3 shows the schematic of the EGFR pathway model developed by Kholodenko et al. [30]. The signal transmission starts with the Epidermal Growth Factor (EGF) binding to the Epidermal Growth Factor receptor (EGFR). It continues via several subpathways that target the SOS (Son of Sevenless) protein. The formation of SOS complex (RShGS) activates the Ras protein through the Ras activation pathway, which initialises phosphorylation in the MAPK cascade.

Fig 4 illustrates the bond graph equivalent network of the EGFR pathway.

The reactions in the EGFR model by Kholodenko et al. are either reversible or irreversible. The reversible reactions are described using the kinetic scheme as:

$$v = k^+ \prod_i q_{r_i} - k^- \prod_j q_{p_j} \tag{4}$$

where $k^+$ and $k^-$ are the forward and reverse kinetic rate constants and $\prod_i q_r$ and $\prod_j q_p$ are the concentrations of reactants and products, respectively. The kinetic model and parameters are given in [30] (Table I & Table II). The irreversible reactions (steps 4, 8, 16) are described using the irreversible **Michaelis–Menten** kinetics as:

$$v = V_{\max} \frac{q_r}{K_m + q_r} \tag{5}$$

where $V_{\max}$ (mol/$s$) is the maximum reaction rate achieved by the system, $K_m$ (mol) is the Michaelis constant referring to the reactant concentration at half of the $V_{\max}$, and $q_r$ (mol) is the reactant concentration. Irreversible reactions are thermodynamically impossible in a bond graph model; we deal with this issue later.

The required energy for the reactions is supplied by Adenosine triphosphate (ATP) hydrolysis, producing Adenosine diphosphate (ADP) during phosphorylation and phosphate (Pi) during dephosphorylation. The reversible phosphorylation reactions (steps 3, 6, 14) follow the kinetic formulation (Eq 4) and the irreversible dephosphorylation reactions (steps 4, 8, 16) follow the Michaelis–Menten kinetics (Eq 5). Kholodenko et al. have not explicitly included ATP, ADP, and Pi in their model, which contravenes mass and energy conservation. Therefore, we considered these species in our bond graph model. ATP, ADP, and Pi are assumed to be chemostats.

To convert the kinetic parameters ($k^+$ and $k^-$ in Eq 4) to those required by bond graphs ($\kappa$, $K_r$, and $K_p$ in Eq 3), we first removed the thermodynamically infeasible irreversible reactions from the network (for their different parameter definitions). Then, we applied the optimisation method described in [33] to the remaining reversible reactions. In brief, by taking logarithms on the constraints of each reaction ($k^+ = \kappa \prod_i K_{r_i}$ and $k^- = \kappa \prod_j K_{p_j}$), the relationship between the kinetic and bond graph parameters can be expressed as a linear matrix. The reader is referred to an example on the generation of the linear matrix of thermodynamic constants in Appendix B for [33]. Due to accounting for the ATP, ADP, and Pi components in our bond graph model, we included them in the constraints of their corresponding reactions.

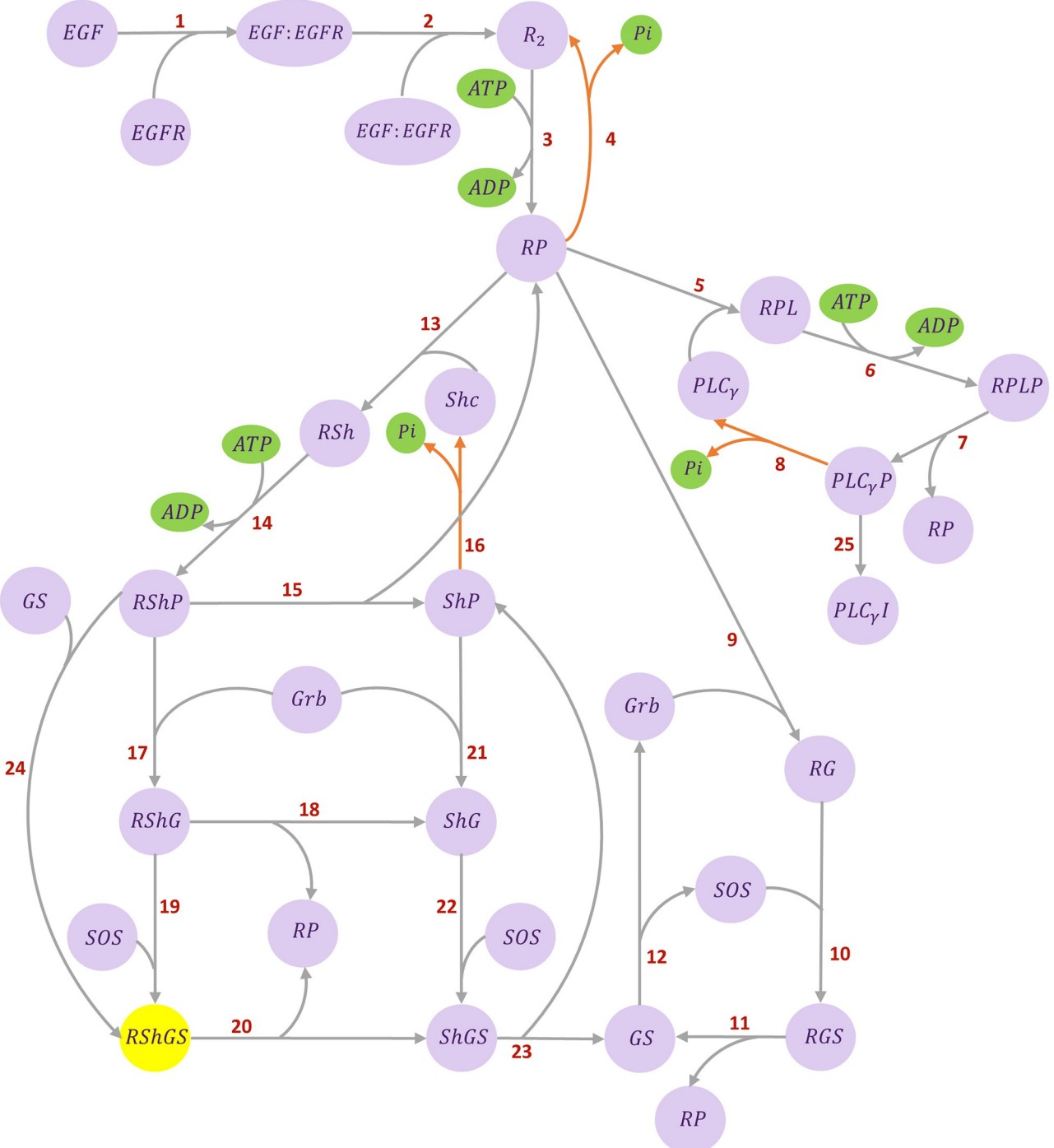

**Fig 3. Kinetic structure of the EGFR pathway.** The ATP hydrolysis species are shown in green (involved in phosphorylation and dephosphorylation reactions). The RShGS complex in yellow is mutual between the EGFR pathway and the Ras activation models. The reactions are numbered as the equations in the CellML source code. Steps 4, 8, 16 in orange represent the irreversible reactions. The network was adapted from [30].

As our selected pathways are in the cytosolic compartment, they use the same sources of potential (ATP, ADP, and Pi); thus, we used the same chemical potentials for these chemostats as we used in the MAPK cascade module. We obtained the thermodynamic constants in the previous phase (except the ones in the irreversible reactions), in which we converted the

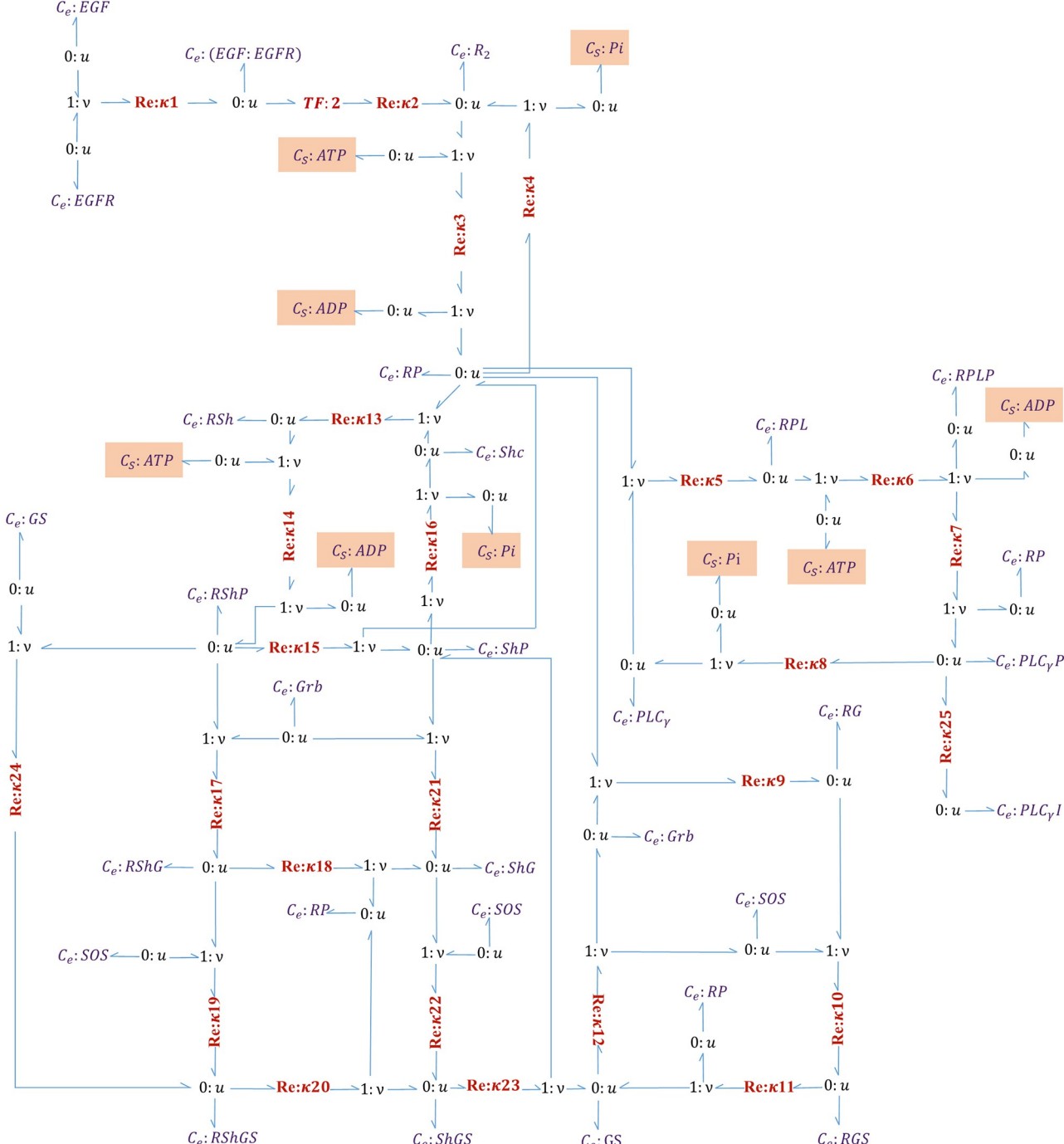

**Fig 4. Bond graph representation of the EGFR pathway model.** *Re* components are numbered according to the steps in [30]. Each $C_e$ or $C_S$ component is connected to a '$0 : u$' junction. Where a species participates in more than one reaction, new bonds are applied to its corresponding '$0 : u$' junction to share a common chemical potential (See R-PL where it is produced in reaction 5 and consumed in reaction 6). The chemostats in orange boxes are added to the reconstructed bond graph version.

kinetic parameters into bond graph parameters. We approximated the irreversible reactions with kinetic quantities (Eq 4) which led to a negligible reverse molar flux. We obtained the time-dependent behaviour of the contributing species in steps 4, 8, and 16 from the reference CellML model for the EGFR pathway and applied curve fitting to estimate the reaction rate

constants for the irreversible steps ($\kappa_4$, $\kappa_8$, & $\kappa_{16}$). As an example, this procedure is shown in B Section in S1 Text for step 4. As we will discuss further in Section 3, an exact fit was not possible because of two reasons: first there are irreversible reactions in the original model, and second, some of the reversible reactions in the original model do not satisfy detailed balance. Since the bond graph parameters are inferred from the original model, an approximation with the least square error is made to generate the closest fit to the data while adhering to detailed balance constraints. The reaction equations along with their participating species are given in S1–S3 Tables compare the parameter amounts of the Kholodenko et al. model with the ones from our reconstructed bond graph model. The code to convert the kinetic parameters into bond graph equivalents for the EGFR pathway is accessible from: https://github.com/Niloofar-Sh/EGFR_MAPK/tree/main/EGF.

**2.2.2 The Ras activation module.** This section describes the reactions, species, and structure of the Ras activation module. We modified an existing CellML model to achieve our desired Ras activation model.

In the EGFR pathway model by Kholodenko et al., RShGS and RGS are considered to trigger the activation of Ras protein through intermediate molecules that transmit the signal to the downstream MAPK cascade and yield Ras. Since none of our initial reference models (the EGFR pathway and the MAPK cascade) included these molecules, we incorporated an intermediate module to account for the missing steps. We created this module by modifying an EGFR-Ras-MAPK pathway model developed by Brightman & Fell [31]. We kept the steps starting from RShGS (to be merged with RShGS in the EGFR pathway model) to Ras (to be merged with Ras is the MAPK cascade model) and removed all other steps. Since RGS was not included in the activation of Ras in the Brightman & Fell model, we did not include it in the Ras activation module. It is worth mentioning that RShGS has a prominent role in localising Ras compared to RGS [39, 40]. Fig 5 represents the kinetic and bond graph structure of the Ras activation module which links the EGFR and MAPK modules.

We converted the kinetic parameters of the reactions into bond graph parameters using the same applied techniques in Section 2.2.1. For the irreversible reactions (steps 2 and 4), we assumed a very small value for the reverse kinetic constants ($k^-$) to limit the reverse flow to a negligible amount. The reaction equations along with their participating species are given in S4–S6 Tables compare the parameter amounts of the Brightman & Fell model with the ones from our reconstructed bond graph model. The code to convert the kinetic parameters into bond graph equivalents for the Ras activation module is accessible from: https://github.com/Niloofar-Sh/EGFR_MAPK/tree/main/Ras.

**2.2.3 The MAPK cascade module.** Here, we discuss the sub-modules of the MAPK cascade model and how a single symbolic bond graph module can characterise the whole cascade.

Fig 6 shows the schematic of the MAPK cascade. Each oval trajectory in Fig 6A represents a cycle. The stimulus signal is amplified sequentially through the cycles in the cascade. MKKK is activated through a single phosphorylation phase by a kinase (Ras) and turns into MKKKP [41]. MKK and MK each phosphorylates in two steps and ultimately produce MKKPP and MKPP. The phosphorylated product of each layer plays a kinase role for the phosphorylation phase in the next downstream layer. Simultaneously, an opposing phosphatase dephosphorylates the product of each cycle (shown by backward arrows) [42]. Each layer in Fig 6A is dephosphorylated by a specific phosphatase: MKKK-Pase in the first layer, MKK-Pase in the second layer, and MK-Pase in the third layer. The dual phosphorylation-dephosphorylation mechanisms in the second and third layers act as amplification, generating ultrasensitive responses.

Although the species in each cycle are different, the structures of the cycles are the same. The similarity in the structures enables us to break the cascade down into five modules of cycles. Hence, we created a symbolic bond graph module for a single cycle and reused this template for

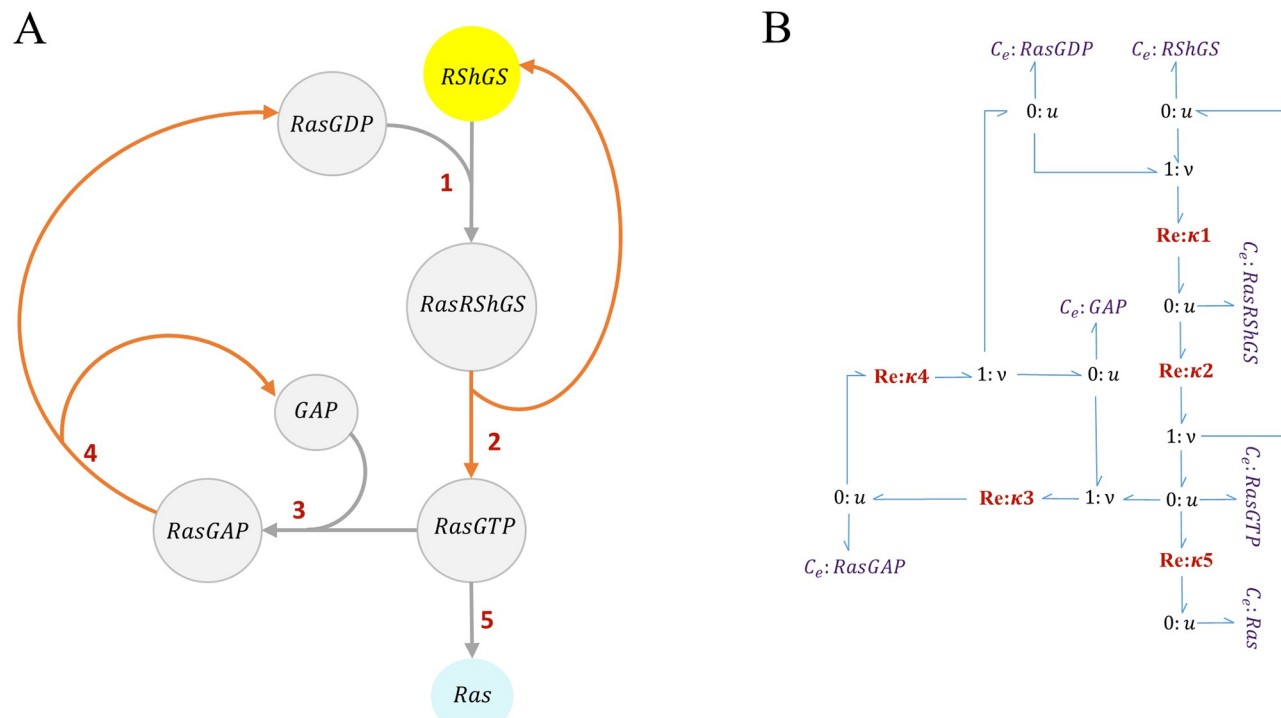

**Fig 5. The structure of the Ras activation module.** (A) Kinetic representation. The RShGS complex in yellow is mutual between the EGFR pathway and the Ras activation modules and Ras protein in blue is mutual between the Ras activation module and the MAPK cascade module. Steps 2, 4 in orange represent the irreversible reactions. The network was adapted from [31]; (B) The bond graph representation of the Ras activation module.

the other four cycles. Fig 6B shows the modular representation of the cascade by reusing the bond graph module in Fig 6C. We used a semantics-based 'white box' approach rather than the 'black box' approach in Pan et al.'s work. We also annotated each cycle separately to automate the model composition. We modelled the MAPK cascade in the absence of feedback. The code for the modular bond graph model of the MAPK cascade in BondGraphTools is accessible from: https://github.com/Niloofar-Sh/EGFR_MAPK/tree/main/MAPK%20cascade.

In the literature, several models of the EGFR-Ras-MAPK signalling pathway have been developed considering the involvement of MKKP as an enzyme in phosphorylation of MK and MKP [31] as well as negative feedback from MKPP to the upstream EGFR pathway [41–45]. This paper aims to demonstrate the reusability and composition of bond graph modules; thus, further involvements and feedback loops are not considered in our composition procedure. However, to verify the behaviour of the final composed model, we studied the response of our bond graph EGFR-Ras-MAPK model under the condition of adding negative feedback from MKPP to incorporate as an inactivating enzyme in the first cycle as studied by Kholodenko et al. [43].

The following section describes the pipeline of our automated model composition framework, which will later be applied to integrate the EGFR pathway, the Ras activation, and the MAPK cascade modules.

## 2.3 Automated model composition pipeline

In this section, we explain our automated model composition pipeline which is mainly based on the application of semantics and bond graphs. We mention the prerequisites to apply our framework in model composition and the structure of our developed framework.

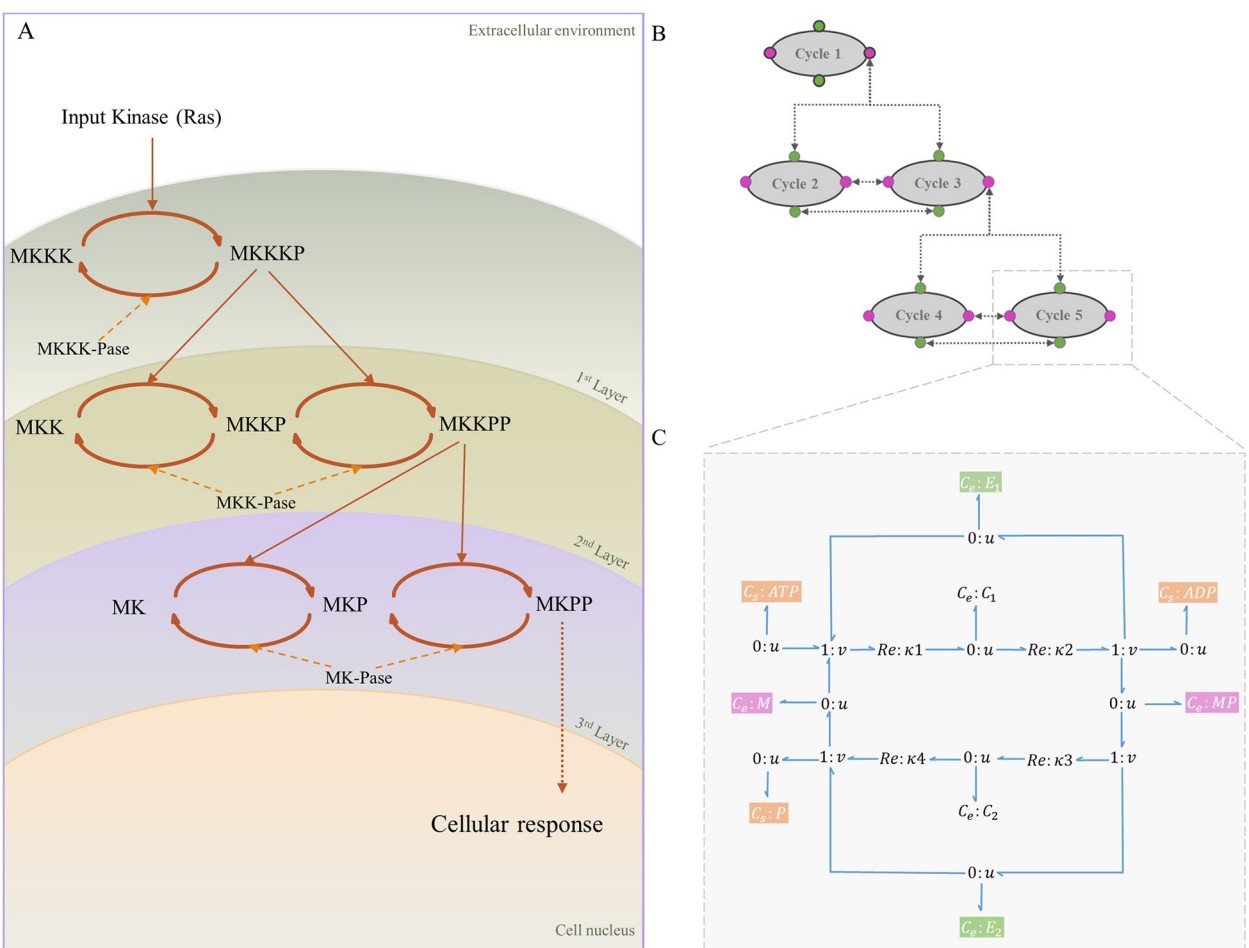

**Fig 6. The structure of the MAPK cascade.** (A) Kinetic representation. The stimulus from the extracellular environment is received (Ras) and transmitted through the MAPK cascade to the cell nucleus. The layers demonstrate the cycles with the same kinase and phosphatase enzymes; (B) MAPK cascade with five modules. Linking species are shown in colours where green corresponds to the linking enzymes and pink corresponds to unphosphorylated/phosphorylated mitogen proteins. Arrows show the links between the modules; (C) The symbolic bond graph model of each cycle. Sources of potential with fixed concentrations ($C_s$:ATP, $C_s$:ADP, and $C_s$:Pi) are shown in orange. (Same sources of potential within the modules are omitted in (A) and (B) for clarity).

We aim to minimise manual input through automation in model composition while using energy-based modules in an open-source environment. In this endeavour, we have provided some exemplar predefined bond graph models in which the parameters do not have any values. We call these predefined bond graph models as symbolic modules. Symbolic modules allow us to determine the parameters' values later where the annotated parameters from the CellML models would accordingly link. The suitable bond graph template is then automatically selected from a list of symbolic bond graph templates by identifying specific annotated parameters (for example the species specific constants) in the CellML models. Here, we have stored symbolic bond graph templates for the EGFR pathway, Ras activation, and the MAPK cascade cycles (Sections 2.2.1, 2.2.2 & 2.2.3). In the current work, when a symbolic bond graph template is parameterised we call it a bond graph module. Merging points are automatically recognised and merged, resulting in a physically consistent model. Due to the hierarchical feature of bond graphs, the needed adjustments during the composition will be systematic, leading to the automation of modifications (adding/deleting bonds between the components).

To apply our method to CellML models, some preparations are required in advance, *i.e.*, installing the bond graph Python library as well as downloading the required ontologies (required in the current framework but optional in the general approach). Once prepared, the user can commence the model composition in a Python environment like Jupyter Notebook.

**2.3.1 The prerequisites.**   Below, we introduce the computational prerequisites in our framework.

- BondGraphTools
The task of automated model composition requires bond graph software that readily supports automation. For this purpose, we have selected BondGraphTools—an open-source Python library for bond graph modelling—created and developed by Cudmore et al. [25], accessible from https://github.com/BondGraphTools/BondGraphTools. BondGraphTools supports modularisation and automation in model building.

- Ontologies (optional)
An ontology is a semantic resource of standard notions and vocabularies of species, structures, and observations in terms of Resource Description Framework (RDF) triples (https://www.w3.org/RDF/). RDF is a standard mechanism to describe and interchange data on the Web and an RDF triple is a subject–predicate–object statement that describes the properties of an entity, often using ontologies [46, 47]. For example, the RDF [OPB00340—CHEBI29103—FMA70022] reads [concentration-potassium-extracellular space] which specifically describes the physical property and location of an entity. Ontologies are useful tools to add meaning to different parts of models to avoid any ambiguous interpretations [48]. Depending on the area of biomedical science in which the researchers annotate their models, one or various reference ontologies might be used. For the scope of this publication, we used the *csv* files for the Ontology of Physics for Biology (OPB) [49] and the Gene Ontology (GO) [50], downloaded from the following links:

  - OPB: https://bioportal.bioontology.org/ontologies/OPB;

  - GO: https://bioportal.bioontology.org/ontologies/GO.

The OPB is a reference ontology for physical principles such as chemical concentration, electrical capacitance, temperature, and fluid volume. The GO provides descriptions for molecular biology such as gene products, biological sequences, and molecular activities. Due to the limited size of uploaded files on GitHub, the required reference ontologies for the current model composition (OPB and GO) are not provided on our GitHub repository. We stored the ontologies locally to interpret the RDFs and use the interpretations where the user needs to make a decision based on the annotations but the approach can be reduced to a framework in which the annotations are only read and compared in RDF format and the interpretations are not given to the user.

**2.3.2 The generic approach.**   In this section, we describe the steps in our model composition framework and the workflow towards it.

Automatic composition of bond graph modules by having bond graph templates and annotated reference models can be performed in any domain. To reuse and compose models deposited on online repositories such as PMR, we need a tool to first convert a non-bond-graph model into an equivalent bond graph one; second, automatically assign the parameters in the models to their equivalent bond graph components; third, identify the same entities in the models as the merging points and make the necessary changes to join the models without any loss of information.

To improve our model composition method toward automation and reuse the models in various formats, we built symbolic bond graph templates and connectivity matrices for some exemplar systems (the EGFR pathway, Ras activation, and the MAPK cascade). A connectivity matrix is a binary square matrix that defines connections between the elements of a system. Here, the number of rows and columns each equals the number of bond graph elements of a system [51]. Connectivity matrices are symmetric for undirected (bidirectional) networks and asymmetric for directed networks. Instead of using the embedded syntax in BondGraphTools to append/delete the bond graph elements, we used connectivity matrices. While not essential to our methodology, we chose this approach because binary representation of models clearly shows the connections and gives the minimal required details to define a network which can be exported to other tools and software for further analysis [51]. To modify a network, one can insert 0 or 1 in the matrix or delete its corresponding row and column. An example in A Section in S1 Text shows how connectivity matrix is defined for a simple network.

To identify the merging points between the modules we used a 'white box' approach. In this approach all or a group of the bond graph components in the modules are mergeable. In a 'black box' composition approach in contrast, only the components predefined as inputs or outputs are accessible [28, 52]. In coupling biological models, all entities are mergeable, hence, we found the 'white box' configuration more compatible with our model composition method. To do this, we need the parameters of the models to be annotated. To summarise, we started our automated model composition method by preparing the bond graph symbolic templates of the models, the connectivity matrix of each bond graph template (optional), and ontologies required for interpreting the annotations (optional).

In a given biological/physiological/physical context, our framework can detect the type of bond graph template that matches the annotated model. This is done by searching for specific groups of biological entities/processes within the annotated CellML files. If a certain group of entities is found in a file, then our framework will link it to its corresponding bond graph symbolic template. Thereafter, a function in our framework finds identical annotations in the models and selects the merging points. Based on this, the required changes in the bond graph components (deleting the duplicates) and the connectivity matrices (deleting or inserting rows and columns) will be made automatically. Ultimately, our framework produces the final model based on the connection/non-connection relationships between all the components (Fig 7).

We have deposited the required ontologies, the bond graph symbolic template models, and their connectivity matrices in our repository. Ontologies and connectivity matrices are required for the current implementation but are removable based on the application and by slight modifications in implementation. Any number of CellML models containing the annotated parameters of a system can be used in our framework as inputs (here, we illustrate two models). Fig 7 depicts the eight main steps in our semantics-based model composition framework as follows:

1. A function in our framework extracts the annotations and values of the CellML models. If exact matches of annotations are not detected between the models, a warning is given. The user should check the models to see if they are appropriate for composition. If there are matching annotations, two pathways are made available: **composition process** and **value allocation**.

2. In this step, a function checks the mergeability of the identically annotated entities. If they are not mergeable, the function ignores the entities [53]. Otherwise, it passes them to the next step. For example, biochemical species are considered mergeable since they can simultaneously participate in multiple reactions but a parameter like temperature cannot be merged as it cannot become a port for external connections. Based on the deleted duplicate

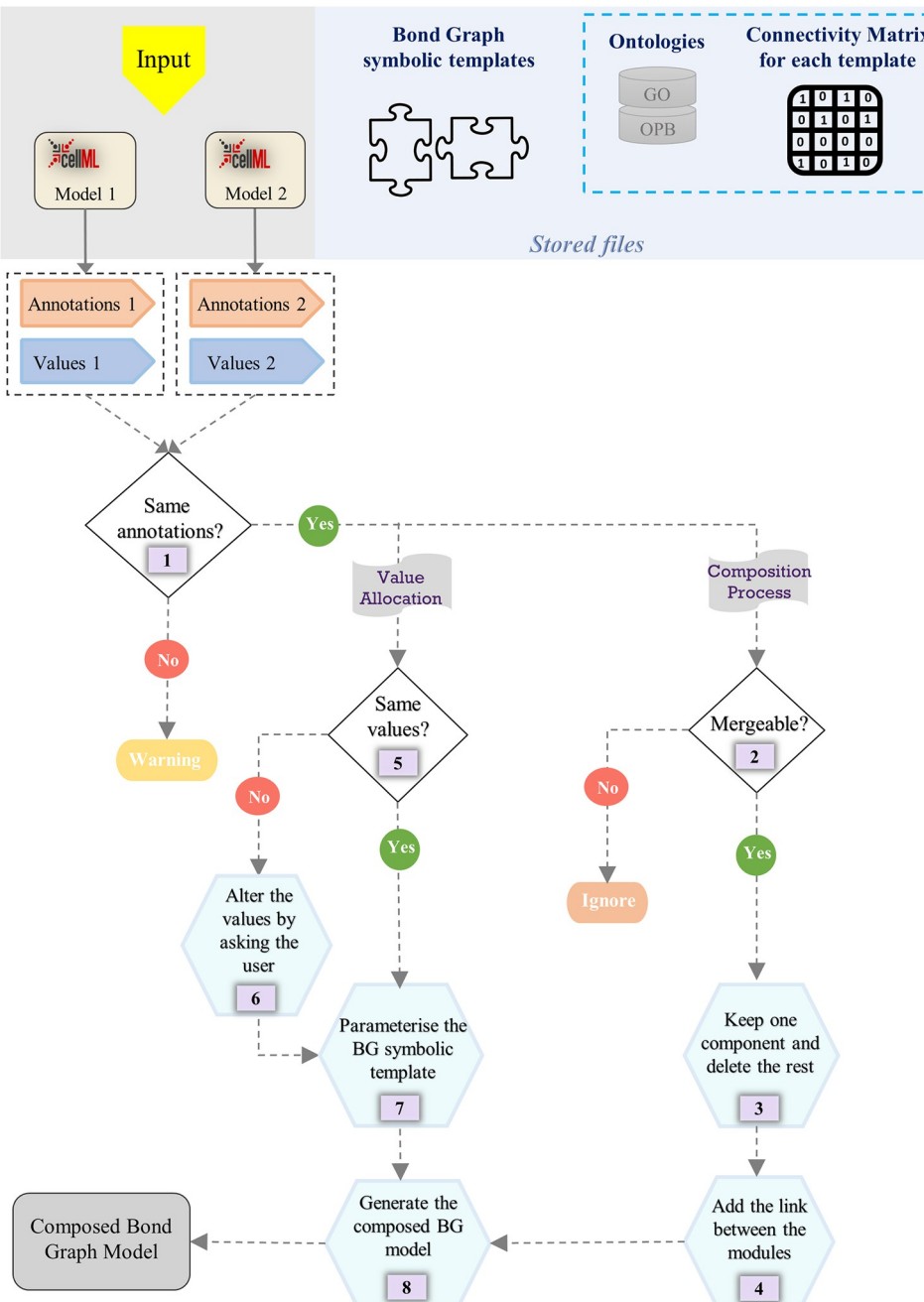

**Fig 7. The generic flowchart of our automated model composition approach.** The *Stored files* section shows the saved files for the current model composition framework. Ontologies and connectivity matrices in the blue dashed box are optional in the generic approach but were used in the current framework. The Input section shows two arbitrary CellML models to be merged using our framework but can be extended to any number of models. The main steps of the framework are denoted by numbers (1–8).

components, the connectivity matrices are combined, allowing the models to be merged (details in Section 2.4).

3. In this step, only one entity is kept from each group of identically annotated mergeable entities and the rest is deleted.

4. In this step, our framework links the modules at each merging point to integrate them. A link is a bond in bond graph terminology and can be added to the system by inserting a 1 in the connectivity matrix (used in the current approach) or adding a syntax to incorporate a new bond between the modules.

5. Step 5 identifies inconsistencies in the values of identically annotated entities. These values include the initial conditions and the entities' thermodynamic constants (as described in Section 2.1).

6. This step prompts the user to choose a value for the identically annotated entities found in step 5. For instance, if a chemical species is present in more than one model (identically annotated in all the models) and has different initial concentrations, the user is asked to select one of the values or insert a new one for that specific chemical species.

7. This step parameterises the bond graph symbolic templates with the values for each annotated entity.

8. Step 8 gathers the information coming from the **composition process** and **value allocation** to generate a bond graph composed model in the form of a system of Ordinary Differential Equations (ODEs).

In the next section, we illustrate the workflow by applying it to an example biochemical network: the EGFR-Ras-MAPK signalling pathway.

## 2.4 Applying the composition method to the EGFR-Ras-MAPK signalling pathway

Here, we implement our model composition method (described in Section 2.3.2) to generate a model of the EGFR-Ras-MAPK signalling pathway. The EGFR-Ras-MAPK signalling pathway is comprised of three modules: the EGFR pathway, the Ras activation pathway, and the MAPK cascade. Since the MAPK cascade includes five structurally repetitive cycles, we broke it down into five sub-modules. As such, we need three template bond graph modules; one for the EGFR pathway (Fig 4), one for the Ras activation pathway (Fig 5B), and one for the cycles in the MAPK cascade (Fig 6C). The connectivity matrix for each module in *csv* format and the annotated CellML files for the parameters of each module/sub-module are available on GitHub: https://github.com/Niloofar-Sh/EGFR_MAPK.

In the composition process, if the identically annotated entities are mergeable, only one bond graph component is kept and our framework removes the rest from the modules (the list of components for each module will be updated). This process works for any number of components to be merged among the models. Furthermore, the rows and columns of the connectivity matrices that correspond to the removed components will be deleted and ultimately, a connectivity matrix describing all the connections between the components of the composed network is needed. Our framework integrates the modified connectivity matrices of all the modules into one by putting the connectivity matrices consecutively in the diagonal direction of a zero square matrix. Thus, the number of rows/columns equals the total number of components in the system. Subsequently, where we need a bond between two modules, our framework inserts an additional 1 in the matrix. Fig 8 demonstrates this with an example.

In the next section, we describe the verification methods taken to evaluate the behaviour of the three bond graph modules and our bond graph composed model of the EGFR-Ras-MAPK pathway.

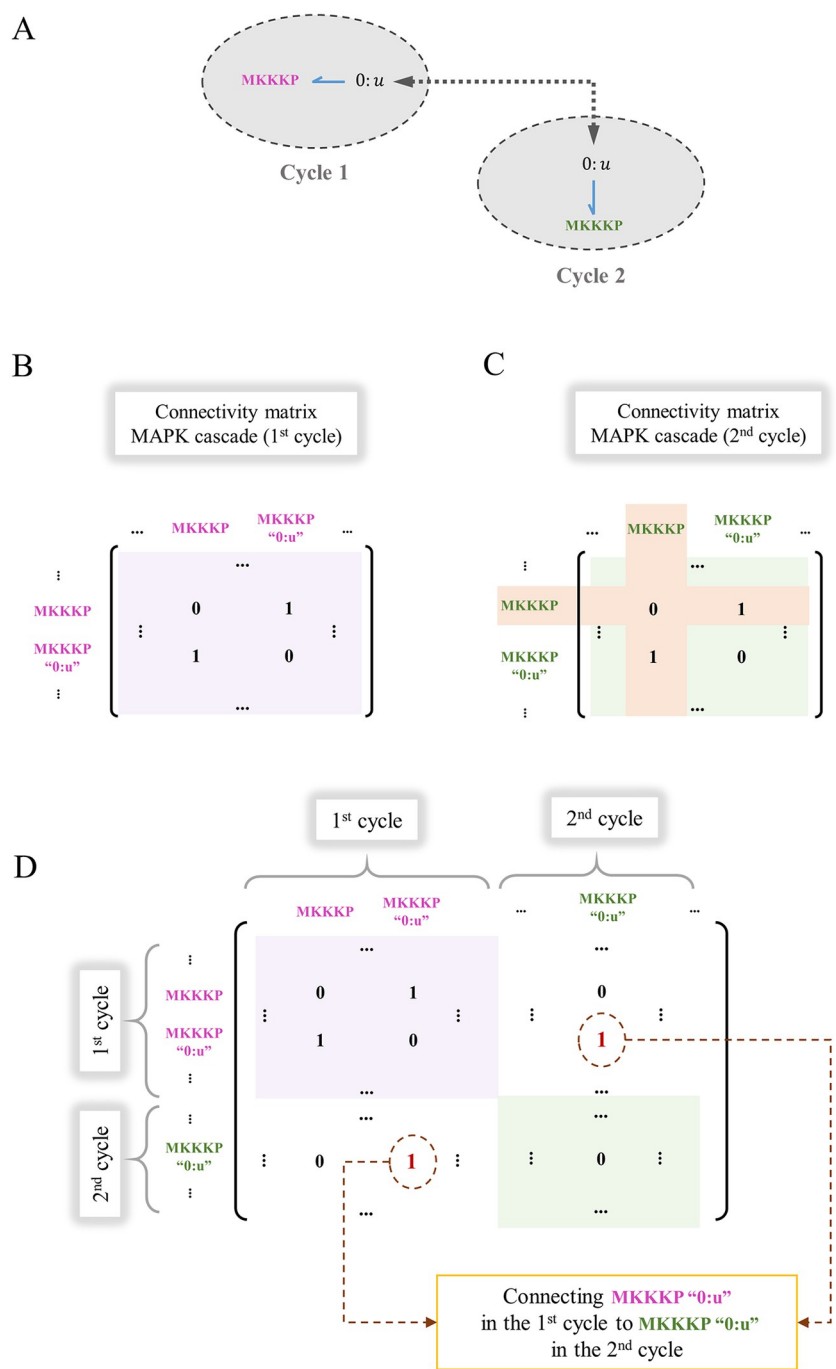

**Fig 8. Construction of the whole-system connectivity matrix for a composed model.** The procedure is illustrated by integrating two connectivity matrices (1st and 2nd cycles in the MAPK cascade). Initially, the two cycles had identical connectivity matrices. (A) MKKKP is a common component between the first and second cycle; (B) The connectivity matrix for the 1st cycle; (C) The modified connectivity matrix for the 2nd cycle where the row and column for the common component (MKKKP) will be removed; (D) The placement of the connectivity matrices for each module on the diagonal of the whole-system connectivity matrix. The pink and green boxes indicate the connectivity matrices for the 1st and 2nd cycles, respectively. The corresponding '0:u' junctions for MKKKP in the two cycles are connected by inserting two 1s (in red) to represent a bond between them (bidirectional connections between the components require the matrix be symmetric).

## 2.5 Verification

To verify the behaviour of the composed EGFR-Ras-MAPK bond graph model, we first compare our bond graph estimation of the EGFR and Ras activation modules to the original models. Bond graph models of biochemical systems are deterministic and generate a set of ODEs which can be solved by any standard ODE solver package. In this paper, the models were simulated using the SUNDIALS package [54]. In future work, there is scope to expand the energy-based approach to model of stochastic systems, using algorithms such as the Gillespie algorithm for simulation.

To compare the simulation results between the models, the normalised root mean square error (NRMSE) was computed as in Eq 6, where $\hat{x}i$ corresponds to the simulation points of our bond graph estimation, and $x_i$ corresponds to the Kholodenko et al. model. The normalisation was performed relative to the difference of maximum and minimum data of the reference model in each simulation.

$$\text{NRMSE} = \frac{\sqrt{\sum_{i=1}^{n} \frac{(\hat{x}_i - x_i)^2}{n}}}{x_{\max} - x_{\min}} \tag{6}$$

Second, we studied the steady-state behaviour of the phosphorylated kinases at the terminal level of each layer of the MAPK cascade under varying stimulus strengths. This denoted how we should expect the kinases to respond to any stimulus coming from the upstream levels (here, the Ras protein).

To further study our composed model, we observed its behaviour under three more conditions: a) We added negative feedback from the terminal phosphorylated kinase in the last layer of the cascade (MKPP) to the dephosphorylation reaction in the initial layer [43] (Fig 9). The effect of adding the negative feedback was then observed and qualitatively verified. b) We simulated the model for different intracellular ATP concentrations and monitored how the concentration of activated kinases was correlated to this change. c) We investigated the behaviour of the composed bond graph model for varying initial concentrations of EGF (the initiating species in the EGFR pathway).

In the following section, we illustrate and explain the results of our verification measures on the composed EGFR-Ras-MAPK bond graph model.

## 3 Results

We used our method to merge the modules within the MAPK cascade and between the pairs {EGFR pathway, Ras activation} and {Ras activation, MAPK}. This yielded the bond graph configuration of the EGFR-Ras-MAPK signalling pathway. Fig 10 shows how the EGFR, Ras activation, and MAPK modules are manipulated to deal with same components existing within the modules. Here, RShGS and Ras in the Ras activation module are removed while RShGS in the EGFR module and Ras in the MAPK module are kept. Also, all ATP-ADP-Pi trio components in the MAPK model are removed while they are kept in the EGFR module. All the reserved mutual components are bonded to the '0: $u$' junctions corresponding to the removed components.

To check the function of our composed model, we verified the simulations in two steps: 1. verification of each bond graph module separately (EGFR, Ras activation, and MAPK); and 2. verification of the bond graph composed model (the EGFR-Ras-MAPK signalling pathway).

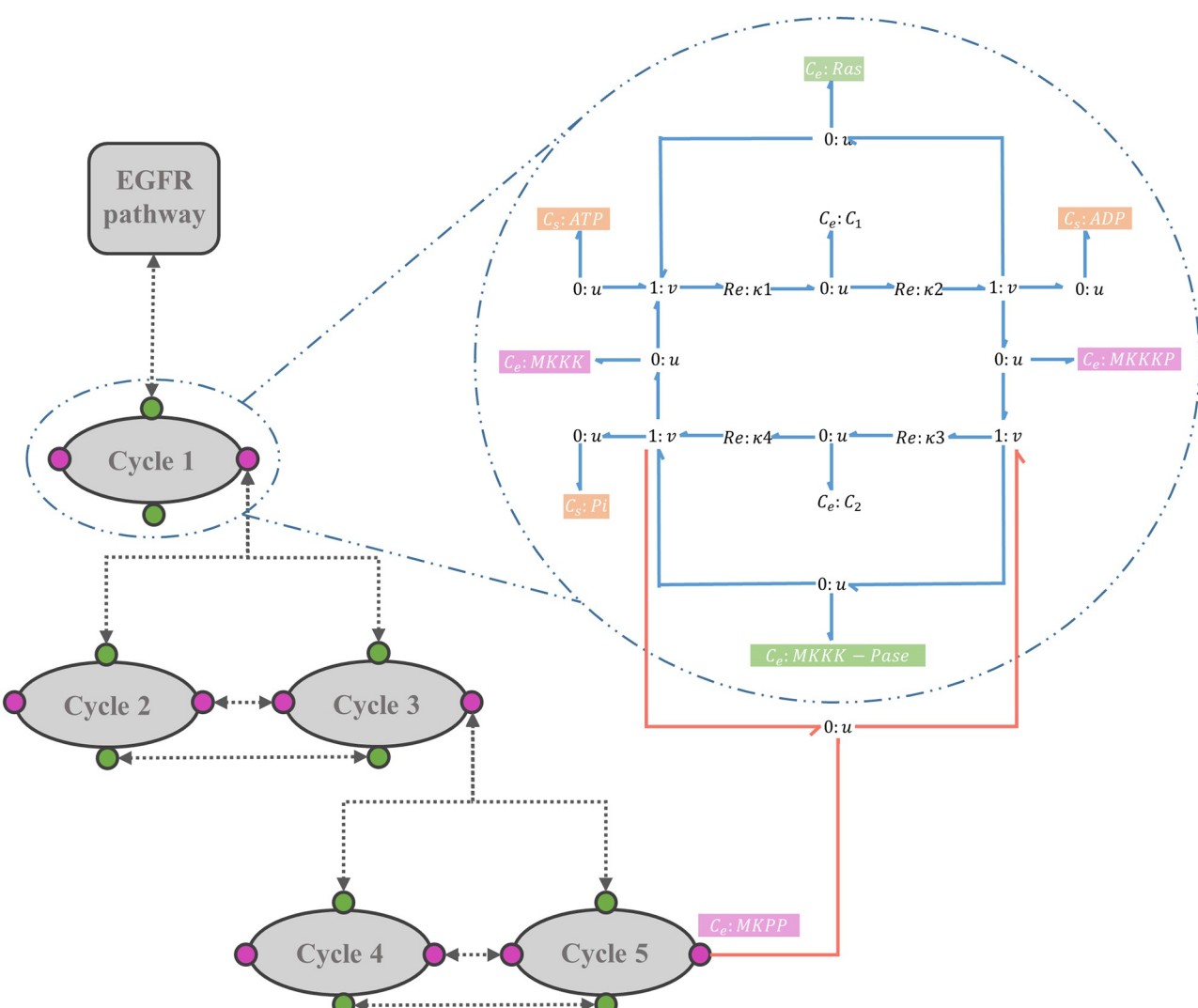

**Fig 9. Bond graph schematic of adding negative feedback in the composed EGFR-Ras-MAPK bond graph model.** The negative feedback loop (red bonds) initiates from MKPP and has an enzymatic role in the first layer's dephosphorylation reaction.

## 3.1 Verification of bond graph modules

In this section, we verify the created bond graph modules (EGFR, Ras activation) against their original non-bond graph models. The functionality of the MAPK cascade bond graph module (consisting of 5 sub-modules) is also studied.

- **The EGFR signalling pathway:** Our approach requires models to be expressed as bond graphs. A bond graph equivalent of the EGFR pathway was not available, which motivated us to convert an existing kinetic model of the EGFR pathway into an equivalent bond graph form. An exact conversion was not possible due to the existence of irreversible reactions and not explicitly accounting for mass conservation. Hence, we approximated the non-bond graph irreversible reactions with bond graph equivalents and included the missing metabolites ATP, ADP, and Pi to provide the energy required to approximate the irreversible reactions.

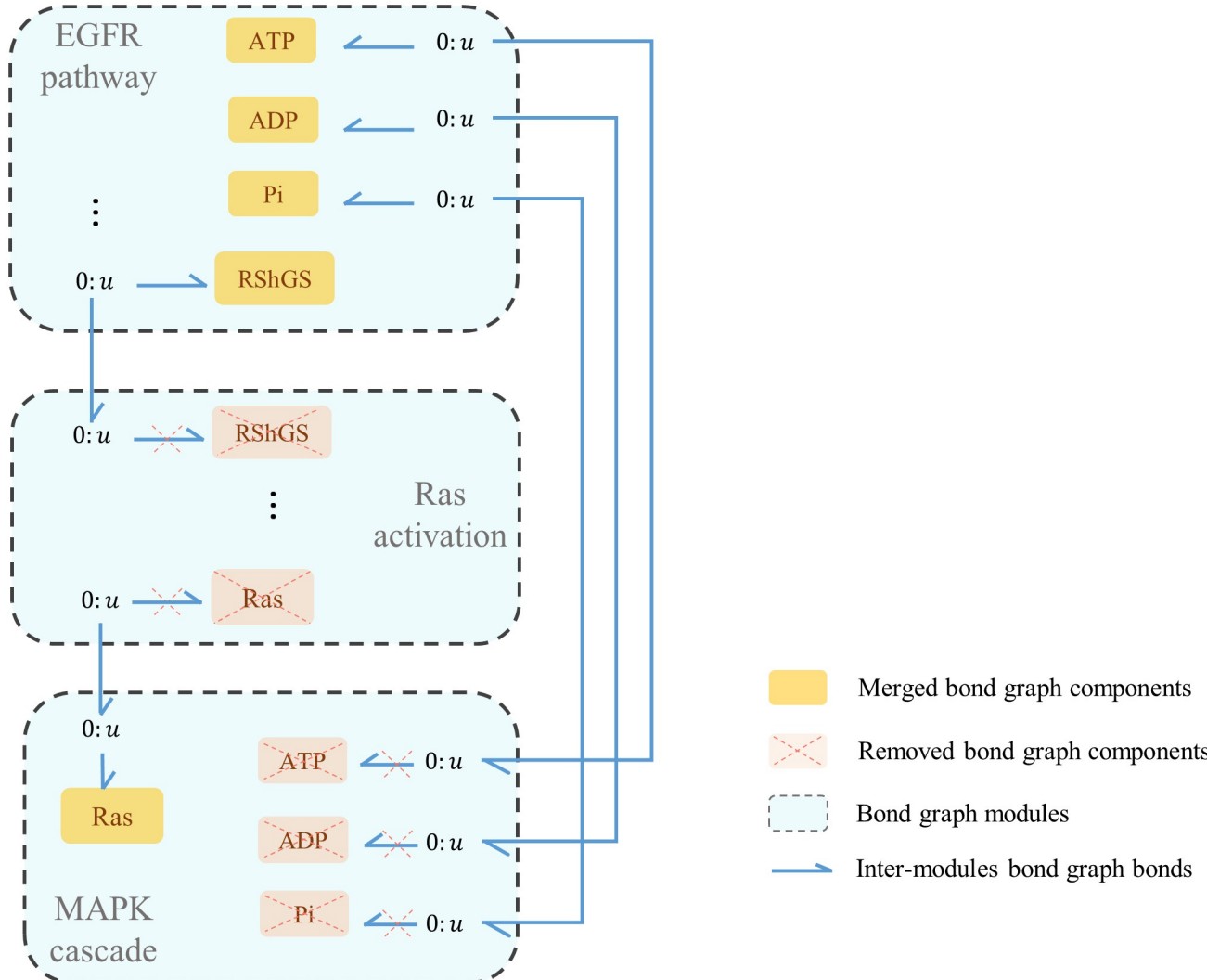

**Fig 10. The composed modular bond graph model of EGFR-Ras-MAPK signalling pathway.** The blue dashed boxes represent the bond graph modules, the yellow boxes show the merged common components between the modules (each sharing a common potential by a '0: $u$' junction), and the blue harpoons represent the bonds between the modules and common components. The inter-module bonds, along with the internal bonds between the components in each module, are defined and automatically applied to the model using the whole connectivity matrix. The EGFR and MAPK cycles also share common potentials with $C_S$:ATP, $C_S$:ADP, and $C_S$:Pi.

The conversion of the kinetic EGFR model into bond graphs was performed by solving a linear matrix of equations for the constraints. The species' responses in the EGFR bond graph module were observed and compared to the ones derived from the Kholodenko et al. model. The responses of four exemplar species in the pathway are demonstrated in Fig 11, and the NRMSE is computed for each comparison in percentage. We see that the bond graph equivalent of the EGFR module functioned similarly to the original kinetic model, although the equations could not be solved perfectly. This implies that the kinetic parameters of the Kholodenko et al. model are not thermodynamically consistent. The bond graph equivalent represents a close-match approximation of the original model in a thermodynamically consistent manner.

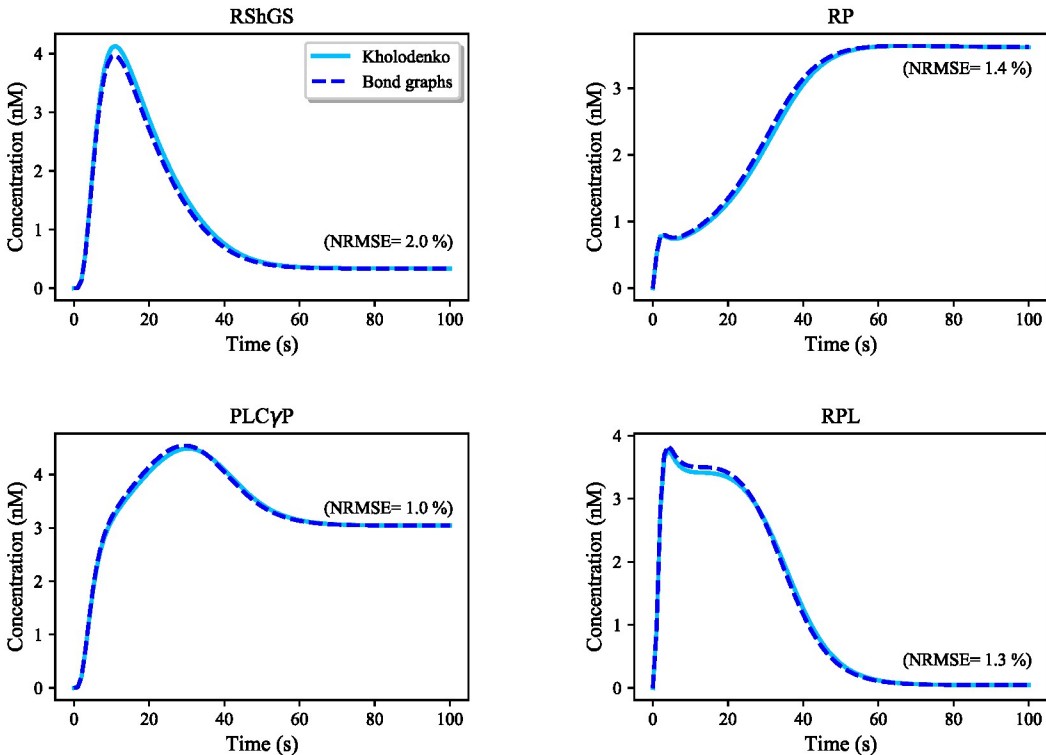

**Fig 11. Comparison between the Kholodenko et al. EGFR model and its bond graph approximation.** The simulations are given for four exemplar species in the pathway. NRMSE is calculated for each comparison in percentage. The initial concentration of EGF (the initiative molecule in the EGFR module) was 680 nM.

- **The Ras activation pathway:**
  The Ras activation pathway included both reversible and irreversible reactions which were expressed in mass action kinetics. We estimated the bond graph parameters of the reactions by applying the parameter balancing technique in which we included an additional constraint (relatively small $k^-$) for each irreversible reaction to limit the reverse flux (Section 2.2.1). Fig 12 demonstrates the behaviour of four species in the reduced Brightman & Fell Ras activation model and its equivalent bond graph approximation. The bond graph equivalent could follow the same trend as in the CellML reduced model with negligible error (0.07%<NRMSE<0.2%). Concentrations have no dimensions in the original CellML model to balance the units [55].

- **The MAPK cascade:** The bond graph model of the MAPK cascade was developed by Pan et al. [32]. We have reused the model here with slightly different configuration of the modules.
  The bond graph version of the MAPK cascade in BondGraphTools was simulated with an initial amount of Ras = $3 \times 10^{-5}$ ($\mu$M). A minor increase in the concentration of the input kinase results in amplified sigmoidal responses of downstream kinases, referred to as ultrasensitivity (S1 Fig) [43]. Amplification in the layers of the MAPK cascade form the ultrasensitive responses, *i.e.*, single phosphorylation-dephosphorylation in the first layer and dual phosphorylation-dephosphorylation in the second and third layers. At this point, we plotted the steady-state responses of the activated kinases against a range of input concentrations ($10^{-8}$—$10^0$ ($\mu$M)) in Fig 13. Note how for inputs less than $7 \times 10^{-5}$ ($\mu$M) MKPP reaches a

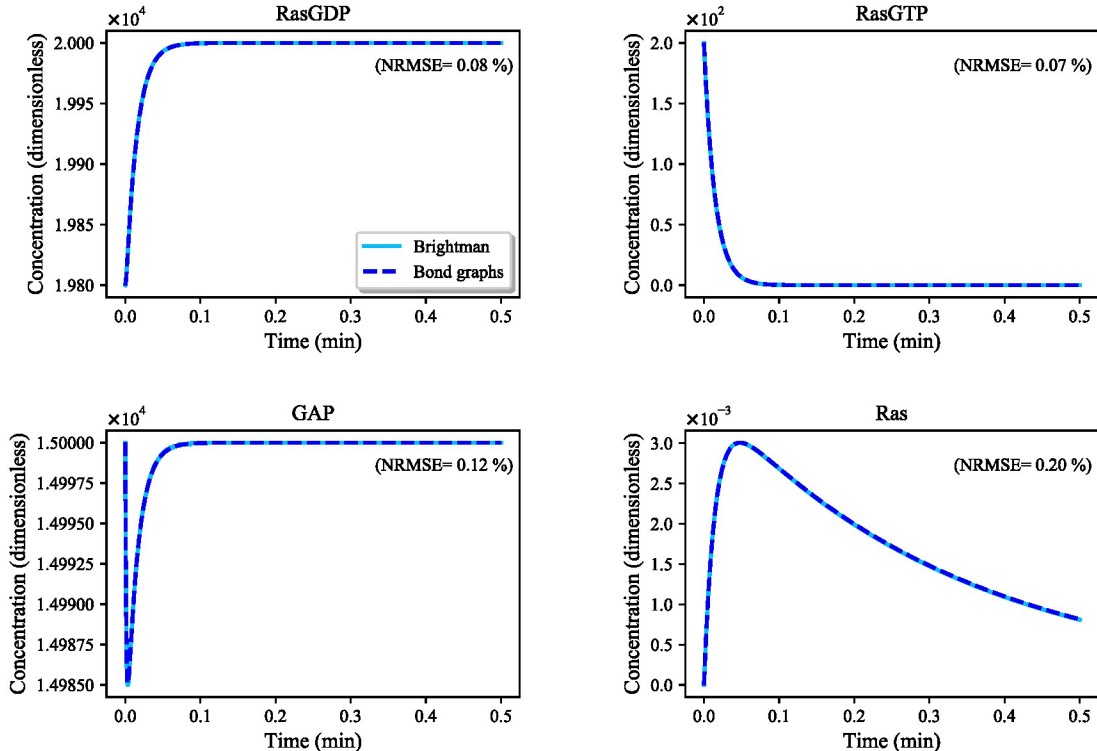

**Fig 12. Comparison between the reduced Brightman & Fell Ras activation model and its bond graph approximation.** The simulations are given for four species. NRMSE is calculated for each comparison in percentage. The initial amounts in this simulation were 0 except for: RasGDP = 19800, RasGTP = 200, GAP = 15000.

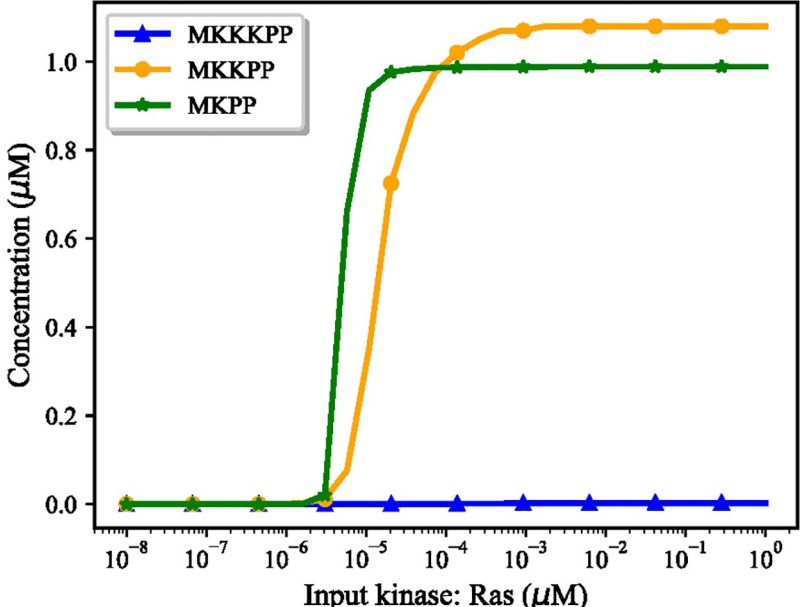

**Fig 13. The steady-state responses of the activated kinases for different input amounts in the MAPK cascade model.** The input Ras concentration is expressed on a logarithmic scale and each curve is normalised to the maximum reached concentration of that species.

**Table 1. Input differences in reaching from 10% to 90% of maximum concentration in kinases.**

| Kinase | 10% maximum response* | 90% maximum response* | Input increase | $n_H$ |
|---|---|---|---|---|
| MKKKP | 0.00027 | 0.0024 | 80-fold | 1.002 |
| MKKPP | 0.108 | 0.972 | 12-fold | 1.768 |
| MKPP | 0.098 | 0.88 | 2.5-fold | 4.795 |

* Concentration amounts are in $\mu$M.

higher concentration than MKKPP while MKKPP overtakes MKPP for higher input concentrations. S2 Fig shows the relative activation of the kinases indicating the lower the layer in the MAPK cascade, the smaller the input concentration activates the kinases [32]. Note that the MKPP (third layer) activation curve is steeper compared to MKKPP (second layer) and MKKKP (first layer), projecting that a higher increase in the stimulus is required for MKKKP to reach its maximum response compared to MKKPP and MKPP (Table 1). The analysis of the behaviour of the MAPK module assisted us to predict how the kinases will respond to the input kinase (Ras) coming from the upstream module (the EGFR pathway) and validate our composed bond graph model.

Table 1 delineates the required stimulus increase for each activated kinase to reach from 10% of its ultimate concentration to 90%. This affirms the ultrasensitive responses to the input as we go to the lower layers of the cascade. To estimate the ultrasensitivity in sigmoidal input-output curves, the Hill coefficient (nH) is also calculated per activated kinase as per Eq 7, where EC90 and EC10 are the input values required to produce 90% and 10% of the maximal response, respectively [56]. The greater the Hill coefficient than 1, the smaller input value is required for the concentration transition from 10% to 90% of its maximum amount. The figures are consistent to the predicted Hill coefficients for MAPK cascade in work by Huang & Ferrell [57].

$$n_H = \frac{\log(81)}{\log(\text{EC90}/\text{EC10})} \tag{7}$$

## 3.2 Verification of the bond graph composed EGFR-Ras-MAPK model

We investigated the behaviour of our bond graph composed model (EGFR-Ras-MAPK pathway) under four conditions: **without negative feedback**, **with negative feedback**, **different ATP concentrations**, and **different EGF concentrations** to examine the functionality of our model under varying conditions. Each of these four conditions imply qualitatively predictable changes in the behaviour of the whole network which we aim to investigate in our composed model.

- **Without negative feedback:**
  The simulated time courses of the three activated kinases (MKKKP, MKKPP, and MKPP) in the composed bond graph model of the EGFR-Ras-MAPK pathway are shown in Fig 14A. Fig 14B predicts the activated kinases at steady-state for various input concentrations. The concentration of the input kinase (Ras) at $t = 100$ (s) was 0.311 nM, which is indicated by the purple dashed line in Fig 14B. The intersection of this line with the MKKKP, MKKPP, and MKPP concentrations shows the expected steady-state concentrations of the aforementioned kinases.

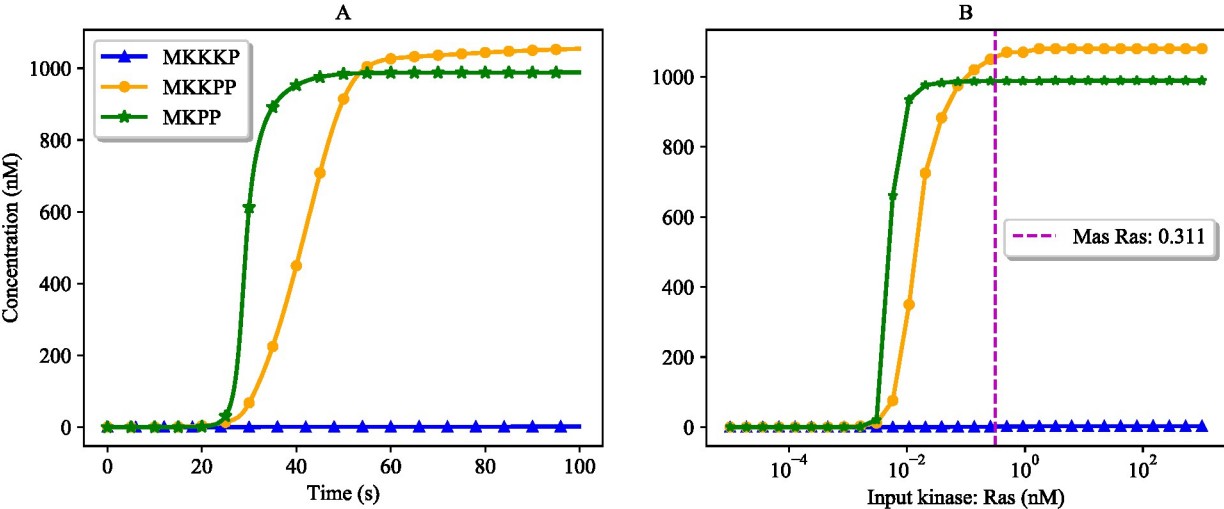

**Fig 14. Verification of the responses of activated kinases to Ras in the composed EGFR-Ras-MAPK bond graph model by comparing with the predicted steady-state responses in the MAPK cascade module.** (A) Ultrasensitity in the composed EGFR-Ras-MAPK bond graph model. The steady-state concentrations of the kinases are: MKKKP = 1.37 nM, MKKPP = 1054.37 nM, MKPP = 987.96 nM; (B) Predicted steady-state concentration of the kinases. The purple dashed line shows the concentration of Ras at $t$ = 100 (s) in the composed EGFR-Ras-MAPK bond graph model. The predicted steady-state concentrations of MKKKP, MKKPP, and MKPP at Ras = 0.311 nM match with the ones in the composed EGFR-Ras-MAPK bond graph model.

- **With negative feedback:**
  Negative feedback in MAPK cascade may lead to inhibited responses or oscillations depending on the stability points of the system [45]. The activated kinases respond differently when a negative feedback loop is added to the system. This feature was also explored in our composed bond graph model of the EGFR-Ras-MAPK pathway.
  Fig 15 compares the activation of kinases in the MAPK cascade model in two cases: without negative feedback (Fig 15A) and with negative feedback (Fig 15B). Under the effect of a

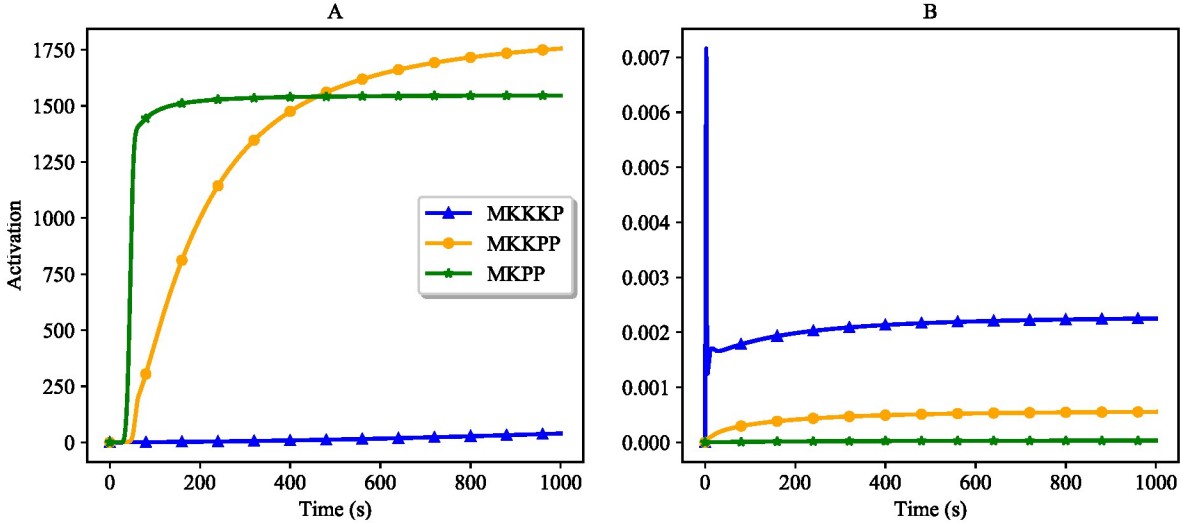

**Fig 15. Activation of terminal kinases with and without negative feedback in the composed EGFR-Ras-MAPK bond graph model.** (A) Without negative feedback; (B) With negative feedback.

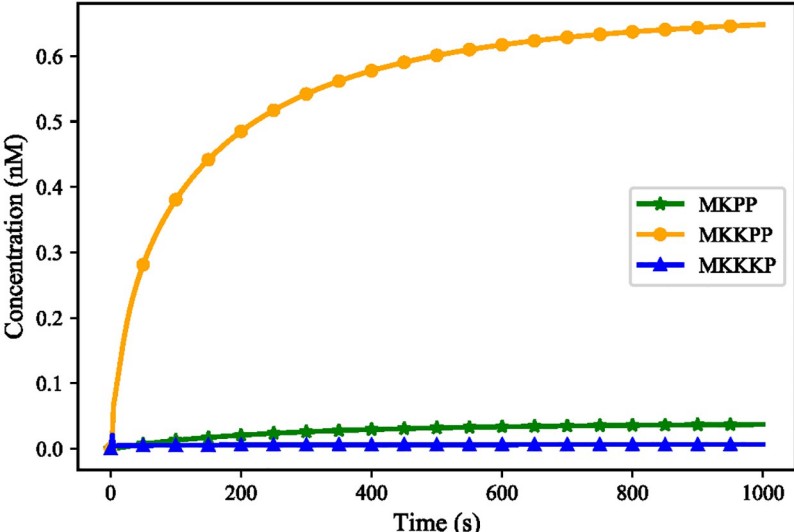

**Fig 16. Time course behaviour of the terminal kinases in the composed EGFR-Ras-MAPK bond graph model with negative feedback.**

negative feedback loop in the MAPK cascade the activation of the kinases decreases as we go further downstream. Fig 15B shows the expected functionality of the MAPK cascade module in the presence of negative feedback where MKPP is less activated than MKKPP and MKKKP. The peak in the activation of MKKKP in Fig 15B corresponds to an initial rise in MKKKP concentration from the upstream MKKK where it is immediately consumed by the downstream species to activate MKKPP and MKPP.

Fig 16 shows the inhibited responses of the terminal kinases and subsequently, the significant delay in reaching their steady-states (compare with Fig 14A). The added Negative feedback in the EGFR-Ras-MAPK model strengthens the dephosphorylation reaction in the first layer of the MAPK cascade module which receives the Ras stimulus. This strengthened dephosphorylation inhibits its corresponding phosphorylation pair and affects phosphorylation in all the proceeding layers.

- **ATP concentration:**

ATP is one of the species involved in prompting wound responses that activates the MAPK pathways in cells [58]. As such, ATP shortage causes delays or failure in activating kinases, and as a result, dysfunction in wound healing responses. The production of ATP in cells might be blocked or reduced due to multiple reasons, such as mitochondrial disorders, ageing, or very intense exercises [59–61].

The impact of ATP concentration on the behaviour of the bond graph EGFR-Ras-MAPK model was investigated by clamping the ATP concentration at **10%**, **30%**, **50%**, and **100%** of its baseline level (Fig 17). Fig 17A–17C illustrate how different levels of cellular ATP (energy) influence the behaviour of activated kinases and also confirm that ATP shortage induces a delay in the responses. Fig 17D compares the steady-state concentration of MKKKP, MKKPP, and MKPP against various ATP concentrations relatively. The lower the ATP production, the lower the steady-state concentration of MKKKP, MKKPP, and MKPP, highlighting the importance of energy for the function of the pathway. The initial concentration of all other species was not changed. Here, the initial concentration of RShGS and Ras (common species between the modules) was 0.

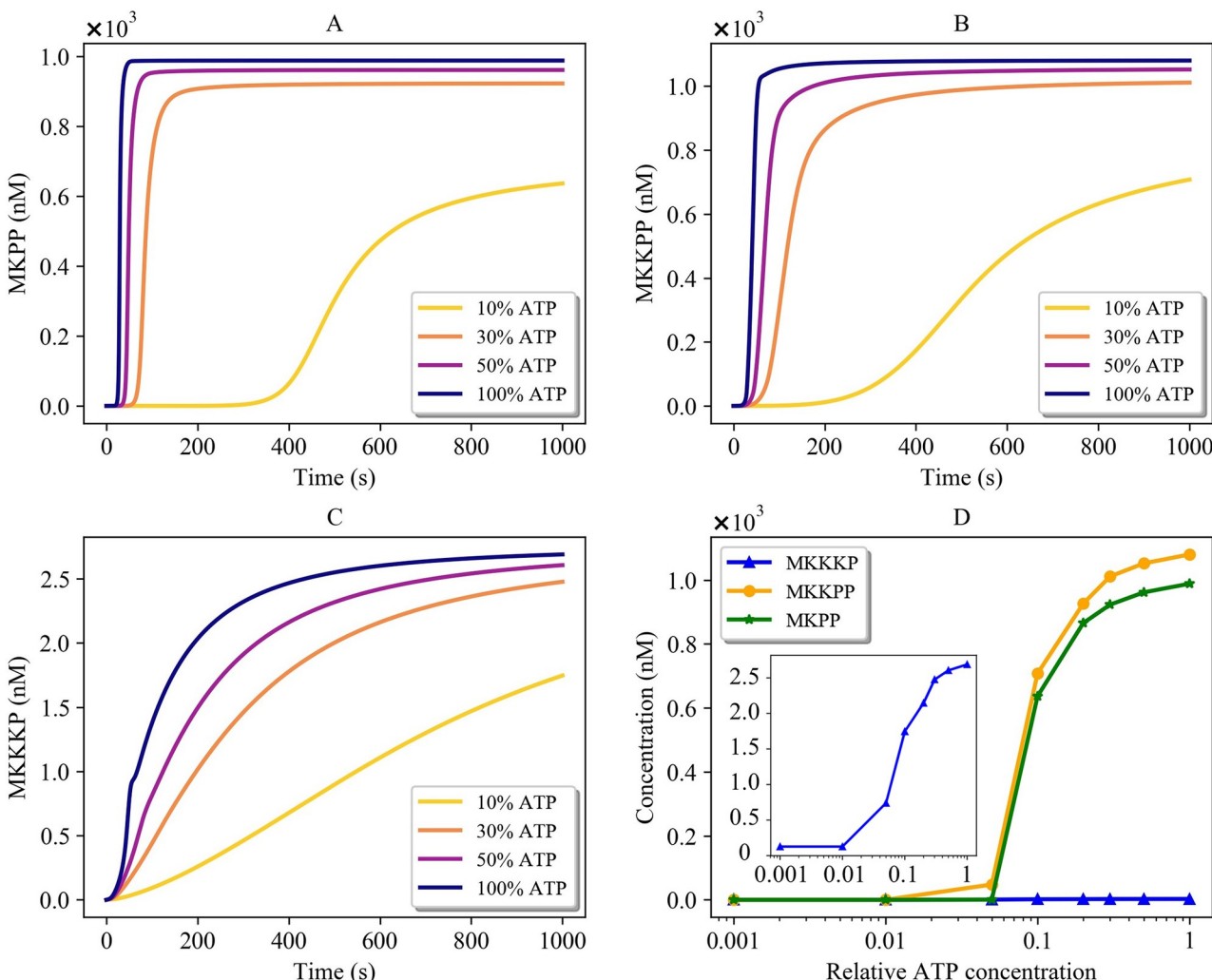

**Fig 17. Effect of different levels of ATP concentration on activated kinases in the composed EGFR-Ras-MAPK bond graph model.** (A) MKPP; (B) MKKPP; (C) MKKKP; (D) Steady-state concentration of MKKKP, MKKPP, and MKPP against relative ATP concentration. MKKKP concentration is also separately shown in a box due to its relatively small amounts compared to MKKPP and MKPP (initial concentration of common species: Ras = 0, RShGS = 0).

- **EGF concentration:**
  We examined the composed bond graph model of the EGFR-Ras-MAPK pathway to analyse and compare its functionality with other similar mathematical models. To do this, we investigated the effect of EGF concentration on MKPP. EGF initiates the EGFR pathway model and MKPP is the last terminal kinase of the MAPK cascade model. Fig 18 illustrates the behaviour of MKPP against various initial concentrations of EGF. Lower concentrations of EGF impose a delay in MKPP to reach its steady-state concentration which emphasises the role of EGF on the downstream species to the end of the MAPK cascade. Note that EGF = 0 nM does not terminate the functionality of the composed model considering that ATP hydrolysis and other intermediate species (such as RasGTP and RasGDP) fuel the subsequent steps and stimulate Ras. The time delay was also studied by Jurado et al. in [62], where lowering the EGF concentration triggered a delay in the MKPP response. Due to the different configuration of the constitutive models and the absence of EGF regulation by MKPP,

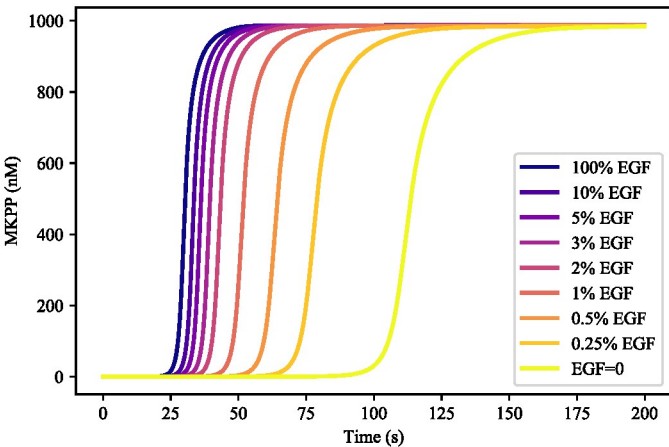

**Fig 18. Effect of different levels of EGF concentration on MKPP in the composed EGFR-Ras-MAPK bond graph model.** The concentration of EGF was set to **0**, **0.25%**, **0.5%**, **1%**, **2%**, **3%**, **5%**, **10%**, and **100%** of its initial concentration (680 nM). The behaviour of MKPP changes by altering the initial concentration of EGF.

the MKPP concentration in our composed model plateaus instead of descending as illustrated in [62].

## 4 Discussion

In this paper, we introduced a generic approach to assemble computational models in biology without starting from scratch. This was enabled by constructing symbolic bond graph modules of biophysical systems and obtaining the required parameters from existing models. To extract and allocate the parameters, the conventional target model needs to be fully and properly annotated. Thereafter, all the biochemical reactions (reversible or irreversible) in the reference models are converted into bond graph compatible ones (Section 2.2.1). The modules will then automatically combine when the common components (species) among them are merged. The resulting composed bond graph model complies with the laws of physics and can be coupled to other bond graph modules. As an example, we applied our method to the EGFR-Ras-MAPK pathway.

Our composed model of the EGFR-Ras-MAPK signalling pathway is different from the ones in the literature in three ways which prevents us from conducting a direct comparison:

- Our reference models of the EGFR pathway, Ras activation, and the MAPK cascade are adopted from different sources which included/excluded some reactions or feedback effects;

- Some reactions in the original EGFR and Ras activation models were irreversible and therefore thermodynamically infeasible;

- Kholodenko et al. regarded RShGS and RGS in their EGFR model to impact the Ras activation module. However, Brightman & Fell did not account for RGS contribution in their model. Hence, we only merged RShGS in the EGFR and Ras activation modules.

We validated the composed model by comparing the behaviour of the terminal kinases to the predicted behaviours from running the MAPK cascade model solely (Fig 13). The purple dashed line in Fig 14B denotes the steady-state concentration forecast of each activated kinase at Ras = 0.311 (n mol). The results gained from our composed model in Fig 14A comply with the predicted ones in Fig 14B.

Merging components across models might raise mismatches in their parameters. Here, **RShGS** in the EGFR and Ras activation models, and **Ras** in the Ras activation and MAPK models were merged. These species have different initial values and/or thermodynamic constants in their corresponding models. In such cases, our framework flags different values for same species. This is solved by asking the user to either select one of the values or insert a new value for the flagged parameter. Since the user may not have the relevant expertise, we aim to provide users with an evaluation of the ambiguous parameter in multiple models available on PMR in the future. This will give the user a better awareness of the range of values for uncertain parameters.

Our bond graph composed model allowed us to investigate the role of EGF concentration and ATP on kinases while this was not possible on the individual models before composing them. While ATP hydrolysis is mentioned in the Kholodenko et al. EGFR model, it is not included in their computational model. Our composed bond graph model accounts for the missing energy sources that firstly provides a more biologically realistic model and secondly, enables us to examine hypotheses on ATP shortage in the EGFR-Ras-MAPK pathway.

In general, systems biology models will frequently omit metabolites such as ATP or $H^+$ from their reactions, causing issues for mass and energy conservation. In cases where the selected reference models do not describe this part of the biology, users can apply their knowledge or search the literature to add any missing steps or subsystems to the composed bond graph model manually. To enhance this procedure in future, we could employ genome-scale metabolic models (GSMMs) as scaffolds to identify the missing entities or reactions [63, 64].

There are situations where different representations of a certain reaction or process are available through the literature. For example, a reaction might be described with or without an allosteric inhibitor. This arises from different applications for different versions of a model and the scope of the studies. In such cases, one has to decide which version of the model they want to use in model composition.

Our parameter optimisation method is similar to the parameter balancing method utilised by Stanford et al. in [14] in using the thermodynamic constants. While parameter balancing is based on assumptions about typical ranges of parameters and probability distributions [65], our parameter optimisation technique concerns the replication of the model performance with the least square error. In the future, we can utilise other techniques such as parameter balancing in our approach to incorporate the experimentally measured values of parameters and create more realistic bond graph models.

As an improvement to our previous approach [28], the present framework overcame the aforementioned limitations:

1. No mathematical formulation of bond graphs is required in the CellML modules (formulating symbolic bond graph modules in BondGraphTools is more straightforward and less error-prone);

2. Auxiliary variables are not needed in the CellML modules as linking ports (ports are automatically detected using 'white box' approach by finding identical annotations);

3. Instead of the semi-automated SemGen merger tool, our approach integrates the modules in a fully-automated manner (our implementation automatically merges the modules and performs the required structural changes).

Here, we have selected models encoded in CellML because CellML can deal with models that are not purely biochemical, but the approach can be applied to models in other formats, such as SBML, as long as they can include a semantic description of the system being modelled. While symbolic templates are required to apply our approach to CellML models, this step is

not required for SBML models. This is because the reactant(s)-reaction-product(s) relationships are explicitly defined within SBML models while this information is not clearly provided in CellML models. In this paper, we aimed to illustrate a possible way to convert CellML biomodels into bond graphs and automatically compose them. In future, we intend to apply the same method on SBML models in a more automated way.

Currently, our model composition approach is capable of detecting exact matches. However, this approach could be improved by allowing the user to specify mergeable components from a shortlist of similarly annotated ones. In the future, if the scientific community defines a globally accepted standard to unify the annotation of similar biological models, finding matching annotations among the models will be much facilitated.

Our energy-based model composition approach is designed to link mathematical models encoded in CellML to their bond graph equivalent and compose them in a consistent and physics-based environment. Currently, there is no general method of automatically converting mathematical models into bond graphs, and each model requires domain-specific expertise to generate a similar bond graph form. To reuse and compose the massive number of existing biological models, the community should either push the researchers to build thermodynamically consistent and physically plausible models or encourage the researchers to develop computational tools that convert existing biological models into bond graphs.

If a model follows the laws of physics and thermodynamics, it can be directly converted into bond graphs. Otherwise one must make assumptions to produce a bond graph that approximates the original model. To facilitate such decisions, we propose establishing an evaluation system to check whether the original model is physically realistic or not. If the model cannot represent a physically plausible system and its bond graph approximation does not fit the data, it highlights some inconsistencies in the original model that must be noted and fixed.

The ultimate goal of applying our model composition method is to provide a foundation for future tool developments to convert any arbitrary CellML/SBML model into bond graphs and then convert it back to a CellML/SBML file. We require the bond graph conversion for appending, deleting, and editing modules. This allows us to firstly avoid any errors or confusions during the process, and secondly, make sure that the model conserves energy and mass and remains thermodynamically and physically consistent as we modify it. Eventually, the generated mathematical equations in the bond graph environment can be exported to CellML for simulation and reproducibility. The regenerated bond graph model encoded in CellML will lose its graphical structure and the model will be expressed as a system of ODEs. Since we can convert the exported bond graph ODEs into MathML format, the biochemical equations would be also expressible in SBML. The structure of such SBML models will be preserved since the required parameters, rate laws, and reactant(s)-reaction-product(s) relationships are extractable from the generated bond graph model.

Models are constructed in different units for parameters and various scales of amounts. Coupling arbitrary models will alter their boundary conditions which induces differences that propagate throughout the models. In the future, we plan to apply nondimensionalization to remove dependencies to the measured units across the models and generate unified composed models, regardless of their units [66]. Nondimensionalization is especially useful in models that are described by differential equations. In this systematic technique, all variables and parameters become unitless by rescaling them relative to a reference value.

Another widely-used formalism in computational biology is rule-based modelling, in which a series of rules describe the mechanistic details of biochemical processes, for example the random binding of multiple ligands to a receptor [67]. Recently, rule-based approaches have incorporated energetic parameters to ensure thermodynamic consistency [68–70]. Danos et al. showed that by computing the free energy of species formation and hence, free energy

inequalities in reactions in rule-based models, one can verify whether a model satisfies the free energy constraints and detailed balance [71]. Moreover, rule-based languages such as BioNet-Gen [72] and Kappa [73] allow annotations. One advantage of bond graph modelling over rule-based modelling is that they can model multi-physical systems such as electrophysiology, whereas rule-based approaches are limited to the biochemical domain.

# 5 Conclusion

We have developed a method that automates the integration of biosimulation models. We utilised the SemGen annotator tool to add metadata to CellML models and the Python library BondGraphTools to generate the bond graph template of models. Describing the bonds between bond graph components with connectivity matrices helped us conveniently delete or add bonds/components to the modules. This minimises user error when a structural change is required in complex systems. Here we have presented a method that automates the composition by taking advantage of semantics in the modules and the systematic structural modification using connectivity matrices. We demonstrated the functionality of our method by coupling two biosimulation models and their sub-models. Likewise, several annotated biosimulation models can be integrated automatically if they have common entities. This is particularly pivotal when dealing with complex and large biological systems where mathematically merging models requires time-consuming and error-prone post-composition adjustments. We believe that our method is one of the initial steps toward multiscale cell-to-organ-level model integration.

# Supporting information

**S1 Fig. Ultrasensitivity in MAPK cascade.** For an input kinase of Ras = $3 \times 10^{-5}$ ($\mu$M), the concentration changes of the activated kinases (MKKKP, MKKPP, and MKPP) show the signal is amplified through each layer.
(TIF)

**S2 Fig. The normalised activation of kinases in the MAPK cascade module for different input amounts (Ras).**
(TIF)

**S1 Table. Reactant(s) and product(s) of each step in EGFR pathway and the reaction rate equations.** Steps 4, 8, and 16 are irreversible reactions, which are approximated by mass action kinetics. $\kappa_i(i \in \{$Step$\})$ in the reaction rate equations represent the reaction rate constants, $K_x$ ($x \in \{$Reactants, Products$\}$) is the thermodynamic constant of each species, and $q_x$ ($x \in \{$Reactants, Products$\}$) is the concentration amount of each species.
(PDF)

**S2 Table. Original and modified parameters of the species in the EGFR pathway model.**
(PDF)

**S3 Table. Original and modified parameters of the reactions in the EGFR pathway model.**
(PDF)

**S4 Table. Reactant(s) and product(s) of each step in the Ras activation pathway and the reaction rate equations.** Steps 2 and 4 are irreversible reactions, which are approximated by mass action kinetics. $\kappa_i(i \in \{$Step$\})$ in the reaction rate equations represent the reaction rate constants, $K_x$ ($x \in \{$Reactants, Products$\}$) is the thermodynamic constant of each species, and $q_x$ ($x \in \{$Reactants, Products$\}$) is the concentration amount of each species.
(PDF)

**S5 Table. Original and modified parameters of the species in the Ras activation pathway model.**
(PDF)

**S6 Table. Original and modified parameters of the reactions in the Ras activation pathway model.**
(PDF)

**S1 Text. Supplementary material.** Appendix A: Connectivity matrix example. B: Parameter estimation for step 4 in the EGFR pathway model. Appendix C: An example of composing two reactions in bond graphs. Fig A: An example network with its connectivity matrix. Fig B: The irreversible Michaelis-Menten and its equivalent approximated reversible mass action kinetics for step 4 in the EGFR signalling pathway model.
(PDF)

# Acknowledgments

NS would like to thank Yuda Munarko for his helpful comments and suggestions. EC passed away before the submission of the final version of this manuscript. NS accepts responsibility for the integrity and validity of the data collected and analysed.

# Author Contributions

**Conceptualization:** Niloofar Shahidi, Michael Pan, Edmund J. Crampin, David P. Nickerson.

**Data curation:** Niloofar Shahidi.

**Formal analysis:** Niloofar Shahidi.

**Funding acquisition:** Edmund J. Crampin, David P. Nickerson.

**Investigation:** Niloofar Shahidi.

**Methodology:** Niloofar Shahidi, David P. Nickerson.

**Project administration:** Michael Pan, Kenneth Tran, Edmund J. Crampin, David P. Nickerson.

**Resources:** Edmund J. Crampin, David P. Nickerson.

**Software:** Niloofar Shahidi.

**Supervision:** David P. Nickerson.

**Validation:** Niloofar Shahidi, Michael Pan, Kenneth Tran, Edmund J. Crampin, David P. Nickerson.

**Visualization:** Niloofar Shahidi, Michael Pan, Kenneth Tran, Edmund J. Crampin, David P. Nickerson.

**Writing – original draft:** Niloofar Shahidi.

**Writing – review & editing:** Niloofar Shahidi, Michael Pan, Kenneth Tran, Edmund J. Crampin, David P. Nickerson.

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
