## [Decision Letter · Decision Letter 0]

28 Feb 2022

PONE-D-21-39103A semantics, energy-based approach to automate biomodel compositionPLOS ONE

Dear Dr. Shahidi,

Thank you for submitting your manuscript to PLOS ONE and please accept my sincere condolences on the loss of Professor Crampin. After careful consideration, we feel that it has merit but does not fully meet PLOS ONE’s publication criteria as it currently stands. Therefore, we invite you to submit a revised version of the manuscript that addresses the points raised during the review process. Please carefully address the comments by all three Reviewers and note, Reviewer #1 has provided a four-page report as an attached pdf file and the below text just shows a summary thereof. Especially, improvements to the structure and clarity of the manuscript text are required. Following comment 3a by Reviewer #1, a verbal description of the hierarchical merging of multiple models is required but no larger model would need to be explicitly constructed here. Regarding comment 3f by Reviewer #1, a general verbal description of model behaviors (e.g. due to closed feedback loops) in the merged model that were broken in the un-merged models is required but no specific new predictions or comparison with experimental data or literature will be required for this EGFR-Ras-MAPK example here. The separately available figure files (tiff) are all of high resolution, so you may ignore the last comment on figure rasterization by Reviewer #2.

We look forward to receiving your revised manuscript.

Kind regards,

Lutz Brusch, Ph.D.

Academic Editor

PLOS ONE

Journal Requirements: 

Reviewers' comments:

Reviewer's Responses to Questions

**Comments to the Author**

1. Is the manuscript technically sound, and do the data support the conclusions?

Reviewer #1: Partly

Reviewer #2: Partly

Reviewer #3: Yes

2. Has the statistical analysis been performed appropriately and rigorously? 

Reviewer #1: N/A

Reviewer #2: N/A

Reviewer #3: N/A

3. Have the authors made all data underlying the findings in their manuscript fully available?

Reviewer #1: Yes

Reviewer #2: Yes

Reviewer #3: Yes

4. Is the manuscript presented in an intelligible fashion and written in standard English?

Reviewer #1: Yes

Reviewer #2: No

Reviewer #3: Yes

5. Review Comments to the Author

Reviewer #1: I am of the opinion that energy-based model composition will play a critical role in expanding the utility of reaction models in biochemistry. The authors have applied multiple innovative strategies such as bond graphs, templating, automated semantic inference, and automated merging to their model composition pipeline, so the premise of the paper is sound. However, it disappoints me to recommend rejecting this manuscript on several counts.

1. It fails to properly introduce the problems being solved, how they originate in biology, and how the proposed methods solve those problems.

2. It fails to introduce the technical background of the methods being deployed in a manner accessible to a general biological audience, or even an audience interested in biochemical models.

3. It has multiple issues with the demonstrated results: questionable choices in merging, insufficient demonstration of proof of concept, poor structuring and comparison of simulation results.

4. Severe readability and copy-editing issues with the text.

Issue 3a and 3f (please see attached PDF for full review) are critical reasons behind recommending rejection over revision.

Reviewer #2: # A semantics, energy-based approach to automate biomodel composition

## Major comments

This paper describes a methdology of using Bond graphs to create complex composite models.

This is a topic many people (myself included) will want to learn more about, and the accompanying software could potentially be very useful.

However, there are several issues with the manuscript text that will need addressing before this can be published.

Most notably:

1. The introduction and methods section are far more abstract than they needs to be. For example, the text frequently refers to "entities", "elements", "components" of bond models without making clear what these words refer to (or if they are the same), it mentions "similar annotations" without saying what this means, etc. Such parts of the text should be rewritten to be both clear and precise.

2. The problem statement is unclear. In several places it seems to be about coupling two (known) models, but parts of the methodology imply it is about identifying models that could potentially be coupled?

3. As the problem statement is an abstract one, an example (perhaps a simple toy problem in addition to the real-life example shown) should be introduced early on (i.e. in the introduction) and used to explain the problem statement. In the current manuscript a general outline of a solution is being sketched long before the reader has been given the tools to understand the problem this aims to solve.

4. The level of detail varies considerably throughout the paper, lots of words are devoted to fairly simple processes such as creating a connectivity matrix and removing components that have been deemed identical, but very little is said about more complex steps, e.g. how annotations are compared, how "thermodynamically consistent parameters" are created from inconsistent ones etc.

## Detailed comments

### Line 17

> The main challenges in hierarchical model composition include: (a) incompatible code languages, (b) different modelling frameworks, (c) post-composition adjustments, and (d) physically implausible resultant models.

This needs a slower and more careful explanation. What is the difference between an "incompatible language" and a "different framework"? What are "post-composition adjustments". Can you give an example?

### Line 20

> A majority of model integration platforms ... require compatibility between

the languages and modelling frameworks

> In contrast, the resultant model still needs further post-merging code-wise adjustments to be executable, yet it might not represent a physically feasible model.

How are these two statements "In contrast"?

Are "post-merging code-wise adjustments" the same thing as "post-composition adjustments"?

### Line 26

> One solution to these issues is using a hierarchical modelling approach (to help with the post-composition adjustments) and an energy-based modelling framework (to guarantee a physically plausible composed model).

I'm not sure if you are (A) stating something I should already be able to understand at this point, of if you are (B) saying this will be explained in the upcoming text. If A, then it needs a lot more explanation, if B then please rewrite the text so that this is clear.

### Line 53

> The annotated data from the CellML modules are then extracted and assigned to their equivalent bond graph components.

By "data" do you mean data (i.e. parameter values?), or e.g. equations, units, etc. ?

And what is a "component"? Should data not be connected to data, or variables to variables? Please rewrite to make statements such as these less abstract and more clear.

### Line 55

> We demonstrate this by an example where a bond graph model is constructed from its constitutive modules, i.e., the Epidermal Growth Factor Receptor (EGFR) signalling pathway and the Mitogen-Activated Protein Kinase (MAPK)

Given this example, I'm a bit surprised you opted for CellML over SBML. This may warrant some discussion.

### Line 79 Materials and methods

Please make this whole section less abstract, e.g. by starting with a detailed example of two models that you wish to connect, so that the reader has some idea what you mean by the various "entities" and "elements" that appear in the text.

### Line 86

> automatically rewiring the connections between components and modules

What are "components" in this sentence? The introduction should probably give a very brief explanation of what a bond graph looks like (maybe a figure?) and explain what the difference between e.g. a bond graph "module" and a bond graph "component" is. (Is it just a variable? Then please say so)

### Line 92

> Our method to expand and integrate biosimulation models provides the foundation for further developments in an open-source environment based on energy-based modules and automation to minimise manual input.

I'm not sure what this sentence means? Without the "to minimise manual input" it reads like part of an abstract or conclusion section. But are you just trying to say you want to minimise manual input when composing models?

### Line 94

> In this endeavour, we have provided some exemplar symbolic bond graph models to which the annotated parameters from the CellML modules would accordingly link.

Does this mean that any time I want to link 2 models, I need to write 2 templates (that presumably match the 2 models I want to link), and then write a connection matrix describing how my two templates are linked? If it's so hard-coded, then why not just write down which variables I want to connect straight away?

### Line 99

> The suitable bond graph model is then automatically selected from the list by

Which list?

### Line 100

> identifying specific annotated components in the CellML modules

What exactly is a "component" here? Is it a CellML component (i.e. a container of variables)?

### Line 117

> Ontologies

I'm not sure who the intended audience for this paper is, but if it's "anyone who wants to compose a big model" you should probably add a line saying what an ontology is, or maybe add a reference.

### Line 121

> We suggest downloading CHEBI, FMA, OPB, and GO ontologies from the following links:

Please explain what domains these 5 are, and why they are appropriate (are they approriate for everything, or just for your two pathway example?)

> We used the OPB and GO ontologies for the particular case study in this paper.

Why do I need the other two then?

Please explain what these ontologies will be used for. Do they provide labels identifying unique variables? Variable types e.g. "is a concentration"? Will we be inferring properties of our variables using these ontologies e.g. units?

### Line 128

> The generic approach

Do you mean generic, i.e. this section explains why the approach works on any model, or "general approach", so that this section presents an outline of the approach taken in the paper?

### Line 131

> the parameters in the models

Please explain in the intro what you expect to find inside each model (should this say "module"?), e.g. variables, parameters, etc. and use these terms throughout instead of "elements", "entities", "components" etc.

Is a distinction made between variables and constants/parameters?

### Line 138

> The number of rows and columns each equals the number of elements in the network in total

What are the "elements" and the "network"? Are elements modules? Variables in a module?

Figure 1 and the accompanying text are far too vague and focus on the wrong aspects: Readers who are interested in this topic will have come accross the topic of a network and a connectivity matrix before, and will be wondering instead how it applies to your problem of tying two pathway models together. This second bit is not explained in any detail here.

If they are variables, then what are we expecting from the on

### Line 142

> "facilitates computational measurements"

What is a "computational measurement" and how is it facilitated by a binary representation of these connections?

### Line 144

> Modifying a network is easily performed by inserting 0 or 1 in the matrix

In two places, to preserve symmetry? Given the zero diagonal, the symmetry (which the user needs to manually maintain), and the presumed sparseness of this matrix in real applications (if elements are variables, as I suspect at this point in my reading), this is not a very compact or easy-to-use representation.

### Line 152

> In a `black box' composition approach, the elements of the modules are not accessible and only the input/output variables can be used for coupling

It's not clear to me why we'd expect the modules to have clearly defined "inputs" and "outputs" at this point, so I'm struggling a bit to see why you need to point out that any variables can be connected.

### Line 154

> almost all the entities can be regarded as merging ports

What is a "merging port"? And do we need this bit of jargon or can it be stated more simply?

### Figure 2

What are "similar annotations"? Do both models have the exact same set of RDF properties? The exact same unique labels? Is it OK if only a subset matches?

This figure seems to show 2 preselected models, but there was also mention of a "list", where does that fit in?

How is the "repository" section related to the two models?

### Sections 2.2 & 2.3

These are very helpful, but are required reading to understand e.g. 2.1, so some re-structuring is necessary!

### Line 216

1. Is the "." in the equation meant to be a \\cdot? Also, if a multiplication symbol is used here it should be present in the other two multiplications (R*T*ln(K_q*q) or RT ln(K_q q))

2. What is u_q in this equation?

### Line 222

> A reaction represents a dissipative process, which in the case of mass-action kinetics...

This could do with some clarification. Are all reactions "dissipative"? Mass-action kinetics are usually phrased in terms of reversible processes.

### Line 250

> and the bond graph model of MAPK cascade is taken from the work by Pan et al. [21]. In this paper, the bond graph representation of the reference MAPK cascade was available. Here, we detail how bond graph models of these systems were constructed.

The second sentence repeats the first. But then the third contradicts it?

### Figure 4

> The network adapted from

Missing is/was

### Line 294

> We applied curve fitting to estimate the reaction rate constants for the irreversible steps (κ4 , κ8 , & κ16 ). We obtained the time-dependent behaviour of the contributing species in steps 4, 8, and 16 (required for curve fitting).

1. If the second sentence is required for the first, it should come first.

2. These two steps contain a lot of work. Much more explanation is needed to make this reproducible.

### Line 299

> some of the reversible reactions do not satisfy detailed balance

I had assumed this would be guaranteed by the bond graph methodology, some comment or a reference could be useful here.

### Line 300

S1 Table. S2 Table and S3 Table  Table S1. Table S2 and Table S3

### Lines 339, 343, 344, 345

> model of EGFR-Ras-MAPK

> and MAKP cascase

> Since MAPK cascade includes

Many missing "the"s throughout the text

### Lines 350

> Due to the limited size ...

This would make more sense in section 2.1.1

### Line 354

> for inconsistencies among the values of similarly annotated components and parameters

What is meant by "inconsistencies" here? Please be precise and give examples.

### Line 510

> For biochemical reactions, if the parameters are thermodynamically inconsistent, they are converted into bond graph compatible ones.

Where is this process explained?

### All figures

This could be just a proof issue, but the figures are all rasterised, and at a low resolution.

Reviewer #3: In their manuscript “A semantics, energy-based approach to automate biomodel composition”, Shahidi et al. describe a new framework for combining biochemical network models based on a mathematical representation in the form of bond graphs. They describe the method and how it is designed to guarantee thermodynamic correctness of the resulting models, and illustrate the procedure with an example case, combining two existing signaling pathway models into a larger, consistent model.

The manuscript is extremely well and clearly written and was a pleasure to read. I think that the method will be very useful. Since it has already been implemented for CellML models, and an implementation for SBML models is conceivable, it has the potential for broad applications in biochemical pathway and network modeling.

I did not check the code.

I have no substantial criticisms. Below I list a few minor points to improve the manuscript, mostly about clarification of words. I leave it to the authors to decide which of these points they would like to account for.

Finally, I would like to express my condolences to the authors for the passing of Professor Crampin. It must have been painful for you to complete the work without your colleague.

Title: The term “bond graph” could be mentioned in the paper title.

2: “Physicians”: I think it’s a dream of modelers that their models will be used by physicians, but I think we’re usually still far from this.

27 (and elsewhere): “Energy-based modeling framework”: since “energy-based” can mean many things, it would be good to explain this term very clearly and explicitly early on.

60: “provides a reliable and consistent framework that first conserves energy” again, not very clear. The need to satisfy thermodynamic Wegscheider conditions does not exactly arise from energy conservation (first law of thermodynamics), but is also related to the second law, and basically the fact that Gibbs free energy is a thermodynamic potential. So “conserves energy” is a bit imprecise, and maybe not very well understandable.

101: “Due to the hierarchical feature of bond graphs,” also, “hierarchical” is not very clearly explained. I guess it refers to the usage of symbolic templates and (“inside” them) the actual network-like model structures. But these are just two layers, and “hierarchical” sounds like there were many hierarchy layers.

109: “A common effort between the components is shown by a ‘0’ junction, while a ‘1’ junction shows a common flow, and the energy is conserved and travels between components bidirectionally through bonds (shown by harpoons).” This explanation is not very easy to get, please epxlain in more detail (e.g. mentioning little examples?)

147: “Notice that the connectivity matrix is symmetric” at first, not clear if the connectivity matrix is always symmetric (by definition) or just happens to be symmetric in this example application.

Section 2.2 explains the usage of bond graph modeling of biochemical reactions, but it remains unclear how parameters, rate law formulae, and other data attached to the nodes will be treated during model composition. Is an enzymatic rate law a property attached to the reaction node? What if in model combination, the same reaction is described (in the two models) in different ways, e.g. with or without an allosteric inhibitor? Is this just a choice between data attached to the reaction node, or a choice between different structures of the bond graph?

247: “As such, we consider Ras protein to be the mutual species in both pathways.” Would there be additional complications if models are connected by several species (e.g. closing thermodynamic loops that were not present in the initial models, but in the combined model”?

In the bond graph model, saturable rate laws were described by irreversible Michaelis-Menten kinetics. Would it also be possible to use reversible Michaelis-Menten kinetics? Or do reversible reactions have to be modelled by mass-action kinetics, for mathematical reasons? (I guess the answer to the latter question is no; maybe it would be good to point this out?)

“The chemostats” - I guess the word refers to species with fixed and given concentrations (sometimes called “external metabolites” in kinetic modelling)? Since the same word is often used in biology in a different meaning (a device with fixed and given concentrations in the INFLOWING medium, not in the bioreactor itself), it would be good to add a short explanation (just say what “chemostat” means in this work).

220: “Hence, a symbolic bond graph module for a cycle could be created and reused.” is “could” past or conditional? Please rephrase.

545: “As an improvement to our previous approach [17], the present framework overcame

the aforementioned limitations: ..” these are indeed great achievements. Congratulation!

583: “Eventually, the generated mathematical equations in the bond graph environment can be converted into CellML for simulation and reproducibility.” Would the results models again have the form of a “normal model”, or do they still look very “bond-graph like”, e.g. with non-biological components representing junctions? (And I have the same question for a (potential, future) conversion to SBML models).

References: “Paynter H. Analysis and Design of Engineering Systems/Paynter HM;.” reference incomplete

Fig 7: The fonts are a bit small

Fig 15: in the legend for subfigure C, maybe mention the little dip in the curve and how it is caused?

Finally, I would like to mention some “historical predecessors” of this work, which tried to establish ways to build biochemical network models from simple “standard elements” while taking thermodynamic feasibility into account.

Ederer M. and Gilles E.D. (2007), Thermodynamically Feasible Kinetic Models of Reaction Networks, Biophysical Journal, Volume 92, Issue 6, 1846-1857,

Stanford N.J., Lubitz T., Smallbone K., Klipp E., Mendes P., Liebermeister W. (2013), Systematic construction of kinetic models from genome-scale metabolic networks, PLoS ONE 8(11): E79195

While the present work is certainly more elegant, it may make sense to cite these earlier works.

Furthermore, the adjustment of parameters to become thermodynamically feasible seems to resemble parameter balancing (which is used in the Stanford et al paper and could also be cited).

6. PLOS authors have the option to publish the peer review history of their article (what does this mean?). If published, this will include your full peer review and any attached files.

Reviewer #1: No

Reviewer #2: **Yes: **Michael Clerx

Reviewer #3: **Yes: **Wolfram Liebermeister

---

## [Author Response · Author response to Decision Letter 0]

19 Apr 2022

Response to Reviewers

A semantics, energy-based approach to automate biomodel composition

We thank the reviewers for their detailed comments on our manuscript. We have addressed the issues as detailed below. The reviewer comments will be shown in black, our responses in green and quotations from the revised manuscript in blue.

Key changes to our manuscript are below:

 (Key change 1) The addition of an extra module for Ras activation

 Reviewer 1 raised issues on both the need to merge more than two models and issues with directly linking Sos to Ras. While we believe that our approach of merging together two models is sufficient to demonstrate the feasibility of our approach, we have now added a RAS activation module as an intermediate between the EGFR and MAPK models to make the model more biologically realistic. This new module is described in Section 2.2.2 and the beginning of Section 3, along with Figs. 5, 10 and 12.

 (121-124) “... describe how the bond graph modules of the EGFR-Ras-MAPK signalling pathway are constructed based on the existing work by Kholodenko et al. [39] on the EGFR model, Brightman & Fell [40] on the Ras model, and Pan et al. [11] on the MAPK model (Section 2.2).”

We outline the model and its parameterisation in the text below:

 (308-318) “Fig 5 represents the kinetic and bond graph structure of the Ras activation module which links the EGFR and MAPK modules.”

 We converted the kinetic parameters of the reactions into bond graph parameters using the same applied techniques in Section 2.2.1. For the irreversible reactions (steps 2 and 4), we assumed a very small value for the reverse kinetic constants (k^- ) to limit the reverse flow to a negligible amount. The reaction equations along with their participating species are given in S4 Table. S5 Table and S6 Table compare the parameter amounts of the Brightman & Fell model with the ones from our reconstructed bond graph model. The code to convert the kinetic parameters into bond graph equivalents for the Ras activation intermediate pathway is accessible from: https://github.com/Niloofar-Sh/EGFR_MAPK/tree/main/Ras.

We have also updated Fig. 10 to show how the intermediate module is merged with the other models.

(558-560) “Here, RShGS and Ras in the Ras activation module are removed while RShGS in the EGFR module and Ras in the MAPK module are kept.”

A comparison of this module to the existing model has been added to the Results

 (591-601) “The Ras activation pathway:

 The Ras activation intermediate pathway included both reversible and irreversible reactions which were expressed in mass action kinetics. We estimated the bond graph parameters of the reactions by applying the parameter balancing technique in which we included an additional constraint (relatively small k^-) for each irreversible reaction to limit the reverse flux (Section 2.2.1). Fig 12 demonstrates the behaviour of four species in the reduced Brightman & Fell Ras activation model and its equivalent bond graph approximation. The bond graph equivalent could follow the same trend as in the CellML reduced model with negligible error (0.07%<NRMSE<0.2%). Concentrations have no dimensions in the original CellML model to balance the units [63].”

 (Key change 2) The connectivity matrix

 Several reviewers raised issues in understanding the connectivity matrix. We acknowledge that the connectivity matrix is not essential to our approach. Nonetheless, we found it to be useful in allowing models to be merged in a general manner.

 We have modified the manuscript to explain this issue:

 (433-438) While not essential to our methodology, we chose this approach because binary representation of models clearly shows the connections and gives the minimal required details to define a network which can be exported to other tools and software for further analysis [46]. To modify a network, one can insert 0 or 1 in the matrix or delete its corresponding row and column. An example in S1 Text.A shows how connectivity matrix is defined for a simple network. 

We agree that describing the details of the connectivity matrix disrupted the flow of the text, so we have moved the figure along with the extensive explanation on connectivity matrices to the supplementary material (S1 Text.A).

To help a reader grasp the merging process facilitated by the connectivity matrix, we have added a simple example in Fig. 2.

Reviewer #1

I am of the opinion that energy-based model composition will play a critical role in expanding the utility of reaction models in biochemistry. The authors have applied multiple innovative strategies such as bond graphs, templating, automated semantic inference, and automated merging to their model composition pipeline, so the premise of the paper is sound. However, it disappoints me to recommend rejecting this manuscript on several counts. Issue 3a and 3f below are critical reasons behind recommending rejection over revision.

1. It fails to properly introduce the problems being solved, how they originate in biology, and how the proposed methods solve those problems.

2. It fails to introduce the technical background of the methods being deployed in a manner accessible to a general biological audience, or even an audience interested in biochemical models.

3. It has multiple issues with the demonstrated results: questionable choices in merging, insufficient demonstration of proof of concept, poor structuring and comparison of simulation results.

4. Severe readability and copy-editing issues with the text.

 We have made substantial changes to the manuscript which we hope address the above issues. To deal with the critical issues 3a and 3f, we have added a third module for the activation of Ras (Section 2.2.2) and included additional simulations showing the effects of EGF on the merged model (Fig. 18).

Issue 1: Introducing the problems being solved

The introduction takes for granted that the reader is aware of scale issues in biochemical modeling. First, they must motivate why system-level modeling of biochemical reactions is difficult. This includes providing information on

 how biochemical complexity affects species semantics, and how that creates problems for model composition,

We have added the following text to describe biochemical complexity and semantics:

 (82-88) “... biological and biochemical complexities can give rise to inconsistencies in semantic annotations. Many species and chemical compounds are not simply defined by a single semantic term. Subtle variations of names for species have long been an obstacle for semantics-based merging tools to integrate models based on identifying similarly annotated species. In this paper, we have used identical semantics for same species and leave it to the scientific community to develop a harmonising system for annotating biomodels.."

 detailed balance and thermodynamic consistency concepts, how they apply to biology, and how current approaches fail at this,

We have added the following text to describe the application of detailed balance and thermodynamic consistency to biology:

(70-79) “In the context of biochemistry, modellers widely use traditional kinetic models. However, in general, kinetic models are not thermodynamically consistent (i.e. energy conserving) unless the parameters satisfy certain detailed balance constraints. Specifically, detailed balance constraints are required to ensure that biochemical loops have zero flux (i.e. dissipate no energy) at equilibrium. These detailed balance constraints become increasingly difficult to derive as biochemical networks become larger. Because bond graph models assign a chemical potential to each species, they automatically adhere to detailed balance constraints. Hence, parameters can be modified without violating thermodynamic consistency [27]. This ensures that model composition respects the constraints on thermodynamics for biochemical systems.”

 how energy-based composition resolves the issue of thermodynamic consistency.

We trust that this has been addressed in the above comment.

The manuscript touches upon these points briefly in several places (e.g., white-box approach, etc.). But it never provides a cohesive structured argument accessible to the reader. It is not sufficient to simply state that other methods produce infeasible models, but to demonstrate what infeasibility means in this context and why it happens.

Issue 2: Introducing the technical background

Both in the introduction and in the methods section, the authors over-explain the mechanics of what they do, but under-explain the concepts relevant to understanding. Thus, it feels like reading a tutorial without grasping the scientific intuition behind it.

 A more detailed representation of related work is necessary. E.g., what are the issues with SBML- hierarchical and state of the art in model composition? What are post-composition adjustments?

 We have added the following text to describe limitations with other model composition methods:

 (23-25) “… post-composition adjustments (manual edits in mathematical equations, rules, or parameters to make the composed models biologically sensible [8]).”

 (26-31) “While existing model integration platforms, such as the SBML Hierarchical package [9] and PySB [10] can resolve issues with code and modelling formalism compatibility, they are limited to the biochemical domain. The resultant model often needs further adjustments to be executable and yet it might not follow the laws of thermodynamics and physics (such as energy, mass, and charge conservation) [12,14]”

 For bond graphs, the authors need to explain separately bond graph theory, bond graph terms and what they mean, and instructions on how to build/visualize/understand bond graphs. Combining all of these into a single section (Sec 2.2) makes it hard to understand. For example, how does one decide where to glue edges on the bond graph? In Fig 3A, 0:u species nodes are connected to a 1:v reaction node, whereas in Fig 5, 0:u nodes are often connected directly to Re:k nodes. My guess is that it does not matter because of some underlying bond graph theory, but I should not have to resort to guesswork in the face of insufficient explanation.

We have added more text to explain the bond graph concepts in Figure 1 (new version):

 (190-194) “As shown in Fig 1.B, the species complexes (A & B as reactants and C & D as products) at either side of the reaction are connected to the Re component through ‘1 : v’ junctions because the pairs share common flows. The corresponding ‘0 : u’ junction for a species can be directly connected to an Re component if it is the only reactant or product of that reaction.”

 Explain the connection between thermodynamic terms and kinetic terms, e.g., is the thermodynamic constant of a species the same as the more familiar free energy of formation of species? What is a dissipative process?

 We have added the following text to define these terms: 

(171-174) “K_q is related to the kinetic free energy of species to participate in reactions and is defined as K_q=1/(V_c q_ref ) e^((u_q^ref)/RT) where Vc is the volume of the compartment, q_ref is the reference concentration (normally 1 mol), and u_q^ref is the standard free energy formation of the species [27].”

(178-179) “In bond graphs, a reaction represents a dissipative process where chemical energy is lost in the form of heat [52].”

 Explain the connection between detailed balance and energy conservation. What does it mean to obey physical laws in the context of reactions (it is not sufficient to just state that they should be obeyed)? The manuscript does not explain it to the general audience: specifically, that on a per-mol basis, energy should be conserved around a loop of reactions, which places constraints on the relationships between kinetic parameters.

 We have added the following text to mention the physical laws in the context of reactions: 

(30-31) “... follow the laws of thermodynamics and physics (such as energy, mass, and charge conservation) ...”

We have added the following text to explain the energy conservation in a loop of reactions (as discussed in issue 1b):

 (70-79) “In the context of biochemistry, modellers widely use traditional kinetic models. However, in general, kinetic models are not thermodynamically consistent (i.e. energy conserving) unless the parameters satisfy certain detailed balance constraints. Specifically, detailed balance constraints are required to ensure that biochemical loops have zero flux (i.e. dissipate no energy) at equilibrium. These detailed balance constraints become increasingly difficult to derive as biochemical networks become larger. Because bond graph models assign a chemical potential to each species, they automatically adhere to detailed balance constraints. Hence, parameters can be modified without violating thermodynamic consistency [27]. This ensures that model composition respects the constraints on thermodynamics for biochemical systems.”

 Explain to a general audience how energy-based composition automatically produces thermodynamically consistent models: specifically that reframing kinetic parameters using energies of formation of species leads to conservation laws preserved around loops.

 We have addressed this comment in Issue 1b.

 Bond graph model simulation is not mentioned at all. However, results from the simulation are shown. How is bond graph model simulation different from or related to known methods like ODE integration and Gillespie SSA? This is something a general modeling audience will be completely unaware of.

The Bond graph formulation produces a set of ODEs which can be solved using any standard solver package. We have added the following text to make this clear:

(529-533) “Bond graph models of biochemical systems are deterministic and generate a set of ODEs which can be solved by any standard ODE solver package. In this paper, the models were simulated using the SUNDIALS package [62]. In future work, there is scope to expand the energy-based approach to model of stochastic systems, using algorithms such as the Gillespie algorithm for simulation. ”

 For semantics-based composition, the methods section has too many special terms that are not fully defined. Sufficient background needs to be provided on annotations and ontologies and a simple example must be used to demonstrate how merging occurs. 

While we believe the term ‘annotations’ to be readily understood by the reader, we have added text to describe what semantic annotation is. 

(80-82) “Semantic annotation is labeling the mathematical content of models or data with standard machine-readable descriptions [28]. These are crucial for the reusability and interoperability of models. ”

We have defined the term “ontology” in the following text.

(391-408) An ontology is a semantic resource of standard notions and vocabularies of species, structures, and observations in terms of Resource Description Framework (RDF) triples (https://www.w3.org/RDF/). RDF is a standard mechanism to describe and interchange data on the Web and an RDF triple is a subject–predicate–object statement that describes the properties of an entity, often using ontologies [41, 42]. For example, the RDF [OPB00340 - CHEBI29103 - FMA70022] reads [concentration-potassium-extracellular space] which specifically describes the physical property and location of an entity. Ontologies are useful tools to add meaning to different parts of models to avoid any ambiguous interpretations [43]. Depending on the area of biomedical science in which the researchers annotate their models, one or various reference ontologies might be used. For the scope of this publication, we used the csv files for the Ontology of Physics for Biology (OPB) [44] and the Gene Ontology (GO) [45], downloaded from the following links: 

– OPB: https://bioportal.bioontology.org/ontologies/OPB;

– GO: https://bioportal.bioontology.org/ontologies/GO. 

The OPB is a reference ontology for physical principles such as chemical concentration, electrical capacitance, temperature, and fluid volume. The GO provides descriptions for molecular biology such as gene products, biological sequences, and molecular activities.

We have added an illustrative example of merging two simple models to help a reader understand the concept.

(204-209) “Fig 2 illustrates an example of composing together two reactions in bond graphs. Our framework recognises that the Ce : C component is the same in both reactions and merges them. When two components from two modules are merged, the conservation equations at their corresponding `0 : u' junction changes. S1 Text.C details the conservation laws and constitutive equations in each reaction separately as well as in the case where both reactions are combined to create the composition.”

 For the composition section, use a small bulleted list to convey the elements of the pipeline. Then explain each element in detail.

 We have added text to describe the key elements of our semantics-based model composition pipeline and modified Fig 7 accordingly.

 (464-496)

 “Fig 7 depicts the eight main steps in our semantics-based model composition framework as follows: 

 1. A function in our framework extracts the annotations and values of the CellML models. If exact matches of annotations are not detected between the models, a warning is given. The user should check the models to see if they are appropriate for composition. If there are matching annotations, two pathways are made available: composition process and value allocation. 

 2. In this step, a function checks the mergeability of the identically annotated entities. If they are not mergeable, the function ignores the entities [48]. Otherwise, it passes them to the next step. For example, biochemical species are considered mergeable since they can simultaneously participate in multiple reactions but a parameter like temperature cannot be merged as it cannot become a port for external connections. Based on the deleted duplicate components, the connectivity matrices are combined, allowing the models to be merged (details in Section 2.4). 

 3. In this step, only one entity is kept from each group of identically annotated mergeable entities and the rest is deleted. 

 4. In this step, our framework links the modules at each merging point to integrate them. A link is a bond in bond graph terminology and can be added to the system by inserting a 1 in the connectivity matrix (used in the current approach) or adding a syntax to incorporate a new bond between the modules. 

 5. Step 5 identifies inconsistencies in the values of identically annotated entities. These values include the initial conditions and the entities’ thermodynamic constants (as described in Section 2.1). 

 6. This step prompts the user to choose a value for the identically annotated entities found in step 5. For instance, if a chemical species is present in more than one model (identically annotated in all the models) and has different initial concentrations, the user is asked to select one of the values or insert a new one for that specific chemical species. 

 7. This step parameterises the bond graph symbolic templates with the values for each annotated entity.

 8. Step 8 gathers the information coming from the composition process and value allocation to generate a bond graph composed model in the form of a system of Ordinary Differential Equations (ODEs).”

Issue 3: Problems with Results

 The EGFR-Ras-MAPK example shown is small enough to be in a tutorial, but it is not sufficient to be a full demonstration of the proof of concept. At the very least, attempt must be made to merge multiple (> 2) models hierarchically.

 We have addressed this comment in Key change 1.

 I’m not very happy with the decision to merge Sos species with Ras. It could’ve been easily avoided by using a third model with two simple Sos-Ras reactions. 

We have added a third module, as discussed in Key change 1.

In fact, this highlights a potential problem with the hierarchical composition approach: what happens when you start to merge two models, but then you identify missing elements missing that require additional modeling? I’m assuming this comes under post-composition adjustments.

Yes, this type of issue is a post-composition adjustment. We argue that this arises from gaps in the modelling literature and our hierarchical composition approach provides a solution to this problem. We have added the following text to highlight this issue:

 (751-757) “In general, systems biology models will frequently omit metabolites such as ATP or H+ from their reactions, causing issues for mass and energy conservation. In cases where the selected reference models do not describe this part of the biology, users can apply their knowledge or search the literature to add any missing steps or subsystems to the composed bond graph model manually. To enhance this procedure in future, we could employ genome-scale metabolic models (GSMMs) as scaffolds to identify the missing entities or reactions [71, 72].”

 The interplay between manual selection (e.g., indicating Sos as Ras) vs automated semantic inference (e.g., inferring Sos to be a specific biochemical entity) should be clearly delineated and its effects discussed.

We have added an intermediate module, so this is no longer an issue (see Key change 1).

 The figures for the results are poorly structured. Whenever figure panels are being compared in the text, they should be juxtaposed in the same figure. For example, it would be useful to place MAPK cascade simulation results and MAPK bond graph simulation results in the same panel for direct comparison and verification.

 We do not compare a mathematical model of the MAPK cascade with its bond graph equivalent, since a bond graph model was previously developed. Fig 13 shows the steady-state responses of kinases within the MAPK module alone. The same plot is later reproduced in Fig 14.B to use as a measure of verification for the curves in Fig 14.A 

 In the section examining effect of ATP concentrations, the inputs provided are not mentioned (e.g., what Ras concentration is used for each curve in Fig 15). In fact, comparing the curves at a single parameter point is not sufficient to make a general statement.

We have added the following text to describe the concentrations used in the simulations.

 (689-690) “The initial concentration of all other species was not changed. Here, the initial concentration of RShGS and Ras (common species between the modules) was 0.”

 A critical shortcoming of the manuscript is that it does not even examine the composed model in detail. The goal of model composition is to enable the pieces of one model to influence the effects of another model. In this case, the goal of merging EGFR model with MAPK is to examine the effect of EGF concentrations on MAPK. What new types of analysis are now possible on your merged model that you couldn’t do with the unmerged models? What predictions does it do that confirm or contradict existing experiments or predictions from the many EGFR-MAPK models in the literature?

While the focus of this manuscript is not to make predictions to confirm or contradict existing experiments, we have demonstrated that our approach can merge together models in order to make predictions that were not otherwise possible.

We have added new results on the effects of EGF on downstream molecules in the MAPK pathway.

(691-706)

 EGF concentration:

We examined the composed bond graph model of the EGFR-Ras-MAPK pathway to analyse and compare its functionality with other similar mathematical models. To do this, we investigated the effect of EGF concentration on MKPP. EGF initiates the EGFR pathway model and MKPP is the last terminal kinase of the MAPK cascade model. Fig 18 illustrates the behaviour of MKPP against various initial concentrations of EGF. Lower concentrations of EGF impose a delay in MKPP to reach its steady-state concentration which emphasises the role of EGF on the downstream species to the end of the MAPK cascade. Note that EGF = 0 nM does not terminate the functionality of the composed model considering that ATP hydrolysis and other intermediate species (such as RasGTP and RasGDP) fuel the subsequent steps and stimulate Ras. The time delay was also studied by Jurado et al. in [70], where lowering the EGF concentration triggered a delay in the MKPP response. Due to the different configuration of the constitutive models and the absence of EGF regulation by MKPP, the MKPP concentration in our composed model plateaus instead of descending as illustrated in [70].

We have also emphasized that our bond graph model now allows one to examine the effects of energy availability on the integrated system. This was not possible with existing models because ATP and related species were not included.

(744-750) Our bond graph composed model allowed us to investigate the role of EGF concentration and ATP on kinases while this was not possible on the individual models before composing them. While ATP hydrolysis is mentioned in the Kholodenko et al. EGFR model, it is not included in their computational model. Our composed bond graph model accounts for the missing energy sources that firstly provides a more biologically realistic model and secondly, enables us to examine hypotheses on ATP shortage in the EGFR-Ras-MAPK pathway. 

Suggestions on related work

The innovation of this work is in the application of bond graphs to reaction composition in biochemical systems. However, it is being increasingly considered that the species complexity of biochemical cells will limit our ability to compose large models from small ones due to inconsistencies in species semantics across models (touched upon in this manuscript). The authors are encouraged to check out rule-based modeling, where species semantics are formally embedded in graph structures and the energy-based extension of rule-based modeling, which largely applies the same thermodynamic principles used in this manuscript and produces consistent models that obey detailed balance. References:

Rule-based modeling:

• Chylek et al. Physical Biology 2015

• Harris et al. Bioinformatics 2016

• Boutillier et al. Bioinformatics 2018

Energy-based rule-based modeling

• Ollivier et al PLoS Comp Bio 2010

• Sekar et al IEEE BIBM 2016

• Justin Hogg Ph.D. dissertation Chapter 2, University of Pittsburgh, 2013

• Thermodynamic Graph Rewriting, Danos et al. arxiv 2015

We have cited the suggested related works in the Discussion.

(834-844) Another widely-used formalism in computational biology is rule-based modelling, in which a series of rules describe the mechanistic details of biochemical processes, for example the random binding of multiple ligands to a receptor [29]. Recently, rule-based approaches have incorporated energetic parameters to ensure thermodynamic consistency [30–32]. Danos et al. showed that by computing the free energy of species formation and hence, free energy inequalities in reactions in rule-based models, one can verify whether a model satisfies the free energy constraints and detailed balance [33]. Moreover, rule-based languages such as BioNetGen [34] and Kappa [35] allow annotations. One advantage of bond graph modelling over rule-based modelling is that they can model multi-physical systems such as electrohpysiology, whereas rule-based approaches are limited to the biochemical domain.

Issue 4: Problems with Readability

Part of scientific communication is to emphasize clarity and directness. As it stands, the text is too verbose and unstructured and is not fully copy-edited. Some suggestions to make it readable:

 Using passive voice unnecessarily makes sentences long and complicated. E.g., Instead of saying “modifying a network is easily performed by…”, you can say “to modify a network, one can…” It also makes things difficult to understand as to whether it was done automatically or manually, particularly in several places in the methods section.

Following the reviewer’s suggestions, we have made modifications to the text to improve the readability of the manuscript.

(436-437) To modify a network, one can insert 0 or 1 in the matrix or delete its corresponding ...

(360) We aim to minimise manual input through automation in model composition ...

(452-453) ...our framework will link it to its corresponding bond graph symbolic template.

(453-454) Thereafter, a function in our framework finds similar annotations in the models ....

(457) Ultimately, our framework produces the final model...

(335) Hence, we created a symbolic bond graph module ...

(517-518) Our framework integrates the modified connectivity matrices ...

(521-522) ... our framework inserts an additional 1 in the matrix.

 Some words are overused and do not convey any meaning to the reader. For example, I fail to understand what is “generic” about the composition pipeline.

We have modified the text to make this clearer.

(140-142) “This is a generic model composition approach since the idea is domain-independent and could in principle be applied to models in different physical domains (e.g. electrical or mechanical).”

 Some sentences are unnecessarily long without providing any additional meaning. E.g., instead of saying “we employed the idea of having symbolic bond graph templates”, you can simply say “we built symbolic bond graph templates”.

 We have modified the text to address this issue.

 (426) “... we built symbolic bond graph templates...”

 (554-556) “We used our method to merge the modules within the MAPK cascade and between the pairs (EGFR pathway, Ras activation) and (Ras activation, MAPK). This yielded the bond graph configuration of the EGFR-Ras-MAPK signalling pathway.”

 (433) “...we employed the concept of a connectivity matrix. � ...we used connectivity matrices.”

 Figure captions need to provide sufficient information so that they can be read in isolation. This means the caption should briefly summarize how the figure is referenced in the paper. E.g. Fig 15 caption does not even mention which model is used.

 We have modified the caption text to provide more information.

 “Fig 13. The steady-state responses of the activated kinases for different input amounts in the MAPK cascade model.”

 “Fig 14. Verification of the responses of activated kinases to Ras in the composed EGFR-Ras-MAPK bond graph model by comparing with the predicted steady-state responses in the MAPK cascade module. (A) Ultrasensitivity in the composed EGFR-Ras-MAPK bond graph model. The steady-state concentrations of the kinases are: MKKKP = 1.37 nM, MKKPP = 1054.37 nM, MKPP = 987.96 nM; (B) Predicted steady-state concentration of the kinases. The purple dashed line shows the concentration of Ras at t = 100 (s) in the composed EGFR-Ras-MAPK bond graph model. The predicted steady-state concentrations of MKKKP, MKKPP, and MKPP at Ras = 0.311 nM match with the ones in the composed EGFR-Ras-MAPK bond graph model.”

 “Fig 15. Activation of terminal kinases with and without negative feedback in the composed EGFR-Ras-MAPK bond graph model. (A) Without negative feedback; (B) With negative feedback.”

 “Fig 17. Effect of different levels of ATP concentration on activated kinases in the composed EGFR-Ras-MAPK bond graph model. (A) MKPP; (B) MKKPP; (C) MKKKP; (D) Steady-state concentration of MKKKP, MKKPP, and MKPP against relative ATP concentration. MKKKP concentration is also separately shown in a box due to its relatively small amounts compared to MKKPP and MKPP (initial concentration of common species: Ras=0, RShGS=0).”

 The text in figures is extremely tiny relative to the size of the figure and unreadable. Effort should be made so that the figure looks good printed on paper.

We have increased the text size for Fig 8, Fig 9, Fig 14, Fig 15, and Fig 17D.

 Many paragraphs begin with extra-long sentences that run on. E.g. lines 48-50 packs too many different concepts into a single sentence. This is unnecessary and can be broken down.

We have modified the text to make this clearer.

 (101-103) “Here, as an extension to our previous work, we have incorporated annotations to bond graphs in a new platform. This platform allows us to automatically construct a composed model from annotated CellML files treated as modules.”

 (264-266) “... we first removed the thermodynamically infeasible irreversible reactions from the network (for their different parameter definitions). Then, we applied the parameter balancing method described in....”

 (554-556) “We used our method to merge the modules within the MAPK cascade and between the pairs (EGFR pathway, Ras activation) and (Ras activation, MAPK). This yielded the bond graph configuration of the EGFR-Ras-MAPK signalling pathway.”

 Sections should begin with a brief paragraph summarizing the section. Each paragraph should have a first sentence summarizing the paragraph.

We have added summary paragraphs to several sections throughout the paper. Sections:

Automated model composition pipeline,

 The prerequisites, 

The generic approach,

Modules for EGFR-Ras-MAPK signalling: Bond graph models of the pathways,

The EGFR pathway module,

The Ras activation intermediate module,

The MAPK cascade module,

Verification of bond graph modules.

 In many places, special terms are used before being defined, which is poor form. For example, physical feasibility in line 20 is defined only in line 26. Similarly, symbolic models in line 96 is used first and then explained. Semantics-based in line 40 has no explanation. Using “… will be explained later” is also poor form and shows lack of narrative.

 We have modified the text to make this clearer.

 Physical feasibility is now defined:

 (29-31) “The resultant model often needs further adjustments to be executable and yet it might not follow the laws of thermodynamics and physics (such as energy, mass, and charge conservation) [12, 14]. This is referred to as physical feasibility.”

We have added a definition for symbolic modules below:

 (361-365) “In this endeavour, we have provided some exemplar predefined bond graph models in which the parameters do not have any values. We call these predefined bond graph models as symbolic modules. Symbolic modules allow us to determine the parameters' values later where the annotated parameters from the CellML models would accordingly link.”

We have now also defined semantic annotation.

 (80-82) “Semantic annotation is labeling the mathematical content of models or data with standard machine-readable descriptions [28]. These are crucial for the reusability and interoperability of models. ”

 

Reviewer #2: # A semantics, energy-based approach to automate biomodel composition 

## Major comments 

This paper describes a methdology of using Bond graphs to create complex composite models. 

This is a topic many people (myself included) will want to learn more about, and the accompanying software could potentially be very useful. However, there are several issues with the manuscript text that will need addressing before this can be published.

Most notably: 

 The introduction and methods section are far more abstract than they needs to be. For example, the text frequently refers to "entities", "elements", "components" of bond models without making clear what these words refer to (or if they are the same), it mentions "similar annotations" without saying what this means, etc. Such parts of the text should be rewritten to be both clear and precise. 

 We have defined the words “entities”, “elements”, and “components” of bond graph models in the following text.

 (52-53) Bond graphs represent systems as graphical representations which consist of a set of elements, i.e, components and junctions.

 (53-58) Components represent physical entities (such as ions, complexes, genes, atoms in microscopic level and resistors, capacitors, dampers, and mass in macroscopic level) and are defined as general configurations of electrical, mechanical, or chemical elements. For instance, C components in bond graphs are charge storage components i.e. capacitors in electrical circuits, springs in mechanical systems, or chemical species in chemical reactions.

 We have changed the term “similar annotations” to “identical annotations” to make it specific for the approach we use.

2. The problem statement is unclear. In several places it seems to be about coupling two (known) models, but parts of the methodology imply it is about identifying models that could potentially be coupled? 

 The focus of this paper is purely about coupling together models in systems biology automatically in an energy-based manner. Through this aim, only a part of our work is looking for models to be merged; this was required because bond graph models are not available for many biological systems. 

We have stated the challenges in hierarchical model composition and how bond graphs (as an energy-based modelling approach) can address these challenges. Also, to automate this process, we are using semantic annotations. We have added the following text to clearly state the problem.

 (7-9) An approach to automatically and hierarchically construct sophisticated models of biology that ultimately leads to the generation of biologically and physically correct models is currently missing.

 Moreover, we have stated the bottlenecks in our previous work and our current approach to address them in the following text.

 (89-103) An automated model composition approach significantly assists researchers in creating large-scale models from existing modules [36]. Shahidi et al. [37] introduced a general hierarchical model composition method by encoding bond graph modules in CellML and constructing a complex model using the SemGen merger tool [38]. The SemGen merger tool uses the biological semantics of the components in models to identify and interpret them unambiguously. Although this method facilitated the integration of annotated bond graph models, bottlenecks might arise when a modification in the CellML bond graph modules is needed (modellers must know the bond graph conservation laws). Moreover, it required adding auxiliary variables as ports to each module and connecting them manually using the semi-automated SemGen merger tool. While annotations are readily incorporated into bond graphs, using annotations in model composition has not been conducted in this context. 

 Here, as an extension to our previous work, we have incorporated annotations to bond graphs in a new platform. This platform allows us to automatically construct a composed model from annotated CellML files treated as modules.

 As the problem statement is an abstract one, an example (perhaps a simple toy problem in addition to the real-life example shown) should be introduced early on (i.e. in the introduction) and used to explain the problem statement. In the current manuscript a general outline of a solution is being sketched long before the reader has been given the tools to understand the problem this aims to solve. 

 We have addressed this issue by giving an example in Fig. 2. We have also moved the bond graph introduction in Section 2.1 to Introduction to give the reader the essential tools to understand the problem.

 The level of detail varies considerably throughout the paper, lots of words are devoted to fairly simple processes such as creating a connectivity matrix and removing components that have been deemed identical, but very little is said about more complex steps, e.g. how annotations are compared, how "thermodynamically consistent parameters" are created from inconsistent ones etc. 

 We have dealt with the issue of connectivity matrices in Key change 2.

 We have also shrunk the extensive explanation on removing identical components and mainly discussed about it in Steps 3, 4 of the framework description as follows.

 (478-483) 

 3. In this step, only one entity is kept from each group of identically annotated mergeable entities and the rest is deleted. 

 4. In this step, our framework links the modules at each merging point to integrate them. A link is a bond in bond graph terminology and can be added to the system by inserting a 1 in the connectivity matrix (used in the current approach) or adding a syntax to incorporate a new bond between the modules. 

 We have stated that our framework merges the components that have identical annotations (lines 454, 471, 478, 484, 487, 489, 511, 777).

 We have explained the equations in the following text and referred the reader to an example for further reading.

 (266-271) ... we applied the optimisation method described in [49] to the remaining reversible reactions. In brief, by taking logarithms on the constraints of each reaction (k^+ = κ∏_i▒K_(r_i ) and k^- = κ∏_j▒K_(p_j ) ), the relationship between the kinetic and bond graph parameters can be expressed as a linear matrix. The reader is referred to an example on the generation of the linear matrix of thermodynamic constants in Appendix B for [49]. 

## Detailed comments ### Line 17

 The main challenges in hierarchical model composition include: (a) incompatible code languages, (b) different modelling frameworks, (c) post-composition adjustments, and (d) physically implausible resultant models.

This needs a slower and more careful explanation. What is the difference between an "incompatible language" and a "different framework"? What are "post-composition adjustments". Can you give an example?

 We have explained the terms "incompatible language", "different framework", and "post-composition adjustments" in the following text with examples.

 (19-25) The main challenges in hierarchical model composition include: (a) incompatible code languages (using dramatically different modelling languages such as Object-oriented, graphical, and continuous/discrete time), (b) different modelling formalisms (such as using rule-based modelling, differential equations, neural networks, and Boolean networks), (c) post-composition adjustments (manual edits in mathematical equations, rules, or parameters to make the composed models biologically sensible [8])

### Line 20

 A majority of model integration platforms ... require compatibility between the languages and modelling frameworks

 In contrast, the resultant model still needs further post-merging code-wise adjustments to be executable, yet it might not represent a physically feasible model.

How are these two statements "In contrast"?

 We have modified the text to make it clearer.

 (26-31) “While existing model integration platforms, such as the SBML Hierarchical package [9] and PySB [10] can resolve issues with code and modelling formalism compatibility, they are limited to the biochemical domain. The resultant model often needs further adjustments to be executable and yet it might not follow the laws of thermodynamics and physics (such as energy, mass, and charge conservation) [12,14]” 

Are "post-merging code-wise adjustments" the same thing as "post-composition adjustments"?

 We changed “post-merging code-wise adjustments” terms into "post-composition adjustments" throughout the manuscript. 

### Line 26

 One solution to these issues is using a hierarchical modelling approach (to help with the post-composition adjustments) and an energy-based modelling framework (to guarantee a physically plausible composed model).

I'm not sure if you are (A) stating something I should already be able to understand at this point, of if you are (B) saying this will be explained in the upcoming text. If A, then it needs a lot more explanation, if B then please rewrite the text so that this is clear.

 We have added text to the introduction to explain how energy-based models help to ensure that models are compliant with the laws of physics.

 (31-40) Several formulations and frameworks have been developed to ensure biochemical models follow the laws of thermodynamics in particular ( [15-17]) but most of them are purely mathematical and are difficult to implement for model composition. Furthermore, most of the model composition tools are not applicable to multi-physics systems and cannot be generalised to more complex biological systems. One solution to these issues is combining a hierarchical modelling approach (to help with the post-composition adjustments) with an energetic and multi-physics framework that explicitly models energy to ensure adherence to the laws of physics and is executable in multi-physics modelling. The bond graph approach addresses these issues. 

### Line 53

 The annotated data from the CellML modules are then extracted and assigned to their equivalent bond graph components.

By "data" do you mean data (i.e. parameter values?), or e.g. equations, units, etc. ?

And what is a "component"? Should data not be connected to data, or variables to variables? Please rewrite to make statements such as these less abstract and more clear.

 We have replaced the word “data” with “parameters” modified the text to make it clearer.

 (106-111) The annotated parameters from the CellML models are then extracted and their values assigned to their equivalent bond graph parameters. Since the equations of a bond graph can be automatically generated from their network structure, we only need to parameterise them using the parameter values from the CellML files. Thereafter, any common biochemical, biological, or physical entities among the modules are identified and merged to render a composed model. 

 We have defined the word “component” with an example in the following text.

 (53-58) Components represent physical entities (such as ions, complexes, genes, atoms in microscopic level and resistors, capacitors, dampers, and mass in macroscopic level) and are defined as general configurations of electrical, mechanical, or chemical elements. For instance, C components in bond graphs are charge storage components i.e. capacitors in electrical circuits, springs in mechanical systems, or chemical species in chemical reactions. 

### Line 55

 We demonstrate this by an example where a bond graph model is constructed from its constitutive modules, i.e., the Epidermal Growth Factor Receptor (EGFR) signalling pathway and the Mitogen-Activated Protein Kinase (MAPK)

Given this example, I'm a bit surprised you opted for CellML over SBML. This may warrant some discussion.

 The main goal of this manuscript is to find a way for CellML model composition using bond graphs since CellML models of biochemical reactions lack some extra information that SBML models already have. Yet in broader applications, CellML can deal with models that are not purely biochemical. We illustrated a solution to overcome the shortcomings of CellML models while this could be done in a more automatic manner for SBML models as we have discussed in the following text.

 (782-791) Here, we have selected models encoded in CellML because CellML can deal with models that are note purely biochemical, but the approach can be applied to models in other formats, such as SBML, as long as they can include a semantic description of the system being modelled. While symbolic templates are required to apply our approach to CellML models, this step is not required for SBML models. This is because the reactant(s)-reaction-product(s) relationships are explicitly defined within SBML models while this information is not clearly provided in CellML models. In this paper, we aimed to illustrate a possible way to convert CellML biomodels into bond graphs and automatically compose them. In future, we intend to apply the same method on SBML models in a more automated way. 

 (814-816) The ultimate goal of applying our model composition method is to provide a foundation for future tool developments to convert any arbitrary CellML/SBML model into bond graphs and then convert it back to a CellML/SBML file. 

### Line 79 Materials and methods

Please make this whole section less abstract, e.g. by starting with a detailed example of two models that you wish to connect, so that the reader has some idea what you mean by the various "entities" and "elements" that appear in the text.

 We have added an example of connecting two reactions in bond graphs in Fig 2 and explained the equations in S1 Text.C.

 (204-209) “Fig 2 illustrates an example of composing together two reactions in bond graphs. Our framework recognises that the Ce : C component is the same in both reactions and merges them. When two components from two modules are merged, the conservation equations at their corresponding `0 : u' junction changes. S1 Text.C details the conservation laws and constitutive equations in each reaction separately as well as in the case where both reactions are combined to create the composition.” 

### Line 86

 automatically rewiring the connections between components and modules

What are "components" in this sentence? The introduction should probably give a very brief explanation of what a bond graph looks like (maybe a figure?) and explain what the difference between e.g. a bond graph "module" and a bond graph "component" is. (Is it just a variable? Then please say so)

 We have defined the term “component” in the following text.

 (53-58) Components represent physical entities (such as ions, complexes, genes, atoms in microscopic level and resistors, capacitors, dampers, and mass in macroscopic level) and are defined as general configurations of electrical, mechanical, or chemical elements. For instance, C components in bond graphs are charge storage components i.e. capacitors in electrical circuits, springs in mechanical systems, or chemical species in chemical reactions. 

 We have given a brief description of bond graphs in the Introduction section (lines 47-69).

 We have given examples of bond graph modelling in Fig 1 and Fig 2.

 We have defined the term “bond graph module” in the following text.

 (370-371) In the current work, when a symbolic bond graph template is parameterised we call it a bond graph module.

### Line 92

 Our method to expand and integrate biosimulation models provides the foundation for further developments in an open-source environment based on energy-based modules and automation to minimise manual input.

I'm not sure what this sentence means? Without the "to minimise manual input" it reads like part of an abstract or conclusion section. But are you just trying to say you want to minimise manual input when composing models?

 We have modified the text to make it clearer.

 (360-361) We aim to minimise manual input through automation in model composition while using energy-based modules in an open-source environment.

### Line 94

 In this endeavour, we have provided some exemplar symbolic bond graph models to which the annotated parameters from the CellML modules would accordingly link.

Does this mean that any time I want to link 2 models, I need to write 2 templates (that presumably match the 2 models I want to link), and then write a connection matrix describing how my two templates are linked? If it's so hard-coded, then why not just write down which variables I want to connect straight away?

 We have addressed this in Key change 2. While it is possible to define a model composition approach by writing down the variables one wants to connect, we found the connectivity matrix to be helpful in implementing the approach using software.

### Line 99

 The suitable bond graph model is then automatically selected from the list by 

Which list?

 We have modified the text to make it clearer.

 (365-367) The suitable bond graph template is then automatically selected from a list of symbolic bond graph templates....

### Line 100

 identifying specific annotated components in the CellML modules

What exactly is a "component" here? Is it a CellML component (i.e. a container of variables)?

 We have modified the text to make it clearer.

 (367-368)...by identifying specific annotated parameters (for example the species specific constants) in the CellML models.

### Line 117

 Ontologies

I'm not sure who the intended audience for this paper is, but if it's "anyone who wants to compose a big model" you should probably add a line saying what an ontology is, or maybe add a reference.

 We have explained “ontologies” in the following text.

 (391-408) An ontology is a semantic resource of standard notions and vocabularies of species, structures, and observations in terms of Resource Description Framework (RDF) triples (https://www.w3.org/RDF/). RDF is a standard mechanism to describe and interchange data on the Web and an RDF triple is a subject–predicate–object statement that describes the properties of an entity, often using ontologies [41, 42]. For example, the RDF [OPB00340 - CHEBI29103 - FMA70022] reads [concentration-potassium-extracellular space] which specifically describes the physical property and location of an entity. Ontologies are useful tools to add meaning to different parts of models to avoid any ambiguous interpretations [43]. Depending on the area of biomedical science in which the researchers annotate their models, one or various reference ontologies might be used. For the scope of this publication, we used the csv files for the Ontology of Physics for Biology (OPB) [44] and the Gene Ontology (GO) [45], downloaded from the following links: 

 – OPB: https://bioportal.bioontology.org/ontologies/OPB;

 – GO: https://bioportal.bioontology.org/ontologies/GO. 

 The OPB is a reference ontology for physical principles such as chemical concentration, electrical capacitance, temperature, and fluid volume. The GO provides descriptions for molecular biology such as gene products, biological sequences, and molecular activities.

### Line 121

 We suggest downloading CHEBI, FMA, OPB, and GO ontologies from the following links:

Please explain what domains these 5 are, and why they are appropriate (are they appropriate for everything, or just for your two pathway example?)

 We used the OPB and GO ontologies for the particular case study in this paper. Why do I need the other two then?

 We were suggesting downloading these 4 ontologies to aid model composition in other contexts. Since they were not required for this study, we have removed the requirement of downloading the other 2 ontologies in this manuscript.

Please explain what these ontologies will be used for. Do they provide labels identifying unique variables? Variable types e.g. "is a concentration"? Will we be inferring properties of our variables using these ontologies e.g. units?

 We have explained this below:

 (406-414) The OPB is a reference ontology for physical principles such as chemical concentration, electrical capacitance, temperature, and fluid volume. The GO provides descriptions for molecular biology such as gene products, biological sequences, and molecular activities. Due to the limited size of uploaded files on GitHub, the required reference ontologies for the current model composition (OPB and GO) are not provided on our GitHub repository. We stored the ontologies locally to interpret the RDFs and use the interpretations where the user needs to make a decision based on the annotations but the approach can be reduced to a framework in which the annotations are only read and compared in RDF format and the interpretations are not given to the user.

### Line 128

 The generic approach

Do you mean generic, i.e. this section explains why the approach works on any model, or "general approach", so that this section presents an outline of the approach taken in the paper?

 We have added the following text to explain the term “generic approach” in our work.

 (140-142) “This is a generic model composition approach since the idea is domain-independent and could in principle be applied to models in different physical domains (e.g. electrical or mechanical).”

### Line 131

 the parameters in the models

Please explain in the intro what you expect to find inside each model (should this say "module"?), e.g. variables, parameters, etc. and use these terms throughout instead of "elements", "entities", "components" etc.

Is a distinction made between variables and constants/parameters?

 We have defined the terms “components”, “elements”, and “entities” in Major Comment #1.

 We have described that our framework extracts from the CellML models and the reason why we do not need to extract variables from the original models.

 (106-110) The annotated parameters from the CellML models are then extracted and their values assigned to their equivalent bond graph parameters. Since the equations of a bond graph can be automatically generated from their network structure, we only need to parameterise them using the parameter values from the CellML files. 

### Line 138

 The number of rows and columns each equals the number of elements in the network in total

What are the "elements" and the "network"? Are elements modules? Variables in a module?

 We have defined the “bond graph elements” in Major Comment #1. We have also rephrased the following sentence to make it clearer. 

 (429-430) Here, the number of rows and columns each equals the number of bond graph elements of a system [46].

Figure 1 and the accompanying text are far too vague and focus on the wrong aspects: Readers who are interested in this topic will have come across the topic of a network and a connectivity matrix before, and will be wondering instead how it applies to your problem of tying two pathway models together. This second bit is not explained in any detail here.

If they are variables, then what are we expecting from the on 

 We have addressed this comment in Key change 2.

 As explained in the answer to the comment for ###Line 138, the rows and columns of a connectivity matrix do not represent the variables but the bond graph elements.

### Line 142

 "facilitates computational measurements"

What is a "computational measurement" and how is it facilitated by a binary representation of these connections?

 We have removed this term from the sentence to avoid any confusion because although reducing the computational cost is one of the advantages of using connectivity matrices, it is not particularly applied in our bond graph model composition. We have modified the following text.

 (433-436) While not essential to our methodology, we chose this approach because binary representation of models clearly shows the connections and gives the minimal required details to define a network which can be exported to other tools and software for further analysis [46]. 

### Line 144

 Modifying a network is easily performed by inserting 0 or 1 in the matrix

In two places, to preserve symmetry? Given the zero diagonal, the symmetry (which the user needs to manually maintain), and the presumed sparseness of this matrix in real applications (if elements are variables, as I suspect at this point in my reading), this is not a very compact or easy-to-use representation.

 We have answered to this comment in Key change 2.

### Line 152

 In a `black box' composition approach, the elements of the modules are not accessible and only the input/output variables can be used for coupling

It's not clear to me why we'd expect the modules to have clearly defined "inputs" and "outputs" at this point, so I'm struggling a bit to see why you need to point out that any variables can be connected.

 To make the sentence clearer, we have explained why we need a white box approach in the context of biophysiology instead of the conventional black box approach in engineering.

 (439-442) To identify the merging points between the modules we used a ‘white box’ approach. In this approach all or a group of the bond graph components in the modules are mergeable. In a ‘black box’ composition approach in contrast, only the components predefined as inputs or outputs are accessible [37, 47].

### Line 154

 almost all the entities can be regarded as merging ports

What is a "merging port"? And do we need this bit of jargon or can it be stated more simply?

 We have modified the sentence to make it clearer.

 (442-443) In coupling biological models, all entities are mergeable,...

### Figure 2

What are "similar annotations"? Do both models have the exact same set of RDF properties? The exact same unique labels? Is it OK if only a subset matches?

 We have replaced the term “similar annotations” with the term “identical annotations” to emphasise that a subset matching in annotations is not enough.

This figure seems to show 2 preselected models, but there was also mention of a "list", where does that fit in?

 The list refers to the list of stored bond graph templates mentioned in the answer to the comment for ### Line 99.

 (365-367) The suitable bond graph template is then automatically selected from a list of symbolic bond graph templates.... 

 In this particular study, we populated the list by either using existing bond graph models or generating them from existing systems biology models

How is the "repository" section related to the two models?

 In this paper, the bond graph models were stored locally. Accordingly, we have replaced the term “repository” with the term “stored files” to be distinguishable from online repositories such as PMR.

 We have explained how the stored files relate to the two models in two steps of the flowchart. The ontologies are used in step 1 and the bond graph modules & connectivity matrices are used in step 4 in the following text.

 (466-470) 1. A function in our framework extracts the annotations and values of the CellML models. If exact matches of annotations are not detected between the models, a warning is given. The user should check the models to see if they are appropriate for composition. If there are matching annotations, two pathways are made available: composition process and value allocation. 

 (480-483) 4. In this step, our framework links the modules at each merging point to integrate them. A link is a bond in bond graph terminology and can be added to the system by inserting a 1 in the connectivity matrix (used in the current approach) or adding a syntax to incorporate a new bond between the modules. 

### Sections 2.2 & 2.3

These are very helpful, but are required reading to understand e.g. 2.1, so some re-structuring is necessary!

 We have moved the introductory text to bond graphs and its principles to the Introduction section (lines 47-69). We have also re-structured the manuscript and moved Sections 2.2 & 2.3 before 2.1.

### Line 216

 Is the "." in the equation meant to be a \\cdot? Also, if a multiplication symbol is used here it should be present in the other two multiplications (R*T*ln(K_q*q) or RT ln(K_q q))

 We have edited the equations so that dots are consistently used for multiplication.

 What is u_q in this equation?

 We have replaced “u_q” with “u” which is defined in the following text.

 (165) The chemical potential is u (J mol−1), stored within the biochemical species, ...

### Line 222

 A reaction represents a dissipative process, which in the case of mass-action kinetics... This could do with some clarification. Are all reactions "dissipative"? Mass-action kinetics are usually phrased in terms of reversible processes.

 We have explained a dissipative process in bond graphs and modified the text to be clearer on mass-action kinetics.

(178-181) In bond graphs, a reaction represents a dissipative process where chemical energy is lost in the form of heat [52]. In the case of reversible mass action kinetics, a reaction is defined in bond graphs by an Re component with the constitutive relation v = κ(e^(u_r/RT) -e^(up/RT)) (Marcelin–de Donder equation), 

### Line 250

 and the bond graph model of MAPK cascade is taken from the work by Pan et al. [21]. In this paper, the bond graph representation of the reference MAPK cascade was available.

Here, we detail how bond graph models of these systems were constructed. The second sentence repeats the first. But then the third contradicts it?

 We have modified the sentence to make it clearer.

 (229-231) The bond graph model of the MAPK cascade is adopted from the work by Pan et al. [11]. Here, we detail how the bond graph modules of these systems were constructed.

 The bond graph model of the MAPK cascade was developed by Pan et al. using the black box approach but we have used a white box approach and annotated the cycles to automate the model composition. We have explained this in the following text.

 (337-339) We used the `white box' approach rather than the `black box' approach in Pan et al.'s work. We also annotated each cycle separately to automate the model composition.

### Figure 4

 The network adapted from Missing is/was

 Fixed.

### Line 294

 We applied curve fitting to estimate the reaction rate constants for the irreversible steps (κ4

, κ8 , & κ16 ). We obtained the time-dependent behaviour of the contributing species in steps 4, 8, and 16 (required for curve fitting).

 If the second sentence is required for the first, it should come first.

 We have modified the text to make it clearer.

 (279-282) We obtained the time-dependent behaviour of the contributing species in steps 4, 8, and 16 from the reference CellML model for the EGFR pathway and applied curve fitting to estimate the reaction rate constants for the irreversible steps (κ4, κ8, & κ16).

 These two steps contain a lot of work. Much more explanation is needed to make this reproducible.

 We have explained these two steps with an example in S1 Text.B and added the following text.

 (282-283) As an example, this procedure is shown in S1 Text.B for step 4.

### Line 299

 some of the reversible reactions do not satisfy detailed balance

I had assumed this would be guaranteed by the bond graph methodology, some comment or a reference could be useful here.

 We were referring to the reversible reactions in the original model, where thermodynamic consistency is not guaranteed. We have modified the sentence to make it clearer and explained it further in the following text. 

 (285-287) ..., some of the reversible reactions in the original model do not satisfy detailed balance. Since the bond graph parameters are inferred from the original model, an approximation with the least square error is made to generate the closest fit to the data while adhering to detailed balance constraints.

### Line 300

S1 Table. S2 Table and S3 Table  Table S1. Table S2 and Table S3

 According to the PLOS ONE journal, the supplementary materials must be named and referred to as S1 Table, S2 Table,... 

 Reference: https://journals.plos.org/plosone/s/supporting-information

### Lines 339, 343, 344, 345

 model of EGFR-Ras-MAPK

 and MAKP cascase

 Since MAPK cascade includes

Many missing "the"s throughout the text

 We have added the missing “the”s.

### Lines 350

 Due to the limited size ...

This would make more sense in section 2.1.1

 We have moved the text to section 2.3.1 (lines 409-411).

### Line 354

 for inconsistencies among the values of similarly annotated components and parameters. What is meant by "inconsistencies" here? Please be precise and give examples.

 We have replaced the term “inconsistencies” with “mismatches” to make the text clearer. We have explained it in the flowchart steps with a general example. 

 (487-491) This step prompts the user to choose a value for the identically annotated entities found in step 5. For instance, if a chemical species is present in more than one model (identically annotated in all the models) and has different initial concentrations, the user is asked to select one of the values or insert a new one for that specific chemical species.

 We have also explained a case-specific example in value mismatches in the Discussion section (following text).

 (735-743) Merging components across models might raise mismatches in their parameters. Here, RShGS in the EGFR and Ras activation models, and Ras in the Ras activation and MAPK models were merged. These species have different initial values and/or thermodynamic constants in their corresponding models. In such cases, our framework flags different values for same species. This is solved by asking the user to either select one of the values or insert a new value for the flagged parameter. Since the user may not have the relevant expertise, we aim to provide users with an evaluation of the ambiguous parameter in multiple models available on PMR in the future. This will give the user a better awareness of the range of values for uncertain parameters. 

### Line 510

 For biochemical reactions, if the parameters are thermodynamically inconsistent, they are converted into bond graph compatible ones.

Where is this process explained?

 We have rephrased the text to make it clearer.

 (712-714) Thereafter, all the biochemical reactions (reversible or irreversible) in the reference models are converted into bond graph compatible ones (Section 2.2.1).

### All figures

This could be just a proof issue, but the figures are all rasterised, and at a low resolution.

 The separately available figure files (tiff) are all of high resolution.

Reviewer #3: 

In their manuscript “A semantics, energy-based approach to automate biomodel composition”, Shahidi et al. describe a new framework for combining biochemical network models based on a mathematical representation in the form of bond graphs. They describe the method and how it is designed to guarantee thermodynamic correctness of the resulting models, and illustrate the procedure with an example case, combining two existing signaling pathway models into a larger, consistent model. 

The manuscript is extremely well and clearly written and was a pleasure to read. I think that the method will be very useful. Since it has already been implemented for CellML models, and an implementation for SBML models is conceivable, it has the potential for broad applications in biochemical pathway and network modeling. 

I did not check the code. 

I have no substantial criticisms. Below I list a few minor points to improve the manuscript, mostly about clarification of words. I leave it to the authors to decide which of these points they would like to account for. 

Finally, I would like to express my condolences to the authors for the passing of Professor Crampin. It must have been painful for you to complete the work without your colleague.

Title: The term “bond graph” could be mentioned in the paper title.

 We decided not to include the term bond graph in the title since we wanted to emphasise the utility of incorporating energy into models of biochemistry rather than focusing on the specific methodology.

2: “Physicians”: I think it’s a dream of modelers that their models will be used by physicians, but I think we’re usually still far from this.

 We have removed Physicians from the sentence.

27 (and elsewhere): “Energy-based modeling framework”: since “energy-based” can mean many things, it would be good to explain this term very clearly and explicitly early on.

 We have modified the following text to be clearer.

 (38-40) .... an energetic and multi-physics framework that explicitly models energy to ensure adherence to the laws of physics and is executable in multi-physics modelling.

60: “provides a reliable and consistent framework that first conserves energy” again, not very clear. The need to satisfy thermodynamic Wegscheider conditions does not exactly arise from energy conservation (first law of thermodynamics), but is also related to the second law, and basically the fact that Gibbs free energy is a thermodynamic potential. So “conserves energy” is a bit imprecise, and maybe not very well understandable.

 We have modified the following text to be clearer.

 (116-118) ... provides a reliable and consistent framework that first tracks energy transfer; secondly ensures that reactions can only operate in the direction of decreasing chemical potential; ...

101: “Due to the hierarchical feature of bond graphs,” also, “hierarchical” is not very clearly explained. I guess it refers to the usage of symbolic templates and (“inside” them) the actual network-like model structures. But these are just two layers, and “hierarchical” sounds like there were many hierarchy layers.

 Although there are two layers of hierarchy in the current model composition, we also benefit from another property of hierarchical composition which is model integration, in general (following text).

 (146-147) Bond graphs allow more than two levels of hierarchy which supports model integration. 

109: “A common effort between the components is shown by a ‘0’ junction, while a ‘1’ junction shows a common flow, and the energy is conserved and travels between components bidirectionally through bonds (shown by harpoons).” This explanation is not very easy to get, please explain in more detail (e.g. mentioning little examples?)

 We have explained this with an example in the following text. We have given the equations in S1 Tect.C instead of the main text to not divert much from the flow of the manuscript.

 (204-209) “Fig 2 illustrates an example of composing together two reactions in bond graphs. Our framework recognises that the Ce : C component is the same in both reactions and merges them. When two components from two modules are merged, the conservation equations at their corresponding `0 : u' junction changes. S1 Text.C details the conservation laws and constitutive equations in each reaction separately as well as in the case where both reactions are combined to create the composition.” 

147: “Notice that the connectivity matrix is symmetric” at first, not clear if the connectivity matrix is always symmetric (by definition) or just happens to be symmetric in this example application.

 We have explained this in the following text.

 (430-431) Connectivity matrices are symmetric for undirected (bidirectional) networks and asymmetric for directed networks.

Section 2.2 explains the usage of bond graph modeling of biochemical reactions, but it remains unclear how parameters, rate law formulae, and other data attached to the nodes will be treated during model composition. Is an enzymatic rate law a property attached to the reaction node? 

 We believe this has been addressed in the comment on line 109.

What if in model combination, the same reaction is described (in the two models) in different ways, e.g. with or without an allosteric inhibitor? Is this just a choice between data attached to the reaction node, or a choice between different structures of the bond graph?

 This type of issue will be dealt with under post-composition adjustments. We argue that this type of issue arises from different applications of certain models and the scope of the work. Our hierarchical composition approach provides a solution to this problem by providing the option of readily removing or adding components to the composed model. We have added the following text to highlight this issue.

 (758-762) There are situations where different representations of a certain reaction or process are available through the literature. For example, a reaction might be described with or without an allosteric inhibitor. This arises from different applications for different versions of a model and the scope of the studies. In such cases, one has to decide which version of the model they want to use in model composition.

247: “As such, we consider Ras protein to be the mutual species in both pathways.” Would there be additional complications if models are connected by several species (e.g. closing thermodynamic loops that were not present in the initial models, but in the combined model”?

 This is a problem in traditional kinetic models as one would never know that they have closed thermodynamic loops. Whereas in bond graphs, it would be impossible to generate such infeasible models. The modeller would be forced to make changes to one or more of the initial models, which was the case in this study.

 (70-79) “In the context of biochemistry, modellers widely use traditional kinetic models. However, in general, kinetic models are not thermodynamically consistent (i.e. energy conserving) unless the parameters satisfy certain detailed balance constraints. Specifically, detailed balance constraints are required to ensure that biochemical loops have zero flux (i.e. dissipate no energy) at equilibrium. These detailed balance constraints become increasingly difficult to derive as biochemical networks become larger. Because bond graph models assign a chemical potential to each species, they automatically adhere to detailed balance constraints. Hence, parameters can be modified without violating thermodynamic consistency [27]. This ensures that model composition respects the constraints on thermodynamics for biochemical systems.”

In the bond graph model, saturable rate laws were described by irreversible Michaelis- Menten kinetics. Would it also be possible to use reversible Michaelis-Menten kinetics? Or do reversible reactions have to be modelled by mass-action kinetics, for mathematical reasons? (I guess the answer to the latter question is no; maybe it would be good to point this out?)

 We have added the following text to explain the possibility of using reversible Michaelis- Menten kinetics and how we can represent it in Bond GraphTools.

 (200-203) Reversible Michaelis-Menten kinetics can also be represented using bond graphs [22]. However, because the default Re components in BondGraphTools follow the mass actions kinetics, we have chosen to approximate Michaelis-Menten kinetics using elementary mass action reactions (see [12]).

“The chemostats” - I guess the word refers to species with fixed and given concentrations (sometimes called “external metabolites” in kinetic modelling)? Since the same word is often used in biology in a different meaning (a device with fixed and given concentrations in the INFLOWING medium, not in the bioreactor itself), it would be good to add a short explanation (just say what “chemostat” means in this work).

 We have explained “chemostats” in the following text.

 (175-177) Following the definition in [51], species with fixed concentrations are called chemostats (CS) in bond graph terminology. Such species have a constant chemical potential [22]. 

220: “Hence, a symbolic bond graph module for a cycle could be created and reused.” is “could” past or conditional? Please rephrase.

 We have rephrased the following sentence to be clearer. 

 (335-336) Hence, we created a symbolic bond graph module for a single cycle and reused this template for the other four cycles.

545: “As an improvement to our previous approach [17], the present framework overcame the aforementioned limitations: ..” these are indeed great achievements. Congratulation!

 We thank the reviewer for their congratulations.

583: “Eventually, the generated mathematical equations in the bond graph environment can be converted into CellML for simulation and reproducibility.” Would the results models again have the form of a “normal model”, or do they still look very “bond-graph like”, e.g. with non-biological components representing junctions? (And I have the same question for a (potential, future) conversion to SBML models).

 We have explained the form of the converted bond graph model in CellML in the following text.

 (822-825) The regenerated bond graph model encoded in CellML will lose its graphical structure and the model will be expressed in a system of ODEs. Since we can convert the exported bond graph ODEs into MathML format, the biochemical equations would be also expressible in SBML.

 We have explained the potential conversion to SBML models in the following text.

 (782-791) Here, we have selected models encoded in CellML because CellML can deal with models that are note purely biochemical, but the approach can be applied to models in other formats, such as SBML, as long as they can include a semantic description of the system being modelled. While symbolic templates are required to apply our approach to CellML models, this step is not required for SBML models. This is because the reactant(s)-reaction-product(s) relationships are explicitly defined within SBML models while this information is not clearly provided in CellML models. In this paper, we aimed to illustrate a possible way to convert CellML biomodels into bond graphs and automatically compose them. In future, we intend to apply the same method on SBML models in a more automated way. 

References: “Paynter H. Analysis and Design of Engineering Systems/Paynter HM;.” reference incomplete

 We have fixed the reference (reference [18]).

Fig 7: The fonts are a bit small

 We have increased the text size for Fig 8 in the new version of the manuscript (Fig 7 in the old version of the manuscript).

Fig 15: in the legend for subfigure C, maybe mention the little dip in the curve and how it is caused?

 Since we have added a third module to our composed model, some simulations in the new version differ from the old version of the manuscript. The dip in Fig 15 (old version) is not noticeable in Fig 17 (new version). Instead, we have explained the appearance of a peak in Fig 15.B (new version) in the following text.

 (663-666) The peak in the activation of MKKKP in Fig 15.B corresponds to an initial rise in MKKKP concentration from the upstream MKKK where it is immediately consumed by the downstream species to activate MKKPP and MKPP.

Finally, I would like to mention some “historical predecessors” of this work, which tried to establish ways to build biochemical network models from simple “standard elements” while taking thermodynamic feasibility into account.

Ederer M. and Gilles E.D. (2007), Thermodynamically Feasible Kinetic Models of Reaction Networks, Biophysical Journal, Volume 92, Issue 6, 1846-1857,

Stanford N.J., Lubitz T., Smallbone K., Klipp E., Mendes P., Liebermeister W. (2013), Systematic construction of kinetic models from genome-scale metabolic networks, PLoS ONE 8(11): E79195

While the present work is certainly more elegant, it may make sense to cite these earlier works.

 We have added the above-mentioned works in References 15 & 16 and cited them in the following text.

 (31-34) Several formulations and frameworks have been developed to ensure biochemical models follow the laws of thermodynamics ([15–17]) but most of them are purely mathematical and are difficult to implement for model composition.

Furthermore, the adjustment of parameters to become thermodynamically feasible seems to resemble parameter balancing (which is used in the Stanford et al paper and could also be cited).

 We have compared our approach to parameter balancing in the following text.

 (763-770) Our parameter optimisation method is similar to the parameter balancing method utilised by Stanford et al. in [16] in using the thermodynamic constants. While parameter balancing is based on assumptions about typical ranges of parameters and probability distributions [73], our parameter optimisation technique concerns the replication of the model performance with the least square error. In the future, we can utilise other techniques such as parameter balancing in our approach to incorporate the experimentally measured values of parameters and create more realistic bond graph models.

---

## [Decision Letter · Decision Letter 1]

10 May 2022

PONE-D-21-39103R1A semantics, energy-based approach to automate biomodel compositionPLOS ONE

Dear Dr. Shahidi,

Thank you for your careful revision. After feedback from all three original reviewers, I decided for a final minor revision by text edits that should further improve the presentation of your method and results. Therefore, I invite you to submit a revised version of the manuscript that may address the new suggestions by the reviewers, see below. The final decision can then be taken quickly after your re-submission.

Please submit your revised manuscript as soon as possible and by Jun 24 2022 11:59PM the latest. If you will need more time than this to complete your revisions, please reply to this message or contact the journal office at plosone@plos.org. Please include the following items when submitting your revised manuscript:A rebuttal letter that responds to each point raised by the academic editor and reviewer(s). You should upload this letter as a separate file labeled 'Response to Reviewers'.A marked-up copy of your manuscript that highlights changes made to the original version. You should upload this as a separate file labeled 'Revised Manuscript with Track Changes'.An unmarked version of your revised paper without tracked changes. You should upload this as a separate file labeled 'Manuscript'.If applicable, we recommend that you deposit your laboratory protocols in protocols.io to enhance the reproducibility of your results. Protocols.io assigns your protocol its own identifier (DOI) so that it can be cited independently in the future. For instructions see: https://journals.plos.org/plosone/s/submission-guidelines#loc-laboratory-protocols. Additionally, PLOS ONE offers an option for publishing peer-reviewed Lab Protocol articles, which describe protocols hosted on protocols.io. Read more information on sharing protocols at https://plos.org/protocols?utm_medium=editorial-email&utm_source=authorletters&utm_campaign=protocols.

We look forward to receiving your revised manuscript.

Kind regards,

Lutz Brusch, Ph.D.

Academic Editor

PLOS ONE

Journal Requirements:

Reviewers' comments:

Reviewer's Responses to Questions

**Comments to the Author**

1. If the authors have adequately addressed your comments raised in a previous round of review and you feel that this manuscript is now acceptable for publication, you may indicate that here to bypass the “Comments to the Author” section, enter your conflict of interest statement in the “Confidential to Editor” section, and submit your "Accept" recommendation.

Reviewer #1: All comments have been addressed

Reviewer #2: All comments have been addressed

Reviewer #3: All comments have been addressed

2. Is the manuscript technically sound, and do the data support the conclusions?

Reviewer #1: Yes

Reviewer #2: Yes

Reviewer #3: Yes

3. Has the statistical analysis been performed appropriately and rigorously? 

Reviewer #1: N/A

Reviewer #2: N/A

Reviewer #3: N/A

4. Have the authors made all data underlying the findings in their manuscript fully available?

Reviewer #1: Yes

Reviewer #2: Yes

Reviewer #3: Yes

5. Is the manuscript presented in an intelligible fashion and written in standard English?

Reviewer #1: Yes

Reviewer #2: Yes

Reviewer #3: Yes

6. Review Comments to the Author

Reviewer #1: The authors have improved the manuscript significantly and I recommend publication. They have addressed the majority of the points I raised, including what I perceived as critical flaws.

In terms of results, they have added a detailed Ras-Sos model to complement their other two models, ultimately enabling a biochemically meaningful model composition. They have added simulation results from the composed model demonstrating the utility of composition. These improvements address the major complaint I had with the content of the previous manuscript.

In terms of exposition, they have more clearly explained the problem of thermodynamic consistency in reaction models and how bond graph tools can address them. They have explained bond graph terminology more clearly, they have added more explanation to bond graph composition, and also explained how bond graph models relate to ODE models and simulation. They have clarified semantic annotations, ontologies and elucidated the steps of the composition pipeline. They have more directly addressed potential problems with their approach, and discuss future directions. They have discussed related work like energy-based rule-based models.

In terms of narrative and readability, they have structured the manuscript so it reads better, with introduction and summary sentences for each paragraph and section. They have also addressed run-on sentences, passive voice usage and other copy-editing issues. They have improved some of the figures and added more detail to the captions.

Reviewer #2: The new version of the text is much improved and adequately addresses my major concerns. The github repo looks good too, but could perhaps benefit from a link to BondGraphTools (or a mention that it's pip installable).

Reviewer #3: Dear authors,

Thank you for answering my comments and for the modifications you made. You addressed all the questions I raised, but some of your points could still be explained more clearly. I won't insist on any one of them. I'm listing them below for your information and leave it to you to decide which changes you would like to make.

Sincerely,

Your reviewer #3

"We have modified the following text to be clearer.

(38-40) .... an energetic and multi-physics framework that explicitly models energy to

ensure adherence to the laws of physics and is executable in multi-physics modeling."

-> For me, the term "multi-physics framework" is still a bit vague, you could briefly explain what you mean by it in this specific context; also "energetic" and "models energy" will not be clear to some readers, because "energy", without further specification, can mean many things.

We have modified the following text to be clearer.

(116-118) ... provides a reliable and consistent framework that first tracks energy transfer;

secondly ensures that reactions can only operate in the direction of decreasing chemical

potential; ...

-> Again, "energy transfer" is not clear and should be explained.

Section 2.2 explains the usage of bond graph modeling of biochemical reactions, but it remains

unclear how parameters, rate law formulae, and other data attached to the nodes will be treated

during model composition. Is an enzymatic rate law a property attached to the reaction node?

We believe this has been addressed in the comment on line 109.

-> I don't see how this is addressed in the comment on line 109, maybe you can explain this more explicitly.

-> The font size in Figure 8 is now ok, but the font size in figure 6C could still be increased

583: “Eventually, the generated mathematical equations in the bond graph environment can be

converted into CellML for simulation and reproducibility.” Would the results models again have the

form of a “normal model”, or do they still look very “bond-graph like”, e.g. with non-biological

components representing junctions? (And I have the same question for a (potential, future)

conversion to SBML models).

We have explained the form of the converted bond graph model in CellML in the following

text.

(822-825) The regenerated bond graph model encoded in CellML will lose its graphical

structure and the model will be expressed in a system of ODEs. Since we can convert the

exported bond graph ODEs into MathML format, the biochemical equations would be also

expressible in SBML.

-> Thank you for the clarification. I can see that it is expressible in SBML; but does it still have the "natural" structure of an SBML model, with concentrations changes described by stoichiometric coefficients and reaction fluxes (described in "reaction" elements), or will it just be a collection of ODEs? Please clarify.

(782-791) Here, we have selected models encoded in CellML because CellML can deal

with models that are note purely biochemical,

-> typo "note"

We have added the above-mentioned works in References 15 & 16 and cited them in the

following text.

(31-34) Several formulations and frameworks have been developed to ensure

biochemical models follow the laws of thermodynamics ([15–17]) but most of them are

purely mathematical and are difficult to implement for model composition.

-> I don't think they would be difficult to implement for model composition (at least, if some standardised rate laws are used), I think the main point here is that they HAVEN'T been implemented for model composition.

7. PLOS authors have the option to publish the peer review history of their article (what does this mean?). If published, this will include your full peer review and any attached files.

Reviewer #1: No

Reviewer #2: **Yes: **Michael Clerx

Reviewer #3: **Yes: **Wolfram Liebermeister

---

## [Author Response · Author response to Decision Letter 1]

12 May 2022

Response to Reviewers

A semantics, energy-based approach to automate biomodel composition

We thank the reviewers for taking the time to read our revised manuscript. We have included the latest suggestions from Reviewer#2 and Reviewer#3 as detailed below. The reviewer comments will be shown in black, our responses in green and quotations from the revised manuscript in blue.

Reviewer #2: 

The new version of the text is much improved and adequately addresses my major concerns. The github repo looks good too, but could perhaps benefit from a link to BondGraphTools (or a mention that it's pip installable).

We have added a link to BondGraphTools installation steps on our GitHub repository (Readme file).

Reviewer #3: 

Dear authors,

Thank you for answering my comments and for the modifications you made. You addressed all the questions I raised, but some of your points could still be explained more clearly. I won't insist on any one of them. I'm listing them below for your information and leave it to you to decide which changes you would like to make.

Sincerely,

Your reviewer #3

(38-40) .... an energetic and multi-physics framework that explicitly models energy to ensure adherence to the laws of physics and is executable in multi-physics modelling."

-> For me, the term "multi-physics framework" is still a bit vague, you could briefly explain what you mean by it in this specific context; also "energetic" and "models energy" will not be clear to some readers, because "energy", without further specification, can mean many things.

We have added an example of “multi-physics” systems and “energetic modelling” in the following text. 

(39-42) ... an energetic and multi-physics framework that explicitly models energy (expressing kinetic rate laws of biochemical reactions in terms of chemical energy level differences [17]) and ensures adherence to the laws of physics and is executable in multi-physics modelling (such as cardiomyocytes electromechanical coupling).

 (116-118) ... provides a reliable and consistent framework that first tracks energy transfer; secondly ensures that reactions can only operate in the direction of decreasing chemical potential; ...

-> Again, "energy transfer" is not clear and should be explained.

We have modified the following text for clarity.

(119-121) ... provides a reliable and consistent framework that is consistent with energy conservation; secondly ensures that reactions can only operate in the direction of decreasing chemical potential; ...

Section 2.2 explains the usage of bond graph modeling of biochemical reactions, but it remains unclear how parameters, rate law formulae, and other data attached to the nodes will be treated during model composition. Is an enzymatic rate law a property attached to the reaction node?

We believe this has been addressed in the comment on line 109.

-> I don't see how this is addressed in the comment on line 109, maybe you can explain this more explicitly.

We have demonstrated this with an example given in Fig 2 (showing how merging in bond graphs occurs graphically) and how the rate law formulae and conservation laws change during model composition in S1 Text.C. 

Yes, the enzymatic rate law is a property attached to the reaction node but instead of relating the reaction fluxes to concentrations, bond graphs relate fluxes to chemical potentials. We have discussed this in lines 40-41 of the manuscript.

-> The font size in Figure 8 is now ok, but the font size in figure 6C could still be increased

We have increased the font size in Figure 6.C.

583: “Eventually, the generated mathematical equations in the bond graph environment can be converted into CellML for simulation and reproducibility.” Would the results models again have the form of a “normal model”, or do they still look very “bond-graph like”, e.g. with non-biological components representing junctions? (And I have the same question for a (potential, future) conversion to SBML models).

We have explained the form of the converted bond graph model in CellML in the following text.

(822-825) The regenerated bond graph model encoded in CellML will lose its graphical structure and the model will be expressed in a system of ODEs. Since we can convert the exported bond graph ODEs into MathML format, the biochemical equations would be also expressible in SBML.

-> Thank you for the clarification. I can see that it is expressible in SBML; but does it still have the "natural" structure of an SBML model, with concentrations changes described by stoichiometric coefficients and reaction fluxes (described in "reaction" elements), or will it just be a collection of ODEs? Please clarify.

We believe that the conversion from bond graph models of biochemical systems to the natural structure of SBML models is possible due to the following reasons:

1. The constitutive equations for Re components (reactions) in bond graphs are expressed in terms of chemical potential (energy) differences and hence automatically account for energy conservation. These equations can be directly used as kinetic law expressions in SBML reactions too. There is no need to change the derived mathematical equations from bond graphs and they can be directly applied to SBML “reaction” elements.

2. The Ce components and TF transformers in bond graphs represent the “species” and “stoichiometry” in SBML models, respectively. Therefore, the transfer of these bond graph elements’ specific parameters to their SBML corresponding elements is conceivable.

3. As well as the reactant(s)-reaction-product(s) are extractable from SBML models, same relationships can be deduced from bond graph models of biochemical systems and inserted in SBML models.

We have modified the following text to include a summary of the above-mentioned ideas.

(825-830) The regenerated bond graph model encoded in CellML will lose its graphical structure and the model will be expressed as a system of ODEs. Since we can convert the exported bond graph ODEs into MathML format, the biochemical equations would be also expressible in SBML. The structure of such SBML models will be preserved since the required parameters, rate laws, and reactant(s)-reaction-product(s) relationships are extractable from the generated bond graph model.

(782-791) Here, we have selected models encoded in CellML because CellML can deal with models that are note purely biochemical,

-> typo "note"

We thank the reviewer for pointing out the typo and have corrected it.

(31-34) Several formulations and frameworks have been developed to ensure biochemical models follow the laws of thermodynamics ([15–17]) but most of them are purely mathematical and are difficult to implement for model composition.

-> I don't think they would be difficult to implement for model composition (at least, if some standardised rate laws are used), I think the main point here is that they HAVEN'T been implemented for model composition.

We have included the reviewer’s comment in the following text to make it clearer.

(31-35) Several formulations and frameworks have been developed to ensure biochemical models follow the laws of thermodynamics ([15–17]) but most of them are purely mathematical and are often difficult to implement for model composition due to non-standardised rate laws, and lacking an easy append/delete graphical structure.

---

## [Editor Report · Decision Letter 2]

23 May 2022

A semantics, energy-based approach to automate biomodel composition

PONE-D-21-39103R2

Dear Dr. Shahidi,

We’re pleased to inform you that your manuscript has been judged scientifically suitable for publication and will be formally accepted for publication once it meets all outstanding technical requirements.

Kind regards,

Lutz Brusch, Ph.D.

Academic Editor

PLOS ONE
---

## [Editor Report · Acceptance letter]

25 May 2022

PONE-D-21-39103R2 

A semantics, energy-based approach to automate biomodel composition 

Dear Dr. Shahidi:

I'm pleased to inform you that your manuscript has been deemed suitable for publication in PLOS ONE. Congratulations! Your manuscript is now with our production department. 

Kind regards, 

on behalf of

Dr. Lutz Brusch 

Academic Editor

PLOS ONE